# The Implicit Bias of Structured State Space Models Can Be Poisoned With Clean Labels

**Yonatan Slutzky**[*]
Tel Aviv University
slutzky1@mail.tau.ac.il

**Yotam Alexander**[*]
Tel Aviv University
yotam.alexander@gmail.com

**Noam Razin**
PLI, Princeton University
noamrazin@princeton.edu

**Nadav Cohen**
Tel Aviv University
cohennadav@tauex.tau.ac.il

## Abstract

Neural networks are powered by an implicit bias: a tendency of gradient descent to fit training data in a way that generalizes to unseen data. A recent class of neural network models gaining increasing popularity is structured state space models (SSMs). Prior work argued that the implicit bias of SSMs leads to generalization in a setting where data is generated by a low dimensional teacher. In this paper, we revisit the latter setting, and formally establish a phenomenon entirely undetected by prior work on the implicit bias of SSMs. Namely, we prove that while implicit bias leads to generalization under many choices of training data, there exist special examples whose inclusion in training completely distorts the implicit bias, to a point where generalization fails. This failure occurs despite the special training examples being labeled by the teacher, *i.e.*, having clean labels! We empirically demonstrate the phenomenon, with SSMs trained independently and as part of non-linear neural networks. In the area of adversarial machine learning, disrupting generalization with cleanly labeled training examples is known as clean-label poisoning. Given the proliferation of SSMs, we believe that delineating their susceptibility to clean-label poisoning, and developing methods for overcoming this susceptibility, are critical research directions to pursue.

## 1 Introduction

Overparameterized neural networks can fit their training data in multiple ways, some of which generalize to unseen data, while others do not. Remarkably, when the training data is fit via gradient descent (or a variant thereof), generalization tends to occur. This phenomenon—one of the greatest mysteries in modern machine learning [91, 8]—is often viewed as stemming from an *implicit bias*: a tendency of gradient descent, when applied to neural network models, to fit training data in a way that complies with common data-generating distributions. The latter view was formalized for several neural network models and data-generating distributions [55, 77, 27, 64].

A recent class of neural network models gaining significant popularity is *structured state space models* (*SSMs*). SSMs are often regarded as a computationally efficient alternative to transformers [85], and underlie prominent neural networks such as S4 [24], Mamba [22], LRU [58], Mega [48], S5 [75] and more [62, 14]. The implicit bias of SSMs, *i.e.*, of gradient descent over SSMs, was formally studied in prior works, *e.g.*, Emami et al. [16], Cohen-Karlik et al. [11, 12]. Notable among these is Cohen-Karlik et al. [12], which considered a setting where data is generated by a low dimensional teacher SSM, and gradient flow (gradient descent with infinitesimally small step size) over a high

---

[*]Equal contribution.

dimensional student SSM fits training data comprising infinitely many sequences of a certain length.[1] In this setting, the student SSM can fit the training data in multiple ways, some of which generalize to sequences longer than those seen in training, while others do not. It was shown in Cohen-Karlik et al. [12] that under mild conditions, an implicit bias leads to generalization.

In this paper, we revisit the setting of Cohen-Karlik et al. [12], with one key exception: rather than training data comprising infinitely many sequences, we consider the realistic case where the number of sequences is finite. Surprisingly, our theory and experiments reveal a phenomenon entirely undetected by prior works on the implicit bias of SSMs. Namely, we find that while implicit bias leads the student SSM to generalize under many choices of training sequences, there exist special sequences which if included in training completely distort the implicit bias, resulting in the student SSM failing to generalize. This failure to generalize takes place despite the fact that the special sequences are labeled by the teacher SSM, *i.e.*, they have clean labels! In the area of adversarial machine learning, the phenomenon of generalization being disrupted by training instances with clean labels is known as *clean-label poisoning*, and has received significant attention, both empirically [71, 33] and theoretically [79, 6]. To our knowledge, the current paper is the first to formally prove susceptibility of SSMs to clean-label poisoning.

Our theoretical analysis comprises two contributions. First, is a dynamical characterization of gradient flow over an SSM, trained individually or as part of a non-linear neural network. The dynamical characterization reveals that *greedy low rank learning* [43, 78, 65, 66]—a sufficient condition for generalization with a low dimensional teacher SSM—is implicitly induced under many, but not all, choices of training sequences. Our second theoretical contribution builds on our dynamical characterization for a fine-grained analysis of gradient flow over an SSM, employing an advanced tool from dynamical systems theory: a *non-resonance linearization theorem* [70]. The analysis proves that there exist situations where: *(i)* training a student SSM on sequences labeled by a low dimensional teacher SSM exhibits an implicit bias that leads to generalization; and *(ii)* adding to the training set special sequences, also labeled by the teacher SSM (*i.e.*, that also have clean labels), entirely distorts the implicit bias, to an extent where generalization fails.

We corroborate our theory via experiments spanning a wide range of settings: from those covered by our theory, to real-world (non-synthetic) settings comprising SSM-based S4 [24], Mamba-2 [14] and LRU [58] neural networks trained on the CIFAR-10 dataset [38]. The experiments validate both our dynamical characterization and the susceptibility of SSMs to clean-label poisoning. In light of the growing prominence of SSMs, particularly in large language models, we believe that delineating this susceptibility, and developing methods for overcoming it, are critical research directions to pursue.

## 2 Preliminaries

### 2.1 Notation

We use non-boldface lowercase letters for denoting scalars (*e.g.*, $\alpha \in \mathbb{R}$, $d \in \mathbb{N}$), boldface lowercase letters for denoting vectors (*e.g.*, $\mathbf{x} \in \mathbb{R}^d$), and non-boldface uppercase letters for denoting matrices (*e.g.*, $A \in \mathbb{R}^{d,d}$). We denote by $\mathbf{1}$ an all-ones vector and by $\mathbf{0}$ an all-zeros vector, with their dimensions to be inferred from context. For $d \in \mathbb{N}$, we let $[d] := \{1, 2, \ldots, d\}$. For $d \in \mathbb{N}$ and $i \in [d]$, we denote by $\mathbf{e}_i$ the $i$'th standard basis vector (*i.e.*, a vector holding one in entry $i$ and zeros elsewhere) of dimension $d$, where the dimension is omitted from the notation and should be inferred from context. We identify scalar sequences of finite lengths with vectors.

### 2.2 Structured State Space Models (SSMs)

A *structured state space model* (*SSM*) *of dimension* $d \in \mathbb{N}$ is parameterized by three matrices: $A \in \mathbb{R}^{d,d}$, a *state transition matrix*, which conforms to a predefined structure (*e.g.*, is constrained to be diagonal); $B \in \mathbb{R}^{d,1}$, an *input matrix*; and $C \in \mathbb{R}^{1,d}$, an *output matrix*. Given the values of $A$, $B$ and $C$, the SSM realizes a mapping $\phi_{(A,B,C)}(\cdot)$ that receives as input a length $k$ scalar sequence $\mathbf{x} \in \mathbb{R}^k$, for any $k \in \mathbb{N}$, and produces as output a scalar $y \in \mathbb{R}$ equal to the last element of the sequence $\mathbf{y} \in \mathbb{R}^k$ defined through the following recursive formula:

$$\mathbf{s}_{k'} = A\mathbf{s}_{k'-1} + Bx_{k'} \ \ , \ \ y_{k'} = C\mathbf{s}_{k'} \ \ , \ \ k' \in [k], \tag{1}$$

---

[1]More precisely, the training data is formed from a continuous (Gaussian) distribution of sequences having a certain length, all labeled by the teacher SSM.

where $(\mathbf{s}_{k'} \in \mathbb{R}^d)_{k' \in [k] \cup \{0\}}$ is a sequence of *states*, and $\mathbf{s}_0 = \mathbf{0}$. It is straightforward to show that the mapping $\phi_{(A,B,C)}(\cdot)$ is fully determined by the sequence $(CA^{k'}B)_{k'=0}^{\infty}$, known as the *impulse response* of the SSM. In particular, for any $k \in \mathbb{N}$ and $\mathbf{x} \in \mathbb{R}^k$:

$$y = \phi_{(A,B,C)}(\mathbf{x}) = (CA^{k-1}B, \ldots, CAB, CB)\mathbf{x} = \sum\nolimits_{k'=0}^{k-1} CA^{k'}B \cdot x_{k-k'} \,. \qquad (2)$$

For convenience, we often identify an SSM with the triplet $(A, B, C)$ holding its parameter matrices, and regard the (single column) matrices $B$ and $C^{\top}$ as vectors.

Perhaps the most common form of structure imposed on SSMs is *diagonality* [25, 28, 58, 48, 22]. Accordingly, unless stated otherwise, we assume that the state transition matrix $A$ of an SSM is diagonal.

Some of our results will account for SSMs that are part of non-linear neural networks, or more specifically, for SSMs whose output undergoes a transformation $\sigma(\cdot, \mathbf{w})$, where: $\sigma : \mathbb{R} \times \mathcal{W} \to \mathbb{R}$ is some differentiable mapping; $\mathcal{W}$ is some Euclidean space, regarded as a parameter space; and $\mathbf{w} \in \mathcal{W}$, regarded as a parameter vector. Given values for $A$, $B$, $C$ and $\mathbf{w}$, such a neural network realizes the mapping $\phi_{(A,B,C),\mathbf{w}}(\cdot) := \sigma(\phi_{(A,B,C)}(\cdot), \mathbf{w})$. This architecture (namely, an SSM followed by a parametric transformation) is ubiquitous among SSM-based neural networks (see, for example, Gu et al. [24], Gupta et al. [28], Gu et al. [25]).

## 2.3 Teacher-Student Setting

We consider the *teacher-student* setting of Cohen-Karlik et al. [12], described hereinbelow. Data is labeled by a *teacher* SSM $(A^*, B^*, C^*)$ of dimension $d^* \in \mathbb{N}$, *i.e.*, the ground truth label of $\mathbf{x} \in \mathbb{R}^k$, for any $k \in \mathbb{N}$, is $\phi_{(A^*,B^*,C^*)}(\mathbf{x}) \in \mathbb{R}$. For some $n \in \mathbb{N}$ and $\kappa \in \mathbb{N}_{>2d^*}$, we are given a training set $\mathcal{S}$ comprising $n$ labeled sequences of length $\kappa$, *i.e.*, $\mathcal{S} := \left( (\mathbf{x}^{(i)}, y^{(i)}) \right)_{i=1}^{n}$, where $\mathbf{x}^{(i)} \in \mathbb{R}^{\kappa}$ and $y^{(i)} = \phi_{(A^*,B^*,C^*)}(\mathbf{x}^{(i)})$ for every $i \in [n]$. A *student* SSM $(A, B, C)$ of dimension $d \in \mathbb{N}_{>\kappa}$ is trained by minimizing the square loss over $\mathcal{S}$, referred to as the *training loss*:

$$\ell_{\mathcal{S}}(A, B, C) := \frac{1}{n} \sum\nolimits_{i=1}^{n} \left( y^{(i)} - \phi_{(A,B,C)}(\mathbf{x}^{(i)}) \right)^2 . \qquad (3)$$

Optimization is implemented via *gradient flow*, which is formally equivalent to gradient descent with infinitesimally small step size (learning rate), and was shown to well-approximate gradient descent with moderately small step size [15]. The dynamics of gradient flow are, for $t \in \mathbb{R}_{\geq 0}$:

$$(\dot{A}(t), \dot{B}(t), \dot{C}(t)) = -\nabla \ell_{\mathcal{S}}(A(t), B(t), C(t)) \,, \qquad (4)$$

where $(\dot{A}(t), \dot{B}(t), \dot{C}(t)) := \frac{d}{dt}(A(t), B(t), C(t))$, and $(A(\cdot), B(\cdot), C(\cdot))$ is a curve representing the optimization trajectory. Generalization of the student at time $t \in \mathbb{R}_{\geq 0}$ of optimization is measured by the extent to which $\phi_{(A(t),B(t),C(t))}(\cdot)$ approximates $\phi_{(A^*,B^*,C^*)}(\cdot)$, not only over input sequences of length $\kappa$ as used for training, but also over input sequences of other lengths. This accounts not only for in-distribution generalization as considered in classical machine learning theory [72], but also for out-of-distribution generalization (extrapolation) as prevalent in modern machine learning [45]. Formally, in line with Equation (2), generalization is quantified by how close the first $k$ elements of the student's impulse response are to the first $k$ elements of the teacher's impulse response, for different values of $k$.

**Definition 1.** The *generalization error of the student SSM over sequence length $k$* is:

$$\max_{k' \in \{0,1,\ldots,k-1\}} \left| BA^{k'}C - B^*(A^*)^{k'}C^* \right| . \qquad (5)$$

Clearly, there exist assignments for $(A, B, C)$ with which the training loss $\ell_{\mathcal{S}}(\cdot)$ is minimized (*i.e.*, equals zero) and the student SSM perfectly generalizes over any sequence length $k$.[2] On the other hand, it was shown in Cohen-Karlik et al. [12] that, regardless of the size of the training set $\mathcal{S}$ (*i.e.*, of $n$) and the input sequences it comprises (namely, $(\mathbf{x}^{(i)})_{i=1}^{n}$), there exist assignments for $(A, B, C)$ with which the training loss $\ell_{\mathcal{S}}(\cdot)$ is minimized, and yet the student has arbitrarily high generalization error over sequence lengths beyond $\kappa$, *e.g.*, over sequence length $\kappa + 1$ (for completeness, we prove this fact in Appendix C). The latter two facts together imply that if minimization of the training

---

[2]This is the case, for example, if $A$, $B$ and $C$ are respectively attained by padding $A^*$, $B^*$ and $C^*$ with zeros on the right and/or bottom.

loss $\ell_S(\cdot)$ via gradient flow (Equation (4)) produces an assignment for $(A, B, C)$ with which the student generalizes over sequence lengths beyond $\kappa$, it must be an outcome of implicit bias. The main result in Cohen-Karlik et al. [12] states that if the training set $S$ is infinite and each entry of each input sequence $\mathbf{x}^{(i)}$ is independently drawn from the standard normal distribution,[3] then under mild conditions the implicit bias of gradient flow is such that it convergences to a solution which generalizes over any sequence length $k$.

In this paper, we focus on the realistic case where the training set $S$ is finite. Surprisingly, our theory and experiments (Sections 3 and 4, respectively) reveal a phenomenon completely undetected by Cohen-Karlik et al. [12], and any other work we are aware of on the implicit bias of SSMs.

## 3 Theoretical Analysis

### 3.1 Dynamical Characterization

In this subsection we derive a dynamical characterization of gradient flow over an SSM, trained individually or as part of a non-linear neural network. The dynamical characterization reveals that *greedy low rank learning* [69, 39, 4, 43, 65, 66]—a sufficient condition for generalization with a low dimensional teacher SSM—is implicitly induced under many, but not all, choices of training sequences. Section 3.2 will build on the dynamical characterization to prove that the implicit bias of SSMs can be poisoned with clean labels.

Our dynamical characterization applies to a setting more general than that laid out in Section 2.3. Specifically, it applies to the same setting, with two exceptions: *(i)* the student SSM is potentially embedded in a non-linear neural network, *i.e.*, the mapping $\phi_{(A,B,C)}(\cdot)$ is replaced by $\phi_{(A,B,C),\mathbf{w}}(\cdot)$ as defined in Section 2.2; and *(ii)* the training labels $(y^{(i)})_{i=1}^n$ need not be assigned by a teacher SSM, *i.e.*, they may be arbitrary. We denote the resulting training loss—a generalization of $\ell_S(\cdot)$ from Equation (3)—by $\tilde{\ell}_S(\cdot)$:

$$\tilde{\ell}_S(A, B, C, \mathbf{w}) := \frac{1}{n} \sum_{i=1}^n \left(y^{(i)} - \phi_{(A,B,C),\mathbf{w}}(\mathbf{x}^{(i)})\right)^2. \quad (6)$$

Proposition 1 below establishes our dynamical characterization: equations of motion for the (diagonal) entries of $A$ during gradient flow over $\tilde{\ell}_S(\cdot)$.

**Proposition 1.** *Consider optimization of the generalized loss $\tilde{\ell}_S(\cdot)$ defined in Equation (6) via gradient flow. That is, consider, for any $t \in \mathbb{R}_{\geq 0}$:*

$$(\dot{A}(t), \dot{B}(t), \dot{C}(t), \dot{\mathbf{w}}(t)) = -\nabla\tilde{\ell}_S(A(t), B(t), C(t), \mathbf{w}(t)),$$

*where $(\dot{A}(t), \dot{B}(t), \dot{C}(t), \dot{\mathbf{w}}(t)) := \frac{d}{dt}(A(t), B(t), C(t), \mathbf{w}(t))$, and $(A(\cdot), B(\cdot), C(\cdot), \mathbf{w}(\cdot))$ is a curve representing the optimization trajectory. For $j \in [d]$, denote by $a_j(\cdot)$ the j'th diagonal entry of $A(\cdot)$, by $b_j(\cdot)$ the j'th entry of $B(\cdot)$, and by $c_j(\cdot)$ the j'th entry of $C(\cdot)$. Assume that the training sequence length $\kappa$ is greater than or equal to two. For $l \in [\kappa]$ and $i \in [n]$, denote the l'th element of the i'th training sequence $\mathbf{x}^{(i)}$ by $x_l^{(i)}$. Then, for any $j \in [d]$ and $t \in \mathbb{R}_{\geq 0}$:*

$$\dot{a}_j(t) := \frac{d}{dt}a_j(t) = b_j(t)c_j(t)\sum_{l=0}^{\kappa-2}\gamma^{(l)}(t) \cdot a_j(t)^l, \quad (7)$$

*where:*

$$\gamma^{(l)}(t) := \frac{2(l+1)}{n}\sum_{i=1}^n \delta^{(i)}(t)\xi^{(i)}(t)x_{\kappa-l-1}^{(i)}, \quad (8)$$

*with:*

$$\delta^{(i)}(t) := y^{(i)} - \phi_{(A(t),B(t),C(t)),\mathbf{w}(t)}(\mathbf{x}^{(i)}), \; \xi^{(i)}(t) := \frac{\partial}{\partial z}\sigma(z, \mathbf{w}(t))\big|_{z=\phi_{(A(t),B(t),C(t))}(\mathbf{x}^{(i)})}. \; (9)$$

*Proof sketch (proof in Appendix D).* The result readily follows from differentiation of $\tilde{\ell}_S(\cdot)$ (Equation (6)) with respect to each diagonal entry of $A$. $\qquad\square$

---

[3]Formally, this condition means that the training loss $\ell_S(\cdot)$ is the expected value of $(y - \phi_{(A,B,C)}(\mathbf{x}))^2$, where the entries of $\mathbf{x}$ are independently drawn from the standard normal distribution, and $y = \phi_{(A^*,B^*,C^*)}(\mathbf{x})$.

### 3.1.1 Interpretation

Proposition 1 implies that during gradient flow, the motion of $a_j(\cdot)$—the $j$'th diagonal entry of the state transition matrix $A(\cdot)$—is given by a degree $\kappa - 2$ polynomial in $a_j(\cdot)$, where the coefficients of the polynomial are time-varying. In particular, at time $t \in \mathbb{R}_{\geq 0}$ of optimization, the coefficient of the $l$'th power in the polynomial, for $l \in \{0, 1, \dots, \kappa - 2\}$, is a product of two factors: *(i)* $\gamma^{(l)}(t)$, which depends on the power $l$ but not on the entry index $j$; and *(ii)* $b_j(t)c_j(t)$ (the $j$'th entry of the input matrix $B(\cdot)$ times the $j$'th entry of the output matrix $C(\cdot)$), which does not depend on the power $l$ but does depend on the entry index $j$.

**Different training sets admit greedy learning.** Consider the case where $A(\cdot)$ emanates from standard near-zero initialization [20, 30, 59], *i.e.*, where $a_j(0) \approx 0$ for all $j$. If the factor $\gamma^{(0)}(\cdot)$ is small throughout—as is the case, *e.g.*, if the penultimate ($\kappa - 1$'th) element of each training sequence $\mathbf{x}^{(i)}$ is small (see Equation (8))—then the constant coefficient (*i.e.*, the coefficient of the zeroth power) in the polynomial determining the motion of $a_j(\cdot)$ is negligible. The dynamics of $(a_j(\cdot))_{j=1}^{d}$ then exhibit greedy learning, similarly to the dynamics of various quantities in various types of neural networks [69, 39, 4, 43, 65, 66]. Namely, $(a_j(\cdot))_{j=1}^{d}$ all progress slowly at first, following near-zero initialization, and then, whenever an entry reaches a critical threshold, it starts moving rapidly—see empirical demonstrations in Figure 2 and Appendix I.1. The greedy learning of $(a_j(\cdot))_{j=1}^{d}$ implies a greedy low rank learning of the state transition matrix. More specifically, it implies a tendency to fit training data with $A$ having low rank, meaning a tendency to generalize if data is generated by a low dimensional teacher SSM.

**Certain training sequences impede greedy learning.** In stark contrast to the above, if the training sequences $(\mathbf{x}^{(i)})_{i=1}^{n}$ are such that the factor $\gamma^{(0)}(\cdot)$ is not small—as can be the case, *e.g.*, if there is a training sequence $\mathbf{x}^{(i)}$ in which the last elements are relatively large—then the polynomials determining the motions of $(a_j(\cdot))_{j=1}^{d}$ have non-negligible constant coefficients, and greedy low rank learning will generally *not* take place—see empirical demonstrations in Figure 2 and Appendix I.1.

### 3.2 Clean-Label Poisoning

In this subsection we employ the dynamical characterization from Section 3.1 for a fine-grained analysis of gradient flow over an SSM. The analysis considers a teacher-student setting as in Section 2.3, and proves existence of situations where: *(i)* training a student SSM on sequences labeled by a low dimensional teacher SSM exhibits an implicit bias that leads to generalization; and *(ii)* adding to the training set certain sequences, also labeled by the teacher SSM (*i.e.*, that also have clean labels), entirely distorts the implicit bias, to an extent where generalization fails. To our knowledge, this constitutes the first formal proof of susceptibility of SSMs to clean-label poisoning. Facilitating the analysis is an advanced tool from dynamical systems theory—a *non-resonance linearization theorem* [70]—which may be of independent interest.

The teacher-student setting considered in this subsection is characterized by Assumptions 1 and 2 below. While this setting is specific, it fulfills the purpose of proving existence of situations where SSMs are susceptible to clean-label poisoning. Moreover, as discussed in Appendix B, our theory can be adapted to account for different extensions of the setting, *i.e.*, for different relaxations of Assumptions 1 and 2. Empirically, we demonstrate in Section 4 that SSMs are susceptible to clean-label poisoning in a wide range of settings: from the specific setting considered in this subsection, to real-world (non-synthetic) settings comprising SSM-based S4 [24], Mamba-2 [14] and LRU [58] neural networks trained on the CIFAR-10 dataset [38].

**Assumption 1.** *The teacher SSM is of dimension $d^* = 2$, and its parameter matrices are given by:*

$$A^* = \begin{pmatrix} 1 & 0 \\ 0 & 0 \end{pmatrix} \quad , \quad B^* = \begin{pmatrix} 1 & \sqrt{d-1} \end{pmatrix}^\top \quad , \quad C^* = \begin{pmatrix} 1 & \sqrt{d-1} \end{pmatrix} . \tag{10}$$

**Assumption 2.** *The input and output matrices of the student SSM, $B$ and $C$, are respectively fixed at $\mathbf{1}$ and $\mathbf{1}^\top$ throughout training.*

Under Assumptions 1 and 2, generalization is straightforward to attain: all that is needed in order for the student SSM to achieve low generalization error over any sequence length (Definition 1) is that one of the diagonal entries of its state transition matrix $A$ be sufficiently close to one while the rest are sufficiently close to zero.

Proposition 2 below proves that generalization takes place in cases where: each training sequence has zeros in its last two elements and non-negative values elsewhere; and at least one training sequence is

not identically zero. Underlying the proof is the dynamical characterization from Section 3.1: with the last elements of each training sequence being small, the characterization establishes greedy low rank learning, which in turn implies generalization (see interpretation in Section 3.1.1).

**Proposition 2.** *Consider the teacher-student setting of Section 2.3, subject to Assumptions 1 and 2. Suppose that for every $i \in [n]$, the training sequence $\mathbf{x}^{(i)}$ has zeros in its last two elements and non-negative values elsewhere. Suppose also that $\mathbf{x}^{(i)} \neq \mathbf{0}$ for some $i \in [n]$. Then, for any $k \in \mathbb{N}$, $\epsilon \in \mathbb{R}_{>0}$ and $\delta \in \mathbb{R}_{>0}$, there exists an open set $\mathcal{I}$ of arbitrarily small initializations for the student SSM,[4] and a corresponding time $t \in \mathbb{R}_{>0}$, such that gradient flow initialized in $\mathcal{I}$ reaches at time $t$ a point where the training loss is at most $\delta$ and the generalization errors over sequence lengths $1, 2, \ldots, k$ are at most $\epsilon$.*

*Proof sketch (proof in Appendix E).* The proof specializes the dynamical characterization from Section 3.1 per Assumptions 1 and 2, and per the conditions posed on the training sequences. Choosing the set $\mathcal{I}$ to be sufficiently concentrated around the origin, the specialized characterization implies that gradient flow initialized in $\mathcal{I}$ exhibits greedy low rank learning (see interpretation of the characterization in Section 3.1.1). The greedy low rank learning leads a single diagonal entry of the state transition matrix $A$ to reach the vicinity of one while the rest of the diagonal entries remain near zero. As stated in the text following Assumptions 1 and 2, this ensures generalization. $\square$

As stated in the text following Assumptions 1 and 2, under these assumptions, generalization is straightforward to attain. Indeed, Proposition 2 above proves that generalization takes place even in cases where the training set $\mathcal{S}$ entails little information, *e.g.*, in the case where $\mathcal{S}$ includes $\mathbf{e}_1$ (one followed by zeros) as a sole training sequence. Theorem 1 below proves that in this latter case, despite generalization being straightforward to attain, adding to $\mathcal{S}$ a single cleanly labeled example—or more specifically, a single carefully selected sequence $\mathbf{x}^\dagger$ of moderate (Euclidean) norm labeled by the teacher SSM—can entirely disrupt generalization.

In the context of Theorem 1, disrupting generalization requires preventing greedy low rank learning. This is achieved, in accordance with the dynamical characterization from Section 3.1 (see its interpretation in Section 3.1.1), by selecting a sequence $\mathbf{x}^\dagger$ whose last elements are relatively large. To prove that such selection indeed disrupts generalization, Theorem 1 relies on an advanced tool from dynamical systems theory—a non-resonance linearization theorem [70]—which may be of independent interest.

Theorem 1, albeit specific, constitutes what is to our knowledge the first formal proof of susceptibility of SSMs to clean-label poisoning. In Appendix B we discuss different extensions of Theorem 1, including an extension that allows for any training set $\mathcal{S}$ permitted by Proposition 2 (*i.e.*, for any choice of training sequences $(\mathbf{x}^{(i)})_{i=1}^n$ where: for every $i \in [n]$, $\mathbf{x}^{(i)}$ has zeros in its last two elements and non-negative values elsewhere; and $\mathbf{x}^{(i)} \neq \mathbf{0}$ for some $i \in [n]$), at the expense of the added sequence $\mathbf{x}^\dagger$ potentially having large norm. Empirically, we demonstrate in Section 4 that the conclusion of Theorem 1 carries over to a wide range of settings, including ones where an SSM is part of a non-linear neural network.

**Theorem 1.** *Consider the teacher-student setting of Section 2.3, subject to Assumptions 1 and 2. Suppose the training sequence length and the dimension of the student SSM respectively satisfy $\kappa \in \{7, 9, 11, \ldots\}$ and $d \geq 8$. Define the sequences $\mathbf{x} := \mathbf{e}_1 \in \mathbb{R}^\kappa$ and $\mathbf{x}^\dagger := \mathbf{e}_{\kappa-1} \in \mathbb{R}^\kappa$. Denote by $y$ and $y^\dagger$ the labels respectively assigned to $\mathbf{x}$ and $\mathbf{x}^\dagger$ by the teacher SSM, i.e., $y := \phi_{(A^*, B^*, C^*)}(\mathbf{x})$ and $y^\dagger := \phi_{(A^*, B^*, C^*)}(\mathbf{x}^\dagger)$. Suppose the training set $\mathcal{S}$ includes $(\mathbf{x}, y)$ as a sole example. Let $k \in \mathbb{N}_{\geq \kappa+2}$ and $\epsilon \in \mathbb{R}_{>0}$. Then, there exists an open set $\mathcal{I}$ of arbitrarily small initializations for the student SSM,[4] such that with any initialization in $\mathcal{I}$:*

- *gradient flow converges to a point where the training loss is minimal (i.e., equals zero) and the generalization errors over sequence lengths $1, 2, \ldots, k$ are at most $\epsilon$; and*

- *appending to $\mathcal{S}$ the (cleanly labeled) example $(\mathbf{x}^\dagger, y^\dagger)$ leads gradient flow to converge to a point where the training loss is minimal and the generalization error over sequence length $k$ is at least $\min\{0.1, (9d)^{-1}(1 - (0.6)^{1/(\kappa-1)})\}$.*

---

[4]That is, for any neighborhood $\mathcal{N}$ of the origin in the space of diagonal matrices in $\mathbb{R}^{d,d}$ (the latter space is identified with $\mathbb{R}^d$), there exists an open subset $\mathcal{I} \subset \mathcal{N}$.

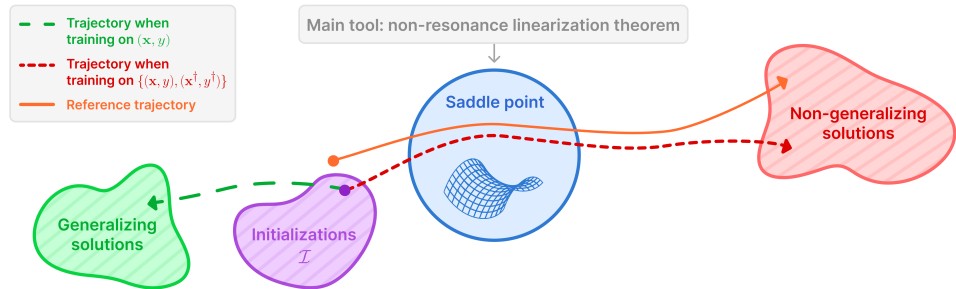

Figure 1: Illustration of the main ideas behind the proof of Theorem 1. See proof sketch for an annotation.

*Proof sketch (proof in Appendix F).* Figure 1 illustrates the main ideas behind the proof. Below is a description of these ideas, along with an annotation of the figure.

As stated in the text following Assumptions 1 and 2, a sufficient condition for the student SSM to achieve low generalization errors is that one of the diagonal entries of its state transition matrix $A$ be close to one while the rest are close to zero. The proof shows that this condition is also necessary, thus the set labeled "generalizing solutions" in Figure 1 is a neighborhood of one-hot assignments for the diagonal entries of $A$. Next, similarly to the proof of Proposition 2, the dynamical characterization from Section 3.1 is used to establish that when training on $(\mathbf{x}, y)$, under choices of $\mathcal{I}$ (set of initializations) sufficiently concentrated around the origin, a gradient flow trajectory emanating from $\mathcal{I}$ exhibits greedy low rank learning, and thus converges to the set of generalizing solutions—as illustrated in Figure 1.

For analyzing the behavior of gradient flow when training on $\{(\mathbf{x}, y), (\mathbf{x}^\dagger, y^\dagger)\}$, the proof makes use of the fact that the last elements of $\mathbf{x}^\dagger$ are relatively large. In accordance with the dynamical characterization from Section 3.1 (see its interpretation in Section 3.1.1), this fact implies that greedy low rank learning does not take place. The proof identifies reference trajectories of gradient flow (when training on $\{(\mathbf{x}, y), (\mathbf{x}^\dagger, y^\dagger)\}$) that converge to non-generalizing solutions; one such reference trajectory is illustrated in Figure 1. These reference trajectories emanate from near-zero initializations that cannot be included in $\mathcal{I}$, since they lead gradient flow to converge to non-generalizing solutions even when training only on $(\mathbf{x}, y)$. However, the proof shows that $\mathcal{I}$ can consist of initializations near those of reference trajectories, since gradient flow emanating from such $\mathcal{I}$: *(i)* converges to a generalizing solution when training on $(\mathbf{x}, y)$; and *(ii)* closely tracks a reference trajectory when training on $\{(\mathbf{x}, y), (\mathbf{x}^\dagger, y^\dagger)\}$, resulting in convergence to a non-generalizing solution—as illustrated in Figure 1. This concludes the proof.

The main technical challenge faced by the proof lies in item *(ii)* above, namely, in establishing that when training on $\{(\mathbf{x}, y), (\mathbf{x}^\dagger, y^\dagger)\}$, an initialization near that of a reference trajectory leads gradient flow to closely track the reference trajectory. Since the training loss is non-convex, gradient flow trajectories may diverge from one another exponentially fast. Establishing that a reference trajectory is tracked thus requires sharp bounds on convergence times. The crux of the challenge is to derive such bounds, as trajectories pass near saddle points, and a-priori, may not escape these saddle points sufficiently fast. To show that saddle points are escaped swiftly, the proof employs an advanced tool from dynamical systems theory which may be of independent interest: a non-resonance linearization theorem [70]. Namely, rather than directly analyzing trajectories in the vicinity of a saddle point, the proof constructs linear approximations, and uses the non-resonance linearization theorem to show that the linear approximations are sufficiently accurate, which in turn implies that the trajectories escape the saddle point sufficiently fast. The non-resonance linearization theorem requires the spectrum of the Hessian of the training loss to be free of certain algebraic dependencies known as resonances. If these resonances are absent—which the proof shows to be the case—the non-resonance linearization theorem provides guarantees on the accuracy of linear approximations that are far better than guarantees attainable via standard smoothness arguments. □

## 4  Experiments

This section presents experiments corroborating our theory. It is organized as follows. Section 4.1 demonstrates the dynamical characterization we derived (in Section 3.1), showcasing that optimization of an SSM implicitly induces greedy low rank learning (a sufficient condition for generalization with a low dimensional teacher SSM) under some, but not all, choices of training sequences. Section 4.2 then demonstrates the clean-label poisoning phenomenon we established (in Section 3.2), by showing

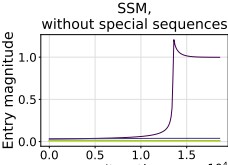
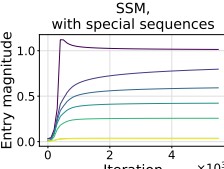
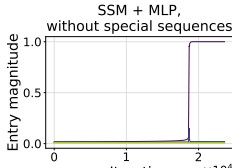
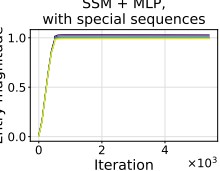

Figure 2: Demonstration of the dynamical characterization from Section 3.1: optimization of an SSM—trained individually or as part of a non-linear neural network—implicitly induces greedy learning of the diagonal entries of the state transition matrix $A$ under some, but not all, choices of training sequences. First (leftmost) plot shows the magnitudes of the entries of $A$ throughout the iterations of gradient descent, in a case where an overparameterized student SSM of dimension $d = 10$ is trained individually on a training set labeled by a teacher SSM of dimension $d^* = 1$, and the training set does not include "special" sequences, *i.e.*, sequences in which the last elements are relatively large. Second plot portrays a scenario that is identical, except that special sequences are included in the training set. Third and fourth plots adhere to the descriptions of first and second plots, respectively, except that the student SSM is trained along with a successive multi-layer perceptron (non-linear neural network), and the teacher SSM is followed by a (fixed) multi-layer perceptron. Notice that, with and without a multi-layer perceptron, greedy learning takes place when special sequences are excluded, and does not take place when they are included. For further experiments (including teacher SSMs of higher dimension, and additional variations) and visualizations (including the effective rank of $A$, and additional quantities, throughout optimization) see Appendix I.1. For further details see Appendix J.1.

that adding special cleanly labeled sequences to the training set of an SSM can completely ruin its generalization. Code for reproducing our experiments can be found at `https://github.com/YoniSlutzky98/imp-bias-ssm-poison`.

## 4.1 Dynamical Characterization

As discussed in Section 3.1.1, the dynamical characterization in Proposition 1 (Equations (7) to (9)) implies that optimization of an SSM—trained individually or as part of a non-linear neural network—implicitly induces greedy learning of the diagonal entries of the state transition matrix $A$ under some, but not all, choices of training sequences. For example, if the last elements of each training sequence are small then greedy learning takes place, and if there are training sequences in which the last elements are relatively large then greedy learning may not take place. Figure 2 empirically demonstrates this, for a standalone SSM as well as one included in a non-linear neural network. Further experiments are reported in Appendix I.1.

## 4.2 Clean-Label Poisoning

Theorem 1 proved existence of situations where clean-label poisoning of an SSM takes place, *i.e.*, situations where: *(i)* training a student SSM on sequences labeled by a low dimensional teacher SSM exhibits an implicit bias that leads to generalization; and *(ii)* adding to the training set special sequences, also labeled by the teacher SSM (*i.e.*, that also have clean labels), entirely distorts the implicit bias, to an extent where generalization fails. Table 1 empirically demonstrates clean-label poisoning of SSMs in three different teacher-student settings: the setting of Theorem 1; a standalone SSM setting beyond Theorem 1 (meaning it does not satisfy the assumptions of Theorem 1, *e.g.*, it includes multiple training sequences, and learning of the input matrix $B$ and output matrix $C$); and a setting where an SSM is part of a non-linear neural network.

We further demonstrate clean-label poisoning of SSMs in real-world (non-synthetic) settings comprising SSM-based S4 [24], Mamba-2 [14] and LRU [58] neural networks trained on the CIFAR-10 dataset [38]. In these settings we do not have access to a teacher (*i.e.*, to a ground truth labeling mapping), and accordingly, cleanly labeled poisonous examples are generated from CIFAR-10 examples by introducing human-imperceptible additive noise to input sequences, while keeping their labels intact. The noise added to an input sequence is rendered to have relatively large last elements—in line with our theory[5]—using an adapted version of the Gradient Matching method from Geiping et al. [18] (see Appendix J.3 for details). Adding to the training set cleanly labeled poisonous examples as described leads to significant deterioration in the generalization of the SSM-based neural networks—see Table 2.

We believe the susceptibility of SSMs to clean-label poisoning goes far beyond the demonstrations herein. In light of the growing prominence of SSMs, particularly in large language models, delineating this susceptibility, and developing methods for overcoming it, are of prime importance.

---

[5]Our theory does not suggest that relatively large last elements are necessary for clean-label poisoning of SSMs in these real-world settings. Accordingly, it may be possible for the noise to have a different pattern.

Table 1: Demonstration of clean-label poisoning of SSMs in three different teacher-student settings: the setting of Theorem 1; a standalone SSM setting beyond Theorem 1 (meaning it does not satisfy the assumptions of Theorem 1, *e.g.*, it includes multiple training sequences, and learning of the input matrix $B$ and output matrix $C$); and a setting where an SSM is part of a non-linear neural network, *i.e.*, is followed by a multi-layer perceptron. In each setting, a high dimensional student is trained until convergence (namely, until the training loss is lower than 0.01), and data is generated (*i.e.*, sequences are labeled) by a low dimensional teacher of the same architecture as the student. Reported are generalization errors (each averaged over four random seeds) for two training sets per setting: *(i)* a training set that does not include "special" sequences, *i.e.*, sequences in which the last elements are relatively large; and *(ii)* a training set obtained by adding to the former special sequences along with the (clean) labels assigned to them by the teacher. In the first two settings (SSMs trained independently) generalization errors are measured via impulse responses, as defined in Definition 1. In the third setting (SSM trained as part of non-linear neural network) generalization errors are measured using a held-out test set. All reported generalization errors are normalized (scaled) such that a zero mapping corresponds to a value of one. Notice that across all settings, special training sequences significantly deteriorate generalization. For further experiments and details see Appendices I.2 and J.2, respectively.

| Setting | Without special sequences | With special sequences |
|---|---|---|
| Per Theorem 1 | $1.27 \times 10^{-3}$ | $5.01 \times 10^{-2}$ |
| Standalone SSM beyond Theorem 1 | 0.194 | 16.62 |
| SSM in non-linear neural network | $1.64 \times 10^{-3}$ | $5.45 \times 10^{-2}$ |

Table 2: Demonstration of clean-label poisoning of SSMs in real-world (non-synthetic) settings comprising SSM-based S4 [24], Mamba-2 [14] and LRU [58] neural networks trained on the (sequential variant of the) CIFAR-10 dataset [38]. Each row in the table summarizes an experiment with a different SSM-based neural network, where cleanly labeled poisonous examples are generated from CIFAR-10 examples by introducing human-imperceptible additive noise to input sequences, while keeping their labels intact. The noise added to an input sequence is rendered to have relatively large last elements—in line with our theory[5]—using an adapted version of the Gradient Matching method from Geiping et al. [18]. The first column in the table specifies the SSM-based neural network being poisoned. The second column reports two cross-entropy losses on the CIFAR-10 test set: one obtained by training the neural network on original CIFAR-10 training examples, and the other obtained by training the neural network on the same examples along with cleanly labeled poisonous examples as described above. The third column is identical to the second, except that it reports classification accuracies rather than cross-entropy losses. Finally, the fourth column in the table reports the relative size of the last elements in the noise introduced for generating a cleanly labeled poisonous example, where the relative size is averaged over all such examples, and quantified by the Euclidean norm of the last 3% of elements divided by the Euclidean norm of all elements. Throughout the table, quantities are averaged over three random seeds. Notice that in all experiments, the addition of cleanly labeled poisonous examples deteriorates generalization. For further details see Appendices I.3 and J.3.

| SSM-based NN | Loss without / with poisoning | Accuracy without / with poisoning | Noise tail size |
|---|---|---|---|
| S4 [24] | 0.739 / 0.839 | 78.8% / 76.9% | 0.406 |
| Mamba-2 [14] | 1.005 / 1.178 | 77.8% / 75.1% | 0.559 |
| LRU [58] | 0.892 / 1.111 | 72.4% / 68.3% | 0.469 |

## 5 Limitations

It is important to acknowledge several limitations of this paper. First, while the dynamical characterization derived in Proposition 1 applies to a broad setting (*e.g.*, it allows the SSM to be embedded in a non-linear neural network), the proof of generalization in Proposition 2 is restricted to a more specific setting (*e.g.*, it requires Assumptions 1 and 2), and the proof of clean-label poisoning in Theorem 1 is restricted to an even more specific setting (*e.g.*, it requires the original training set to include a single example, in addition to Assumptions 1 and 2). Appendix B extends the settings of Proposition 2 and Theorem 1, but even under such extension the settings remain fairly restricted. A second limitation of this paper relates to the guarantees provided by Proposition 2 and Theorem 1: each ensures existence of a set $\mathcal{I}$ of initializations with which a desired result holds, and while $\mathcal{I}$ has positive volume (it is open), this volume may be low. Third, except for the dynamical characterization derived in Proposition 1 and the real-world (non-synthetic) experiments reported in Section 4.2, our theory and experiments pertain to near-zero initialization, which is common for neural networks [20, 30], but does not account for modern SSM initializations designed to alleviate vanishing gradients [23, 25]. Fourth, due to vanishing gradients, maintaining reasonable run-times in experiments with near-zero

initialization necessitated a relatively small scale (in terms of, *e.g.*, SSM dimension and training sequence length). Addressing these limitations is an important set of directions for future work.

## 6    Conclusion

The proliferation of SSMs, particularly in large language models, renders it crucial to understand their implicit bias. In this paper, we revisited prior beliefs by which the implicit bias of SSMs leads to generalization when data is generated by a low dimensional teacher. We formally proved that, in stark contrast to these beliefs, there exist special examples whose addition to training data can completely distort the implicit bias, to a point where generalization with a low dimensional teacher fails. This failure occurs despite the special examples being labeled by the teacher, *i.e.*, having clean labels! We corroborated our theory via experiments spanning a wide range of settings: from those analyzed theoretically, to real-world settings comprising prominent SSM-based neural networks. The experiments confirmed that generalization in SSMs can be disrupted by cleanly labeled training examples, *i.e.*, that SSMs are susceptible to clean-label poisoning.

Our results point to significant challenges in both theory and practice of SSMs. On the theoretical front, our results suggest that generalization in SSMs cannot be explained via the traditional view of implicit complexity minimization [26, 77, 90], or through the nascent view by which generalization is typical [82, 53, 54, 7, 2]. Indeed, if generalization in SSMs was due to the implicit bias finding a solution which, among all solutions fitting training data, minimizes some (data-independent) complexity measure, then training with additional cleanly labeled examples would not change the solution found, and thus would not disrupt generalization.[6] Moreover, if generalization in SSMs was due to typicality, *i.e.*, to the majority of solutions fitting training data being ones that generalize, then additional cleanly labeled training examples would only improve generalization, as they enhance the dominance of such majority. We believe fundamentally new approaches may be needed in order to theoretically pinpoint the source of generalization in SSMs.

Moving to the practical side, the fact that SSMs are susceptible to clean-label poisoning raises significant concerns regarding safety, robustness and reliability. For example, large language models, which are becoming more and more reliant on SSMs [19, 61, 3], are often fine-tuned via supervised learning on public internet data [41, 40, 81], and in this process, it may be easy for a malicious actor to add cleanly labeled training examples, *e.g.*, by adding unlabeled training examples prior to label generation. We believe that delineating the susceptibility of SSMs to clean-label poisoning, and developing methods for overcoming this susceptibility, are critical research directions to pursue.

Our results suggest that *learning dynamics* may facilitate progress in the foregoing research directions. Indeed, in settings covered by our theory, clean-label poisoning correlates with a dynamical factor $\gamma^{(0)}(\cdot)$ (defined in Proposition 1) being large during learning, *i.e.*, during optimization (see Section 3.1.1 as well as Figures 16 to 19 in Appendix I). By monitoring $\gamma^{(0)}(\cdot)$ throughout learning, one may detect clean-label poisoning and take responsive action (*e.g.*, prevent deployment of a poisoned model). Moreover, it is plausible that by adding to the training loss a penalty term that encourages $\gamma^{(0)}(\cdot)$ to be small, one may improve resilience to clean-label poisoning. Investigating such methods, and extending them to settings beyond our theory, may promote safer, more robust and more reliable deployment of SSMs.

**Acknowledgments and Disclosure of Funding**

We thank Eshbal Hezroni for assistance in preparing the illustrative figure, and Itamar Zimerman for contributions to the real-world experiments. This work was supported by the European Research Council (ERC) grant NN4C 101164614, a Google Research Scholar Award, a Google Research Gift, Meta, the Yandex Initiative in Machine Learning, the Israel Science Foundation (ISF) grant 1780/21, the Tel Aviv University Center for AI and Data Science, the Adelis Research Fund for Artificial Intelligence, Len Blavatnik and the Blavatnik Family Foundation, and Amnon and Anat Shashua. NR is supported in part by the Zuckerman STEM Leadership Program.

---

[6]Prior work argued that generalization in different neural networks cannot be explained via implicit complexity minimization [83], but to our knowledge, such results do not apply to SSMs.

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

## A  Related Work

An SSM can be viewed as a special case of a *linear dynamical system* (*LDS*): a classic object of study in areas such as systems theory [57] and control theory [76]. The problem of learning from data an SSM that admits in-distribution and out-of-distribution generalization is an instance of what is known in the LDS literature as *system identification* [74]. Determination of whether a high dimensional SSM realizes a mapping that is also realizable by a low dimensional SSM (in our context, these are a student and a teacher, respectively) is considered in the LDS literature under the topic of *minimal realization theory* [73]. Despite these connections, our work is distinct from classic LDS literature: it studies the implicit bias of gradient descent, a phenomenon brought to light by the recent rise of overparameterized neural networks [55].

A significant line of research has characterized the implicit bias of gradient descent in overparameterized neural networks [69, 39, 56, 77, 27, 46, 34, 88, 9, 90, 5, 52, 47, 13, 89, 87, 51, 37, 86, 42, 63]. Several recent works formally studied the implicit bias of gradient descent in the context of recurrent neural networks [44, 16, 12]: a broad class of models that includes SSMs. Some of these works [16, 12] focus specifically on SSMs, in particular Cohen-Karlik et al. [12] which we extend (by lifting the unrealistic assumption of infinite training data—see Section 1). However, to our knowledge, none of the prior works on the implicit bias of gradient descent over SSMs or recurrent neural networks have formally established susceptibility to clean-label poisoning, as we do.

Since its demonstration in Shafahi et al. [71], clean-label poisoning has received significant empirical attention [93, 92, 33, 1]. It was also studied theoretically for convex models [79, 6].[7] To our knowledge, none of the prior works on clean-label poisoning have formally established the phenomenon for SSMs, whose optimization is non-convex.

## B  Extensions of Theorem 1 and Proposition 2

In this appendix, we extend Theorem 1 to allow for additional training sets. Furthermore, we outline extensions of Theorem 1 and Proposition 2 to settings in which: *(i)* the teacher SSM is of arbitrary dimension $d^* \geq 2$; *(ii)* the input and output matrices of the teacher SSM vary; and *(iii)* the input and output matrices of the student SSM are learned (as opposed to being fixed throughout training).

**Clean-label poisoning with additional training sets.**  Theorem 2 below extends Theorem 1 to allow for any training set $\mathcal{S}$ permitted by Proposition 2 (*i.e.*, for any choice of training sequences $(\mathbf{x}^{(i)})_{i=1}^n$ where: for every $i \in [n]$, $\mathbf{x}^{(i)}$ has zeros in its last two elements and non-negative values elsewhere; and $\mathbf{x}^{(i)} \neq \mathbf{0}$ for some $i \in [n]$), at the expense of the added (cleanly labeled poisonous) sequence $\mathbf{x}^\dagger$ potentially having large norm.[8]

**Theorem 2.** *Consider the teacher-student setting of Section 2.3, subject to Assumptions 1 and 2. Suppose that for every $i \in [n]$ the training sequence $\mathbf{x}^{(i)}$ has zeros in its last two elements and non-negative values elsewhere. Suppose also that $\mathbf{x}^{(i)} \neq \mathbf{0}$ for some $i \in [n]$. Finally, suppose that the dimension of the student SSM satisfies $d \geq 5$. Then, for any $k \in \mathbb{N}_{\geq 3}$, $\delta \in \mathbb{R}_{>0}$ and $\epsilon \in \mathbb{R}_{>0}$, there exists a scalar $c \in \mathbb{R}_{>0}$ and an open set $\mathcal{I}$ of initializations for the student SSM,[4] such that with any initialization in $\mathcal{I}$:*

- *there exists a time $t \in \mathbb{R}_{>0}$ at which gradient flow reaches a point where the training loss is at most $\delta$, and the generalization errors over sequence lengths $1, 2, \ldots, k$ are at most $\epsilon$; and*

- *appending to $\mathcal{S}$ the (cleanly labeled) example $(\mathbf{x}^\dagger, y^\dagger)$, where $\mathbf{x}^\dagger := c \cdot \mathbf{e}_{\kappa-1} \in \mathbb{R}^\kappa$ and $y^\dagger := \phi_{(A^*, B^*, C^*)}(\mathbf{x}^\dagger)$, leads to the existence of a time $t \in \mathbb{R}_{>0}$ at which gradient flow reaches a point where the training loss is at most $\delta$, and the generalization error over sequence length $k$ is at least $0.5 - 2/d$.*

*Proof.* See Appendix G. □

---

[7]Non-convex models were theoretically treated as well [49, 50, 17], but such treatments considered a different type of poisoning, namely, one where poisonous examples come instead of (rather than in addition to) original training examples.

[8]An additional limitation of Theorem 2 compared to Theorem 1 is that its guarantees pertain to some point along the gradient flow trajectory, whereas those of Theorem 1 pertain to the trajectory's limit.

**Teacher of arbitrary dimension.** For any $d^* \geq 2$, consider the following parameter assignments for the teacher SSM:

$$
A^* = \begin{pmatrix} 1 & 0 & \cdots & 0 \\ 0 & 0 & \cdots & 0 \\ \vdots & \vdots & \ddots & \vdots \\ 0 & 0 & \cdots & 0 \end{pmatrix} \in \mathbb{R}^{d^*, d^*} \;\;,\;\; B^* = \begin{pmatrix} 1 \\ \sqrt{\frac{d-1}{d^*-1}} \\ \sqrt{\frac{d-1}{d^*-1}} \\ \vdots \\ \sqrt{\frac{d-1}{d^*-1}} \end{pmatrix} \in \mathbb{R}^{d^*, 1} \;\;,\;\; C^* = \begin{pmatrix} 1 \\ \sqrt{\frac{d-1}{d^*-1}} \\ \sqrt{\frac{d-1}{d^*-1}} \\ \vdots \\ \sqrt{\frac{d-1}{d^*-1}} \end{pmatrix}^{\top} \in \mathbb{R}^{1, d^*} .
$$

In this setting, the mapping $\phi_{(A^*, B^*, C^*)}(\cdot)$ realized by the teacher SSM is the same as the mapping realized by the teacher SSM defined in Equation (10) (where the teacher has dimension $d^* = 2$). Accordingly, Proposition 2 and Theorem 1 and their proofs apply as stated.

**Varying teacher input and output matrices.** Given any teacher SSM $(A^*, B^*, C^*)$ with which Proposition 2 and Theorem 1 hold (including a high dimensional teacher as described above), similar results hold with the teacher SSM $(A^*, \alpha_1 B^*, \alpha_2 C^*)$, where $\alpha_1, \alpha_2 \in \mathbb{R}_{\neq 0}$ are arbitrary. Indeed, if we likewise scale the values of the (fixed) student parameters $B$ and $C$, *i.e.* we replace $B$ by $\alpha_1 B$ and $C$ by $\alpha_2 C$, then for every sequence $\mathbf{x}$:

$$
\phi_{(A^*, \alpha_1 B^*, \alpha_2 C^*)}(\mathbf{x}) = \alpha_1 \alpha_2 \phi_{(A^*, B^*, C^*)}(\mathbf{x})
$$

and likewise:

$$
\phi_{(A, \alpha_1 B, \alpha_2 C)}(\mathbf{x}) = \alpha_1 \alpha_2 \phi_{(A, B, C)}(\mathbf{x}) .
$$

The training loss and its derivatives thus scale by a positive factor, and so do generalization errors (Definition 1). Accordingly, the proofs of Proposition 2 and Theorem 1 carry through.

**Learned student input and output matrices.** Below we outline a modification of Proposition 2 and Theorem 1 that accounts for a setting in which the input and output matrices of the student SSM are learned. Suppose these input and output matrices—$B$ and $C$, respectively—are learned with a learning rate (step size) that may be different from the learning rate of the student's state transition matrix $A$. Formally, suppose the optimization trajectory $(A(\cdot), B(\cdot), C(\cdot))$ is governed by the following dynamics:

$$
\begin{aligned}
\dot{A}(t) &= -\frac{\partial}{\partial A} \ell_{\mathcal{S}}(A(t), B(t), C(t)) \\
\dot{B}(t) &= -\eta \cdot \frac{\partial}{\partial B} \ell_{\mathcal{S}}(A(t), B(t), C(t)) \;\;,\;\; t \in \mathbb{R}_{\geq 0} , \\
\dot{C}(t) &= -\eta \cdot \frac{\partial}{\partial C} \ell_{\mathcal{S}}(A(t), B(t), C(t))
\end{aligned}
\tag{11}
$$

where $\eta > 0$ represents the ratio between the learning rate of $B$ and $C$, and the learning rate of $A$. Consider a trajectory induced by Equation (11), and a corresponding trajectory that emanates from the same initialization, but where only $A$ is learned (or equivalently, where $\eta$ in Equation (11) is replaced by zero). Arguments similar to those used in the proofs of Proposition 2 and Theorem 1 can be used to show that the divergence between these two trajectories is upper bounded by a quantity that depends on $\eta$, and in particular tends to zero as $\eta$ does. Accordingly, if $\eta$ is sufficiently small, generalization errors attained when $A$, $B$ and $C$ are learned jointly (*i.e.*, when optimization is governed by Equation (11)), are close to those attained when only $A$ is learned. Proposition 2 and Theorem 1—which apply to a setting where only $A$ is learned—thus translate to results that apply to a setting where $B$ and $C$ are also learned.

## C  Low Training Loss with High Generalization Error

As stated in Section 2.3, it was shown in Cohen-Karlik et al. [12] that, regardless of the size of the training set $\mathcal{S}$ (*i.e.*, of $n$), and the length $\kappa$ input sequences it comprises (namely, $(\mathbf{x}^{(i)} \in \mathbb{R}^{\kappa})_{i=1}^{n}$), there exist assignments for the student SSM $(A, B, C)$ which minimize the training loss $\ell_{\mathcal{S}}(\cdot)$, yet suffer from arbitrarily high generalization error over sequence lengths beyond $\kappa$, *e.g.* over sequence length $\kappa + 1$. For completeness, we prove this fact below.

**Proposition 3.** *For any $\epsilon \in \mathbb{R}_{>0}$ there exist assignments for $(A, B, C)$ with which the generalization error is zero over sequence length $\kappa$,[9] yet it is at least $\epsilon$ over sequence length $\kappa + 1$.*

*Proof.* Let $d > \kappa$, which we will take as the dimension of our student SSM, and let

$$(A, B, C) = \big(\text{Diag}(a_1, \ldots, a_d), (b_1, \ldots, b_d), (c_1, \ldots, c_d)^\top\big).$$

We we will consider the system of equations

$$(CA^i B)_{0 \leq i \leq d-1} = \mathbf{r},$$

where $\mathbf{r} \in \mathbb{R}^d$. First, observe that our claim follows if a solution $(A, B, C)$ exists for for every $\mathbf{r}$. Indeed, take $\mathbf{r}$ such that its first $\kappa$ entries coincide with $(C^*(A^*)^i B^*)_{0 \leq i \leq \kappa-1}$, and each of its final $d - \kappa$ entries differs from the corresponding entry of $(C^*(A^*)^i B^*)_{\kappa \leq i \leq d-1}$ by $\epsilon$ or more. Then $(A, B, C)$ solving these equations would provide the required assignment. To see that this system is indeed solvable for every $\mathbf{r} \in \mathbb{R}^d$, note that it can be rewritten as $V^\top \mathbf{g} = \mathbf{r}$, where

$$V = \begin{pmatrix} 1 & a_1 & a_1^2 & \cdots & a_1^{d-1} \\ 1 & a_2 & a_2^2 & \cdots & a_2^{d-1} \\ \vdots & \vdots & \vdots & \ddots & \vdots \\ 1 & a_d & a_d^2 & \cdots & a_d^{d-1} \end{pmatrix},$$

and $\mathbf{g} = (b_1 c_1, \ldots, b_d c_d)^\top$. $V$ is a Vandermonde matrix, and it is well known that it is invertible as long as $a_1, \ldots, a_d$ are all distinct. Therefore for any such $\mathbf{r}$, and fixed, distinct $a_1, \ldots, a_d$, one can solve the equation by setting $\mathbf{g} = (V^\top)^{-1} \mathbf{r}$. To obtain $(B, C)$ which satisfy $\mathbf{g} = (V^\top)^{-1} \mathbf{r}$, one can simply set $B = \mathbf{1}$ and $C = \mathbf{g}^\top$.

$\square$

# D    Proof of Proposition 1 (Dynamical Characterization)

Fix $t \geq 0$. We use the following shorthands for simplicity:

$$\widetilde{\phi}(\mathbf{x}^{(i)}) := \phi_{(A(t), B(t), C(t)), \mathbf{w}(t)}(\mathbf{x}^{(i)}) \quad , \quad \phi(\mathbf{x}^{(i)}) := \phi_{(A(t), B(t), C(t))}(\mathbf{x}^{(i)}).$$

The objective $\tilde{\ell}_S$ in time $t$ takes the following form:

$$\tilde{\ell}_S(A(t), B(t), C(t)) = \frac{1}{n} \sum_{i=1}^{n} (y^{(i)} - \widetilde{\phi}(\mathbf{x}^{(i)}))^2.$$

Fix $j \in [d]$. Deriving w.r.t $a_j(t)$ and consecutively applying the chain rule we obtain the following

$$
\begin{aligned}
\frac{\partial}{\partial a_j(t)} \tilde{\ell}_S(A(t), B(t), C(t)) &= \frac{1}{n} \sum_{i=1}^{n} \frac{\partial}{\partial \widetilde{\phi}(\mathbf{x}^{(i)})} (y^{(i)} - \widetilde{\phi}(\mathbf{x}^{(i)}))^2 \cdot \frac{\partial}{\partial a_j(t)} \widetilde{\phi}(\mathbf{x}^{(i)}) \\
&= \frac{2}{n} \sum_{i=1}^{n} \underbrace{(\widetilde{\phi}(\mathbf{x}^{(i)}) - y^{(i)})}_{=-\delta^{(i)}(t)} \cdot \frac{\partial}{\partial a_j(t)} \sigma\big(\phi(\mathbf{x}^{(i)}), \mathbf{w}(t)\big) \\
&= -\frac{2}{n} \sum_{i=1}^{n} \delta^{(i)}(t) \underbrace{\frac{\partial}{\partial z} \sigma(z, \mathbf{w}(t))|_{z = \phi(\mathbf{x}^{(i)})}}_{=\xi^{(i)}(t)} \cdot \frac{\partial}{\partial a_j(t)} \phi(\mathbf{x}^{(i)}) \\
&= -\frac{2}{n} \sum_{i=1}^{n} \delta^{(i)}(t) \xi^{(i)}(t) \cdot \frac{\partial}{\partial a_j(t)} \left( \sum_{l=1}^{\kappa} C(t) A(t)^{\kappa-l} B(t) x_l^{(i)} \right) = (*).
\end{aligned}
$$

---

[9] Note that this implies zero training loss, *i.e.* $\ell_S(A, B, C) = 0$ for any training set $S$ of sequence length $\kappa$, regardless of its size.

Recalling that $A$ is diagonal, we have that $C(t)A(t)^{\kappa-l}B(t)x_l^{(i)} = \sum_{j'=1}^d c_{j'}(t)a_{j'}(t)^{\kappa-l}b_{j'}(t)x_l^{(i)}$. Hence,

$$(*) = -\frac{2}{n}\sum_{i=1}^n \delta^{(i)}(t)\xi^{(i)}(t) \cdot \frac{\partial}{\partial a_j(t)}\left(\sum_{l=1}^\kappa \sum_{j'=1}^d c_{j'}(t)a_{j'}(t)^{\kappa-l}b_{j'}(t)x_l^{(i)}\right)$$

$$= -\frac{2}{n}\sum_{i=1}^n \delta^{(i)}(t)\xi^{(i)}(t)\left(\sum_{l=1}^\kappa (\kappa-l)c_j(t)a_j(t)^{\kappa-l-1}b_j(t)x_l^{(i)}\right) = (**).$$

Reversing the order of summation and reordering we receive the following:

$$(**) = -b_j(t)c_j(t)\sum_{l=0}^{\kappa-2} a_j(t)^l \cdot \underbrace{\frac{2(l+1)}{n}\left(\sum_{i=1}^n \delta^{(i)}(t)\xi^{(i)}(t)x_{\kappa-l-1}^{(i)}\right)}_{=\gamma^{(l)}(t)}.$$

The proof concludes by noting that $\dot{a}_j(t) = -\frac{\partial}{\partial a_j(t)}\tilde{\ell}_S((A(t), B(t), C(t)))$. $\qquad\square$

# E  Proof of Proposition 2 (Generalization)

Proposition 2 follows from Proposition 4, which is identical up to allowing the non-zero diagonal element of $A^*$ to be any positive value instead of 1.

**Proposition 4.** *Consider the teacher-student setting of Section 2.3, subject to Assumption 2 and the teacher SSM given by*

$$A^* = \begin{pmatrix} a^* & 0 \\ 0 & 0 \end{pmatrix}, \quad B^* = \begin{pmatrix} 1 & \sqrt{d-1} \end{pmatrix}^\top, \quad C^* = \begin{pmatrix} 1 & \sqrt{d-1} \end{pmatrix}, \tag{12}$$

*for some $a^* > 0$. Suppose that for every $i \in [n]$ the training sequence $\mathbf{x}^{(i)}$ has zeros its last two elements and non-negative values elsewhere. Suppose also that $\mathbf{x}^{(i)} \neq \mathbf{0}$ for some $i \in [n]$. Then, for any $k \in \mathbb{N}$, $\epsilon \in \mathbb{R}_{>0}$ and $\delta \in \mathbb{R}_{>0}$, there exists a time $t \in \mathbb{R}_{>0}$ and an open set $\mathcal{I}$ of initializations for the student SSM, such that gradient flow initialized in $\mathcal{I}$ reaches at time $t$ a point at which the training loss $\ell_S(\cdot)$ is no greater than $\delta$, and the generalization errors over sequence lengths $1, 2, \ldots, k$ are no greater than $\epsilon$.*

*Proof.* For $k \in \mathbb{N}$ and $A \in \mathbb{R}^{d,d}$, we denote by $Gen_k(A)$ the generalization error over sequence length $k$ (Definition 1). Note that $B$ and $C$ are omitted from this notation, as they are fixed to the values $B = \mathbf{1}, C = \mathbf{1}^\top$ throughout the proof. With slight abuse of notation, we also denote $\ell_S(A) := \ell_S(A, B, C)$. Consider the point $A_0 = (a_0, 0, \ldots, 0)$ where $0 < a_0 < a^*$. We will first show that if we initialize at $A_0$, gradient flow will converge to $(a^*, 0, \ldots, 0)$, and therefore achieve perfect generalization. Indeed, writing 3 in terms of the entries of $A$ we get:

$$\ell_S(A(\tau)) = \frac{1}{n}\sum_{i=1}^n \left(\sum_{l=2}^{\kappa-1}(a^*)^l x_{\kappa-l}^{(i)} - \sum_{l=2}^{\kappa-1}\left(\sum_{j=1}^d a_j(\tau)^l\right)x_{\kappa-l}^{(i)}\right)^2.$$

The derivative of $\ell_S(A)$ with respect to $a_p$ is therefore:

$$\frac{\partial}{\partial a_p}\ell_S(A) = -\frac{2}{n}\sum_{i=1}^n \left(\sum_{l=2}^{\kappa-1}(a^*)^l x_{\kappa-l}^{(i)} - \sum_{l=2}^{\kappa-1}\left(\sum_{j=1}^d a_j^l\right)x_{\kappa-l}^{(i)}\right)\left(\sum_{l=2}^{\kappa-1}l\, a_p^{l-1}x_{\kappa-l}^{(i)}\right).$$

For $j > 2$, $a_j(0) = 0$ and thus $\dot{a}_j(0) = -\frac{\partial}{\partial a_j}\ell_S(A(0)) = 0$. Therefore for all $j > 2$, $a_j(t) = 0$ for all $\tau > 0$. Hence it suffices to show that $a_1(\tau)$ converges to $a^*$ as $\tau \to \infty$. To see this, note that because $a_j(\tau) = 0$ for all $t > 0$ the dynamics simplify to

$$\dot{a}_1(\tau) = -\frac{\partial}{\partial a_1}\ell_S(A(\tau)) = \frac{2}{n}\sum_{i=1}^n \left(\sum_{l=2}^{\kappa-1} x_{\kappa-l}^{(i)}((a^*)^l - a_1(\tau)^l)\right)\left(\sum_{l=2}^{\kappa-1}l\, a_1(t)^{l-1}x_{\kappa-l}^{(i)}\right).$$

For all $i \in [n]$, at $\tau = 0$ it holds that

$$\left( \sum_{l=2}^{\kappa-1} x_{\kappa-l}^{(i)} ((a^*)^l - a_1(t)^l) \right) \left( \sum_{l=2}^{\kappa-1} l \, a_1(t)^{l-1} x_{\kappa-l}^{(i)} \right) \geq 0$$

by the non negativity of the entries of $\mathbf{x}^{(i)}$. Additionally, by the positivity of at least one of the entries of $\mathbf{x}^{(i)}$ for some $i \in [n]$, at least one of the above expressions is positive. Furthermore, this expression can equal zero if and only if $a_1 = a^*$ (in which case $\dot{a}_1(\tau) = 0$). Therefore $\dot{a}_1(\tau) > 0$ for all $\tau > 0$, and hence $a_1(\tau)$ is monotonically increasing. It is also bounded from above (by $a^*$), thus it converges. Furthermore, the limit must be a point where the derivative vanishes, and therefore it must equal $a^*$.

Because $A(\tau)$ converges to $(a^*, 0, \ldots, 0)$ when initialized at $A_0$, for any $\epsilon, \delta > 0$ there exists $t > 0$ such that $\ell_{\mathcal{S}}(A(t)) < \frac{\epsilon}{2}$, $Gen_k(A(t)) < \frac{\delta}{2}$. Now by the continuity of $\ell_{\mathcal{S}}(\cdot)$, $Gen_k(\cdot)$ and by Lemma 29, there exists an open set $\mathcal{I}$ such that, if we initialize gradient flow from $\widetilde{A}(0) \in \mathcal{I}$, resulting in the trajectory $\widetilde{A}(\tau)$, we get that $\|A(t) - \widetilde{A}(t)\|_2$ is sufficiently small to ensure that

$$\ell_{\mathcal{S}}(\widetilde{A}(t)) < \delta, \quad Gen_k(\widetilde{A}(t)) < \epsilon,$$

as required. $\qquad\square$

## F  Proof of Theorem 1 (Clean-Label Poisoning)

The outline of the proof is as follows. Appendix F.1 details the setting and additional notation. Appendix F.2 analyzes gradient flow over $\ell_{\mathcal{S}}$, where the dataset $\mathcal{S}$ does not include "poisoned" samples, and shows that it converges to a generalizing solution. Appendix F.3 analyzes gradient flow after the addition of "poisoned" samples, establishing that generalization degrades. Appendix F.4 proves that the different initialization sets considered in Appendix F.2 and Appendix F.3 intersect, and that one can construct an open set $\mathcal{I}$ such that both phenomena occur.

### F.1  Setting and Additional Notation

We will slightly change our notation and use $L$ to denote the sequence length, and $k$ as an index. For any $\mathbf{x} \in \mathbb{R}^d$ and any $r \geq 0$ we use $B_r(\mathbf{x})$ to denote

$$B_r(\mathbf{x}) := \{\mathbf{z} \in \mathbb{R}^d : \|\mathbf{x} - \mathbf{z}\|_2 < r\} \tag{13}$$

and $\overline{B_r}(\mathbf{x})$ to denote

$$\overline{B_r}(\mathbf{x}) := \{\mathbf{z} \in \mathbb{R}^d : \|\mathbf{x} - \mathbf{z}\|_2 \leq r\}. \tag{14}$$

For any $\mathbf{x} \in \mathbb{R}^d$ and any $\mathcal{V} \subseteq \mathbb{R}^d$ we define the Euclidean distance between $\mathbf{x}$ and $\mathcal{V}$ as

$$\mathrm{Dist}(\mathbf{x}, \mathcal{V}) := \inf_{\mathbf{z} \in \mathcal{V}} \|\mathbf{x} - \mathbf{z}\|_2. \tag{15}$$

We use $\mathcal{W}_1$ and $\mathcal{W}_2$ to respectively denote

$$\mathcal{W}_1 := \mathrm{Span}\{\mathbf{1}\} \quad, \quad \mathcal{W}_2 := \mathrm{Span}\{\mathbf{e}_1 - \mathbf{e}_2, \ldots, \mathbf{e}_1 - \mathbf{e}_d\}. \tag{16}$$

Note that for any $j \in \{2, \ldots, d\}$ it holds that

$$\mathbf{1}^\top (\mathbf{e}_1 - \mathbf{e}_j) = 1 - 1 = 0.$$

Hence $\mathcal{W}_1$ and $\mathcal{W}_2$ are orthogonal. Additionally, it holds that

$$\mathcal{W}_1 \cap \mathcal{W}_2 = \{\mathbf{0}\} \quad, \quad \dim \mathcal{W}_1 = 1 \quad, \quad \dim \mathcal{W}_2 = d - 1,$$

hence $\mathcal{W}_1 \cup \mathcal{W}_2 = \mathbb{R}^d$. Finally, for any $\psi \geq 0$ we use $\mathrm{Diff}(\psi)$ to denote

$$\mathrm{Diff}(\psi) := \left\{ \mathbf{x} \in \mathbb{R}^d : \; \forall i, j \in [d], |x_i - x_j| \leq \psi \right\} \tag{17}$$

and $\mathrm{Diff}(\psi)^{\mathcal{C}}$ to denote

$$\mathrm{Diff}(\psi)^{\mathcal{C}} := \left\{ \mathbf{x} \in \mathbb{R}^d : \; \exists i, j \in [d] \; s.t. \; |x_i - x_j| > \psi \right\}. \tag{18}$$

Recall that the teacher SSM (Equation (10)) is given by $(A^*, B^*, C^*)$, where

$$A^* = \begin{pmatrix} 1 & 0 \\ 0 & 0 \end{pmatrix} \;,\quad B^* = \begin{pmatrix} 1 & \sqrt{d-1} \end{pmatrix}^\top \;,\quad C^* = \begin{pmatrix} 1 & \sqrt{d-1} \end{pmatrix}.$$

We claim that the teacher is equivalent, *i.e.* has the same impulse response, as a $d$-dimensional SSM with $A^d = \mathrm{Diag}(1, 0, \ldots, 0)$, $B^d = \mathbf{1}$ and $C^d = \mathbf{1}^\top$.

**Proposition 5.** *For all $i \geq 0$*

$$C^*(A^*)^i B^* = C^d (A^d)^i B^d.$$

*Proof.* It is easy to see that both expressions evaluate to $d$ when $i = 0$, and to 1 when $i \geq 1$. □

We will henceforth abuse notation slightly and redefine the teacher $(A^*, B^*, C^*)$ to equal this $d$ dimensional teacher, *i.e.* we set $A^* := A^d, B^* := B^d, C^* := C^d$.

We denote the generalization error on sequences of length $L$ (Definition 1) by $Gen_L(A)$, *i.e.*

$$Gen_L(A) := \max_{L' \in \{0,1,\ldots,L-1\}} \left| BA^{L'}C - B^*(A^*)^{L'}C^* \right|.$$

For a training set $\mathcal{S} = \left( (\mathbf{x}^{(i)}, y^{(i)}) \right)_{i=1}^n$, we make a slight abuse of notation and denote the training error of a weight matrix $A \in \mathbb{R}^{d,d}$ to be

$$\ell_\mathcal{S}(A) := \ell_\mathcal{S}(A, B, C) = \frac{1}{n} \sum_{i=1}^n \left( y^{(i)} - \phi_{(A,B,C)}(\mathbf{x}^{(i)}) \right)^2.$$

Note that $B$ and $C$ are kept implicit in these notations, as they are fixed to the values $B = \mathbf{1}, C = \mathbf{1}^\top$ throughout our analysis.

Examining the teacher weights $(A^*, B^*, C^*)$, one can note that for any $j \in [L-1]$ and any $z \in \mathbb{R}$ it holds that

$$\phi_{(A^*,B^*,C^*)}(z \cdot \mathbf{e}_j) = \sum_{k=1}^d c_k^*(a_k^*)^{L-j} b_k^* z = 1 \cdot 1^{L-j} \cdot 1 \cdot z + 0 \cdot z = z.$$

To facilitate clearer distinction, we denote $\mathcal{S}_1$ the training set considered in the first case, and $\mathcal{S}_2$ the training set considered in the second case. We provide below an explicit description of the training sets and their induced losses.

**Definition 2.** The training sets $\mathcal{S}_1, \mathcal{S}_2$ are defined as follows:

$$\mathcal{S}_1 := \{(\mathbf{x}, y)\} = \left\{ \left( \mathbf{e}_1, \phi_{A^*,B^*,C^*}(\mathbf{e}_1) \right) \right\},$$

$$\mathcal{S}_2 := \{(\mathbf{x}, y), (\mathbf{x}^\dagger, y^\dagger)\} = \left\{ \left( \mathbf{e}_1, \phi_{A^*,B^*,C^*}(\mathbf{e}_1) \right), \left( \mathbf{e}_{L-1}, \phi_{A^*,B^*,C^*}(\mathbf{e}_{L-1}) \right) \right\}.$$

The objective $\ell_{\mathcal{S}_1}(\cdot)$ takes the following form:

$$\ell_{\mathcal{S}_1}(A) = \left( \phi_{(A^*,B^*,C^*)}(\mathbf{e}_1) - \phi_{(A,B,C)}(\mathbf{e}_1) \right)^2 = \left( 1 - \sum_{k=1}^d a_k^{L-1} \right)^2. \tag{19}$$

For any time $t \geq 0$ and any index $j \in [d]$ the gradient flow update $\dot{a}_j(t; \mathcal{S}_1)$ takes the following form

$$\dot{a}_j(t; \mathcal{S}_1) = -\frac{\partial}{\partial a_j(t; \mathcal{S}_1)} \ell_{\mathcal{S}_1}\left( A(t; \mathcal{S}_1) \right) = 2(L-1) \left( 1 - \sum_{k=1}^d a_k(t; \mathcal{S}_1)^{L-1} \right) a_j(t; \mathcal{S}_1)^{L-2}. \tag{20}$$

The objective $\ell_{\mathcal{S}_2}(\cdot)$ takes the following form:

$$\ell_{\mathcal{S}_2}(A) = \frac{1}{2} \left( \left( \phi_{A^*,B^*,C^*}(\mathbf{e}_1) - \phi_{(A,B,C)}(\mathbf{e}_1) \right)^2 + \left( \phi_{(A^*,B^*,C^*)}(\mathbf{e}_{L-1}) - \phi_{(A,B,C)}(\mathbf{e}_{L-1}) \right)^2 \right)$$

$$= \frac{1}{2} \left( \left( 1 - \sum_{k=1}^d a_k^{L-1} \right)^2 + \left( 1 - \sum_{k=1}^d a_k \right)^2 \right). \tag{21}$$

For any time $t \geq 0$ and any index $j \in [d]$ the gradient flow update $\dot{a}_j(t; \mathcal{S}_2)$ takes the following form

$$
\begin{aligned}
\dot{a}_j(t; \mathcal{S}_2) &= -\frac{\partial}{\partial a_j(t; \mathcal{S}_2)} \ell_{\mathcal{S}_2}\big(A(t; \mathcal{S}_2)\big) \\
&= (L-1)\bigg(1 - \sum_{k=1}^d a_k(t; \mathcal{S}_2)^{L-1}\bigg) a_j(t; \mathcal{S}_2)^{L-2} + 1 - \sum_{k=1}^d a_k(t; \mathcal{S}_2) \, .
\end{aligned}
\tag{22}
$$

Note that by Lemma 23 the above flows are defined for all $t \geq 0$. We denote by $\mathcal{I}_0$ a set of initial values for the matrix $A$ which we will use throughout the proof:[10]

$$
\mathcal{I}_0 := \bigg\{ \alpha \cdot (\zeta_1, \dots, \zeta_d)^\top \in \mathbb{R}^d : \alpha \in (0, \tfrac{1}{2d}), 1 = \zeta_1 > \zeta_2 > \cdots > \zeta_d > 0 \bigg\} \, .
\tag{23}
$$

Throughout Appendix F.2 and Appendix F.3 we will be concerned with subsets of $\mathcal{I}_0$ for which the respective claims hold.

## F.2 Gradient Flow Under $\mathcal{S}_1$ Generalizes

Throughout this part, we omit the dependence on $\mathcal{S}_1$ for simplicity. We begin by proving that when initializing at some $A(0) \in \mathcal{I}_0$, the parameters of $A$ converge to a point where the training loss equals zero.

**Lemma 1.** *Suppose we initialize at $A(0) \in \mathcal{I}_0$ and evolve $A(t)$ according to the gradient flow dynamics in Equation (19). Then the limit $\lim_{t \to \infty} A(t) =: \widehat{A}_1$ exists and satisfies*

$$
\ell(\widehat{A}_1) = 0 \, .
$$

*Proof.* We first prove that for any $j \in [d]$ and for any time $t \geq 0$ it holds that

$$
\alpha \zeta_j \leq a_j(t) \leq 1 \, .
$$

Recall that

$$
\dot{a}_j(t) = 2(L-1)\bigg(1 - \sum_{k=1}^d a_k(t)^{L-1}\bigg) a_j(t)^{L-2} \, .
$$

Hence, by Equation (23) it must hold that $\dot{a}_j(t) \geq 0$ for any $t \geq 0$; Since $\alpha \zeta_j > 0$ and since $L - 1$ is even we have

$$
1 - \sum_{k=1}^d (\alpha \zeta_k)^{L-1} \geq 1 - d \cdot (\alpha \zeta_1)^{L-1} \geq 1 - d(\tfrac{1}{2d})^{L-1} > 0 \, .
$$

Hence at time $t = 0$ we have $\dot{a}_j(0) > 0$. For any $t > 0$, if the derivative equals zero then either $1 - \sum_{k=1}^d a_k(t)^{L-1} = 0$ or $a_j(t) = 0$, implying the derivative must remain equal to zero for $t' > t$. Hence, $\alpha \zeta_j \leq a_j(t)$ for any $t \geq 0$. Additionally, for any time $t \geq 0$ it holds that $1 - \sum_{k=1}^d a_k(t)^{L-1} \geq 0$. At initialization it is positive by the above, and again if at some point it is equals zero then it must remain zero thereafter. Therefore, $a_j(t)$ can never reach 1: since $L - 1$ is even and since all entries are strictly positive, if it were to reach or cross 1 we would reach a contradiction to the previous argument. Thus, we have showed that the gradient flow trajectory is contained in the following open and bounded set:

$$
\mathcal{V} = B_d(\mathbf{0}) \setminus \overline{B_{\frac{\alpha \zeta_d}{2}}(\mathbf{0})} \, .
$$

Note that the teacher $A^*$ is within $\mathcal{V}$. Next, we claim that within $\mathcal{V}$ the objective $\ell$ satisfies the PL condition (see Definition 13) with PL coefficient $2(L-1)^2(\frac{\alpha \zeta_d}{2\sqrt{d}})^{2L-4}$. Indeed, for any $A \in \mathcal{V}$ and any $j \in [d]$ it holds that

$$
\frac{\partial}{\partial a_j} \ell(A) = 2(L-1)\bigg(1 - \sum_{k=1}^d a_k^{L-1}\bigg) a_j^{L-2} \, .
$$

---

[10] $A$ is a diagonal matrix, so we treat $\mathcal{I}_0$ as a subset of $\mathbb{R}^d$.

For any $A \in \mathcal{V}$ there must exist an index $j^* \in [d]$ for which $|a_{j^*}| \geq \frac{\alpha \zeta_d}{2\sqrt{d}}$ and thus

$$
\begin{aligned}
\|\nabla \ell(A)\|_2^2 &\geq \left( 2(L-1)\left(1 - \sum_{k=1}^{d} a_k^{L-1}\right) a_{j^*}^{L-2} \right)^2 \\
&= 4(L-1)^2 a_{j^*}^{2L-4} \ell(A) \\
&\geq 2 \cdot 2(L-1)^2 \left( \frac{\alpha \zeta_d}{2\sqrt{d}} \right)^{2L-4} \ell(A).
\end{aligned}
$$

Finally, there exists some constant $M > 0$ such that within $\mathcal{V}$ the objective $\ell$ has $M$-Lipschitz gradients, since $\ell$ is analytic in $\mathbb{R}^d$ and since $\mathcal{V}$ is contained within the compact and bounded $\overline{B_d}(\mathbf{0})$. The above conditions allows us to invoke Lemma 25 which states that the limit $\lim_{t\to\infty} A(t) =: \widehat{A_1}$ exists and satisfies $\ell(A(t)) = 0$ as required. $\qquad \square$

We now introduce a set $\mathcal{I}_1 \subseteq \mathcal{I}_0$, under which we prove the rest of the claims in this section.

**Definition 3.** Let $\eta_1 > 0$. We use $\mathcal{I}_1(\eta_1)$ to denote the following subset of $\mathcal{I}_0$:

$$
\mathcal{I}_1(\eta_1) := \left\{ A \in \mathcal{I}_0 : \forall j \in \{2, \ldots, d\}. \ \alpha \leq \left( \frac{1 - (1-\eta_1)^{L-1} - \eta_1}{d-1} \right)^{\frac{1}{L-1}} \frac{1}{\zeta_j} (1 - \zeta_j^{L-3})^{\frac{1}{L-3}} \right\}.
$$

We now prove that if $A(0) \in \mathcal{I}_1$, the first diagonal entry tends to 1, while the rest of the entries must remain close to 0.

**Proposition 6.** Let $\eta_1 > 0$. Suppose we initialize at $A(0) \in \mathcal{I}_1(\eta_1)$ and evolve $A(t)$ according to the gradient flow dynamics in Equation (19). For any $j \in \{2, \ldots, d\}$ and for any time $t \geq 0$ it holds that:

$$
0 \leq a_j(t) \leq \left( \frac{1 - (1-\eta_1)^{L-1} - \eta_1}{d-1} \right)^{\frac{1}{L-1}}.
$$

Additionally, there exists some time $t^* \geq 0$ such that for any time $t \geq t^*$ it holds that:

$$
1 - \eta_1 \leq a_1(t) \leq 1.
$$

*Proof.* Per the proof of Lemma 1, $\dot{a}_j(t) \geq 0$ for any $j \in [d]$ and $t \geq 0$ and thus the entries $a_j(t)$ are positive and non-decreasing (as functions of $t$). Reordering the dynamics, we have the following for any $j \in \{2, \ldots, d\}$ and for any time $\tau \geq 0$:

$$
\dot{a}_j(\tau) a_j(\tau)^{-L+2} = \frac{\dot{a}_j(\tau)}{a_j(\tau)^{L-2}} = 2(L-1)\left(1 - \sum_{k=1}^{d} a_k(\tau)^{L-1}\right) = \frac{\dot{a}_1(\tau)}{a_1(\tau)^{L-2}} = \dot{a}_1(\tau) a_1(\tau)^{-L+2}.
$$

Integrating both sides w.r.t time, we receive the following for any time $t \geq 0$:

$$
\begin{aligned}
\frac{a_j(t)^{-L+3}}{-L+3} - \frac{a_j(0)^{-L+3}}{-L+3} &= \int_0^t \dot{a}_j(\tau) a_j(\tau)^{-L+2} d\tau \\
&= \int_0^t \dot{a}_1(\tau) a_1(\tau)^{-L+2} d\tau \\
&= \frac{a_1(t)^{-L+3}}{-L+3} - \frac{a_1(0)^{-L+3}}{-L+3}.
\end{aligned}
$$

Organizing the equation and plugging the initial values, we get that

$$
a_j(t)^{-L+3} = a_1(t)^{-L+3} + (\alpha \zeta_j)^{-L+3} - \alpha^{-L+3}.
$$

Both sides are positive by our first argument and since $\alpha\zeta_j < \alpha$, and so taking the $\frac{1}{L-3}$ root yields

$$a_j(t) = \left( \frac{1}{a_1(t)^{-L+3} + (\alpha\zeta_j)^{-L+3} - \alpha^{-L+3}} \right)^{\frac{1}{L-3}}$$

$$\leq \left( \frac{1}{\frac{1}{(\alpha\zeta_j)^{L-3}} - \frac{1}{\alpha^{L-3}}} \right)^{\frac{1}{L-3}}$$

$$= \left( \frac{(\alpha\zeta_j)^{L-3}}{1 - \zeta_j^{L-3}} \right)^{\frac{1}{L-3}}$$

$$= \alpha\zeta_j \left( \frac{1}{1 - \zeta_j^{L-3}} \right)^{\frac{1}{L-3}}.$$

Since $A(0) \in \mathcal{I}_1(\eta_1)$, we obtain that

$$a_j(t) \leq \left( \frac{1 - (1-\eta_1)^{L-1} - \eta_1}{d-1} \right)^{\frac{1}{L-1}} \frac{1}{\zeta_j} (1 - \zeta_j^{L-3})^{\frac{1}{L-3}} \zeta_j \left( \frac{1}{1 - \zeta_j^{L-3}} \right)^{\frac{1}{L-3}}$$

$$= \left( \frac{1 - (1-\eta_1)^{L-1} - \eta_1}{d-1} \right)^{\frac{1}{L-1}}$$

as desired. We now show that there exists $t^* \geq 0$ such that for any time $t \geq t^*$ it holds that

$$a_1(t) \geq 1 - \eta_1.$$

By Lemma 1, there exists time $t^* \geq 0$ such that for any $t \geq t^*$ it holds that

$$\ell(A(t)) = \left( 1 - \sum_{k=1}^{d} a_k(t)^{L-1} \right)^2 \leq \eta_1^2.$$

Therefore, for any time $t \geq t^*$ we have

$$\left| 1 - \sum_{k=1}^{d} a_k(t)^{L-1} \right| \leq \eta_1 \implies 1 - \eta_1 \leq \sum_{k=1}^{d} a_k(t)^{L-1} \leq 1 + \eta_1.$$

Focusing on the left hand side and plugging the bound on the rest of the entries, we receive

$$1 - \eta_1 \leq a_1(t)^{L-1} + (d-1) \cdot \frac{1 - (1-\eta_1)^{L-1} - \eta_1}{d-1} = a_1(t)^{L-1} + 1 - (1-\eta_1)^{L-1} - \eta_1.$$

Rearranging yields

$$(1 - \eta_1)^{L-1} \leq a_1(t)^{L-1} \implies 1 - \eta_1 \leq a_1(t).$$

Additionally, $a_1(t)$ can never cross 1 - since $L - 1$ is even and since all entries are strictly positive, if it were to cross 1 we would reach a contradiction to the argument in Lemma 1 stating that the residual $1 - \sum_{k=1}^{d} a_k(t)^{L-1}$ is always non-negative. With this we complete our proof. $\qquad\square$

An immediate result from Proposition 6 is the following corollary regarding the student's recovery of the teacher.

**Corollary 1.** *Let $\eta_1 > 0$. Suppose we initialize at $A(0) \in \mathcal{I}_1(\eta_1)$ and evolve $A(t)$ according to the gradient flow dynamics in Equation (19). The limit $\lim_{t\to\infty} A(t) =: \widehat{A_1}$ satisfies*

$$\|\widehat{A_1} - A^*\|_2 \leq \sqrt{\eta_1^2 + (d-1)\left( \frac{1 - (1-\eta_1)^{L-1} - \eta_1}{d-1} \right)^{\frac{2}{L-1}}}.$$

*Proof.* By Proposition 6, there exists time $t^* \geq 0$ such that for any time $t \geq t^*$ it holds that
$$\|A(t) - A^*\|_2$$

$$= \sqrt{(1 - a_1(t))^2 + \sum_{k=2}^{d} \left( 0 - a_k(t) \right)^2} \leq \sqrt{\eta_1^2 + (d-1)\left( \frac{1 - (1-\eta_1)^{L-1} - \eta_1}{d-1} \right)^{\frac{2}{L-1}}}.$$

The argument follows from Lemma 1 and from continuity. $\qquad\square$

**Remark 1.** *Note that the upper bound in corollary 1 satisfies the following*

$$\lim_{\eta_1 \to 0} \sqrt{\eta_1^2 + (d-1)\left(\frac{1 - (1 - \eta_1)^{L-1} - \eta_1}{d - 1}\right)^{\frac{2}{L-1}}}$$

$$= \sqrt{\lim_{\eta_1 \to 0} \eta_1^2 + (d-1)\left(\frac{1 - (1 - \eta_1)^{L-1} - \eta_1}{d - 1}\right)^{\frac{2}{L-1}}}$$

$$= 0 \,.$$

*Hence, for any recovery threshold $\delta > 0$ there exists $\eta_{1,\delta} > 0$ such that if $A(0) \in \mathcal{I}_1(\eta_{1,\delta})$ then $\widehat{A_1}$ recovers $A^*$ with an error of no more than $\delta$.*

So far, we have argued that the parameters of A converge to a point which is close $A^*$. We conclude by showing that this leads to low generalization error.

**Proposition 7.** *Let $L' \geq L + 2$. For any $\epsilon > 0$ there exists an open set of initializations $\mathcal{I}_1 := \mathcal{I}_1(\delta_\epsilon)$ such that under $\mathcal{S}_1$, A converges to a point such that $Gen_{L'}(A) \leq \epsilon$.*

*Proof.* Under the dataset $\mathcal{S}_1$, we have shown above that for any $\delta > 0$ there exists an open set of initializations $\mathcal{I}_1(\delta)$ such that GF will converge to a solution a whose parameters satisfy $\|A - A^*\|^2 \leq \delta$ . It follows from the continuity of the length $L'$ impulse response that there is an open set of initializations from which we converge to a point $Gen_{L'}(A) \leq \epsilon$. $\qquad\square$

We abuse notation slightly and denote $\mathcal{I}_1(\epsilon) := \mathcal{I}_1(\eta_{1,\delta_1})$ where $\delta_1$ is the maximal $\delta$ that guarantees $Gen_{L'}(\widehat{A_1}) \leq \epsilon$.

### F.3 Gradient Flow Over $\mathcal{S}_2$ Converges but Does Not Generalize

In this section, we show that one can find a set of initializations $\mathcal{I}_2$ such that gradient flow under $\mathcal{S}_2$ converges to a point with high generalization error. The proof shows that gradient flow trajectories initialized in $\mathcal{I}_2$ evolve similarly to reference trajectories which provably stays away from any permutation of $A^*$.[11] Since the training loss is non-convex, gradient flow trajectories can diverge from one another exponentially fast. Establishing that a reference trajectory is tracked thus requires sharp bounds on convergence times. The proof in this section is rather involved and is thus split into several parts.

- Appendix F.3.1 defines the reference trajectories and shows their poor ability of generalization.

- Appendix F.3.2 characterizes the critical points of the objective $\ell$, focusing on a specific saddle point of interest (which we denote $\mathbf{s}$).

- Appendix F.3.3 presents relevant background on dynamical systems, introducing a linearization result needed for the rest of the proof.

- In Appendix F.3.4 we start analyzing the trajectories themselves, showing that they must pass near $\mathbf{s}$.

- Appendix F.3.5 shows that the trajectories must escape sufficiently fast from $\mathbf{s}$ using the tools presented in F.3.3.

- Appendix F.3.6 proves that after escaping from $\mathbf{s}$ the trajectories converge to global minima.

- Appendix F.3.7 shows that the overall divergence between trajectories emanating from $\mathcal{I}_2$ and their corresponding reference trajectories can be bounded from above, implying the former trajectories have poor generalization.

Throughout this part, we omit the dependence on $\mathcal{S}_2$ for simplicity.

---

[11] Any permutation of $A^*$ yields a system with the same impulse response.

### F.3.1 Reference Trajectories

We begin by proving the following useful lemma which states that gradient flow maintains the order of the entries of $A$.

**Lemma 2.** *Suppose we initialize at $A(0) \in \mathbb{R}^d$ and evolve $A(t)$ according to Equation (22). Let $\pi : [d] \to [d]$ be a permutation such that for any $j \in [d-1]$:*

$$a_{\pi(j)}(0) \geq a_{\pi(j+1)}(0).$$

*Then for any $j \in [d-1]$ and any $t \geq 0$ it holds that*

$$a_{\pi(j)}(t) \geq a_{\pi(j+1)}(t).$$

*Proof.* Recall the dynamics from Equation (22):

$$\dot{a}_j(t) = (L-1)\left(1 - \sum_{k=1}^d a_k(t)^{L-1}\right)a_j(t)^{L-2} + 1 - \sum_{k=1}^d a_k(t).$$

Fix $j \in [d-1]$. By the linearity of the derivative, we obtain the following equality by plugging the above dynamics

$$\frac{d}{dt}\left(a_{\pi(j)}(t) - a_{\pi(j+1)}(t)\right) = \dot{a}_{\pi(j)}(t) - \dot{a}_{\pi(j+1)}(t)$$

$$= 2(L-1)\left(1 - \sum_{k=1}^d a_k(t)^{L-1}\right)\left(a_{\pi(j)}(t)^{L-2} - a_{\pi(j+1)}(t)^{L-2}\right).$$

Assume on the contrary there exists some time $t_1 \geq 0$ for which $a_{\pi(j)}(t_1) < a_{\pi(j+1)}(t_1)$. By the assumption, $t_1 > 0$. By continuity, there must exist some time $t_2 \in [0, t_1)$ for which $a_{\pi(j)}(t_2) = a_{\pi(j+1)}(t_2)$. This would imply that for any $t \geq t_2$, the derivative $\frac{d}{dt}\left(a_{\pi(j)}(t) - a_{\pi(j+1)}(t)\right)$ is equal zero, which in turn would imply that

$$a_{\pi(j)}(t) - a_{\pi(j+1)}(t) = a_{\pi(j)}(t_2) - a_{\pi(j+1)}(t_2) = 0$$

in contradiction to the assumption on $t_1$. $\qquad\square$

In what follows, we define the notion of *reference initialization*.

**Definition 4.** Let $A \in \mathcal{I}_0$ be some initialization of the parameters. The corresponding *reference initialization* $A^{ref}$ is defined as

$$\forall j \in [d]. \quad a_j^{ref} = \begin{cases} a_1, & j = 1, 2 \\ a_j, & \text{otherwise} \end{cases}.$$

We use $A^{ref}(t)$ to denote the gradient flow trajectories emanating from the reference initializations.

We now prove that any point with zero training loss which is sufficiently close to a reference trajecory has poor generalization.

**Lemma 3.** *Let $L' \geq L+2$. There exists some $\delta_2 > 0$ such that any point $A = (a_1, a_2, \ldots, a_d) \in \mathbb{R}^d$ which satisfies:*

- $\ell(A) = 0$

- $a_1 \geq a_2 \geq \cdots \geq a_d$

- $\|A - A^{eq}\| \leq \delta_2$ *for some point* $A^{eq} = (a_1^{eq}, \ldots, a_d^{eq}) \in \mathbb{R}^d$ *such that* $a_1^{eq} = a_2^{eq}$

*must satisfy* $Gen_{L'}(A) \geq \min\left\{0.1, \frac{1}{9d} \cdot (1 - (0.6)^{\frac{1}{L-1}})\right\}$.

*Proof.* Let $L^* \in \{L+1, \ldots, L'\}$ such that $L^*$ is even. We now show that

$$\sum_{k=1}^{d} a_k^{L^*-1} \leq 1 - c$$

for some constant $c > 0$ which is independent of $L'$. This in turn implies that

$$Gen_{L'}(A) \geq (1 - CA^{L^*-1}B) \geq c$$

which gives us the desired lower bound. To do this, we write

$$\sum_{k}^{d} a_k^{L^*-1} = \sum_{k}^{d} a_k^{L-1} a_k^{L^*-L}.$$

First note that $|a_k| \leq 1$ for all $k \in [d]$ - this follows from the fact that $L-1$ is even and from the fact that $\ell(A) = 0$ and hence $\sum_k a_k^{L-1} = 1$. Therefore, for all $k \in [d]$ we have

$$|a_k^{L^*-1}| = |a_k^{L-1} a_k^{L^*-L}| = |a_k^{L-1}| \cdot |a_k^{L^*-L}| \leq a_k^{L-1}.$$

Assume first that $a_1 = a_2 = a$. Then clearly $a_1^{L-1} + a_2^{L-1} = 2a^{L-1} \leq 1$ and hence

$$a_1^{L-1}, a_2^{L-1} \leq \frac{1}{2} \implies a_1, a_2 \leq \left(\frac{1}{2}\right)^{\frac{1}{L-1}}.$$

Now by continuity it follows that for sufficiently small $\delta_2 > 0$, we have that if $\|A - A^{eq}\| \leq \delta_2$ then

$$a_1, a_2 \leq (0.6)^{\frac{1}{L-1}}.$$

Let $J := \{r : a_r \leq 0\}$. For such indices we have $a_r^{L^*-1} \leq 0$. Suppose that

$$\sum_{k \in J} a_k^{L-1} \geq 0.1.$$

Then we have that

$$\sum_{k=1}^{d} a_k^{L^*-1} \leq \sum_{k \notin J} a_k^{L^*-1} = \sum_{k \notin J} a_k^{L-1} a_k^{L^*-L} \leq \sum_{k \notin J} a_k^{L-1} = 1 - \sum_{k \notin J} a_k^{L-1} \leq 0.9$$

so we can take $c = 0.1$. Otherwise we have that

$$\sum_{k \notin J} a_k^{L-1} \geq 0.9$$

so there exists some $k^* \notin J$ such that $a_{k^*}^{L-1} \geq \frac{1}{9d}$. On the other hand, we have

$$a_{k^*} \leq a_1 \leq |a_1| \leq (0.6)^{\frac{1}{L-1}}.$$

Therefore, since $k^* \notin J$ we have $0 \leq a_{k^*}^{L^*-L} \leq a_{k^*}$ and so

$$a_{k^*}^{L-1} - a_{k^*}^{L^*-1} = a_{k^*}^{L-1}(1 - a_{k^*}^{L^*-L}) \geq \frac{1}{9d}(1 - a_{k^*}) \geq \frac{1}{9d}\left(1 - (0.6)^{\frac{1}{L-1}}\right).$$

This yields the following:

$$1 - \sum_{k=1}^{d} a_k^{L^*-1} = \sum_{k=1}^{d} a_k^{L-1} - \sum_{k=1}^{d} a_k^{L^*-1} = \sum_{k=1}^{d}(a_k^{L-1} - a_k^{L^*-1}) \geq \frac{1}{9d}\left(1 - (0.6)^{\frac{1}{L-1}}\right)$$

which gives us $c = \frac{1}{9d}\left(1 - (0.6)^{\frac{1}{L-1}}\right)$. In either case we can find a constant $c > 0$ proving the argument. $\square$

Lemma 3 motivates us to find an open subset of initializations under which the respective gradient flow trajectories remain close to their reference trajectory counterparts, as this would allow us to lower bound generalization error.

### F.3.2 Characterization of Critical Points

In this section we characterize the critical points of the objective $\ell$.

**Lemma 4.** *Let $A \in \mathbb{R}^d$ be a point such that*

$$\nabla \ell(A) = 0.$$

*Then either $A$ is a global minimum, i.e. $\ell(A) = 0$, or exists $s \in \mathbb{R}$ such that $A = s \cdot \mathbf{1}$.*

*Proof.* By Equation (22), for any $j \in [d]$ it holds that

$$\frac{\partial}{\partial a_j} \ell(A) = (L-1)\left(\sum_{k=1}^{d} a_k^{L-1} - 1\right) a_j^{L-2} + \sum_{k=1}^{d} a_k - 1 = 0.$$

If $\sum_{k=1}^{d} a_k^{L-1} - 1 = 0$, then the above implies that $\sum_{k=1}^{d} a_k - 1 = 0$. This in turn yields that

$$\ell(A) = \frac{1}{2}\left(\left(1 - \sum_{k=1}^{d} a_k^{L-1}\right)^2 + \left(1 - \sum_{k=1}^{d} a_k\right)^2\right) = 0,$$

*i.e.* $A$ is a global minimum. Suppose $\sum_{k=1}^{d} a_k^{L-1} - 1 \neq 0$. Then we obtain by rearranging that

$$a_j^{L-2} = \frac{\left(1 - \sum_{k=1}^{d} a_k\right)}{(L-1)\left(\sum_{k=1}^{d} a_k^{L-1} - 1\right)}.$$

$L-2$ is odd, and so taking the $L-2$ root on both sides we obtain that

$$a_j = \left(\frac{\left(1 - \sum_{k=1}^{d} a_k\right)}{(L-1)\left(\sum_{k=1}^{d} a_k^{L-1} - 1\right)}\right)^{\frac{1}{L-2}} := s$$

completing our proof. $\qquad\square$

Lemma 4 establishes that critical points of $\ell$ which are not global minima must reside within $\mathcal{W}_1 = \mathrm{Span}\{\mathbf{1}\}$ (Equation (16)). These saddle points pose an obstacle to the convergence of gradient flow to a global minimum. The following lemma outlines the type of points gradient flow could encounter assuming we initialize at $\mathcal{I}_0$ or at a reference initialization.

**Lemma 5.** *Suppose we initialize at $A(0) \in \mathcal{I}_0$ and at $A^{ref}(0)$, and evolve $A(t)$ and $A^{ref}(t)$ according to Equation (22). Then for any time $t \geq 0$ it holds that*

$$\ell\big(A(t)\big), \ell\big(A^{ref}(t)\big) \leq 1.$$

*Proof.* Per Equation (23) the entries at initialization are arranged in descending order. Since $L-1$ is even we have that the initializations satisfy the inequalities

$$1 - \sum_{k=1}^{d} a_k(0)^{L-1} = 1 - \sum_{k=1}^{d} (\alpha \zeta_k)^{L-1} \geq 1 - 2(\alpha \zeta_1)^{L-1} - \sum_{k=3}^{d} (\alpha \zeta_k)^{L-1} = 1 - \sum_{k=1}^{d} a_k^{ref}(0)^{L-1}$$

and

$$1 - \sum_{k=1}^{d} a_k(0) = 1 - \sum_{k=1}^{d} (\alpha \zeta_k) \geq 1 - 2(\alpha \zeta_1) - \sum_{k=3}^{d} (\alpha \zeta_k) = 1 - \sum_{k=1}^{d} a_k^{ref}(0).$$

By Equation (23) it holds that $\alpha \zeta_1 < \frac{1}{2d}$, thus we have that

$$1 - 2(\alpha \zeta_1)^{L-1} - \sum_{k=3}^{d} (\alpha \zeta_k)^{L-1} \geq 1 - d \cdot (\alpha \zeta_1)^{L-1} \geq 1 - d(\frac{1}{2d})^{L-1} > 0$$

and

$$1 - 2(\alpha\zeta_1) - \sum_{k=3}^{d}(\alpha\zeta_k) \geq 1 - d\cdot(\alpha\zeta_1) \geq 1 - d(\frac{1}{2d}) > 0\,.$$

On the other hand, by Equation (23) it also holds that $\alpha\zeta_d > 0$, thus we have that

$$1 - \sum_{k=1}^{d}(\alpha\zeta_k)^{L-1} \leq 1 - d\cdot(\alpha\zeta_d)^{L-1} < 1$$

and

$$1 - \sum_{k=1}^{d}(\alpha\zeta_k) \leq 1 - d\cdot(\alpha\zeta_d) < 1\,.$$

Therefore, the quantities

$$1 - \sum_{k=1}^{d} a_k(0)^{L-1} \quad,\quad 1 - \sum_{k=1}^{d} a_k^{ref}(0)^{L-1} \quad,\quad 1 - \sum_{k=1}^{d} a_k(0) \quad,\quad 1 - \sum_{k=1}^{d} a_k^{ref}(0)$$

are all within the interval $(0,1)$. Thus, the objective at both initializations is no more than 1 since both satisfy

$$\ell(A) = \frac{1}{2}\big((1 - d\cdot a^{L-1})^2 + (1 - d\cdot a)^2\big) \leq \frac{1}{2}(1^2 + 1^2) \leq 1\,.$$

The proof is completed by the argument in Lemma 22 which states that under gradient flow the objective is non-increasing. $\qquad\square$

The following lemma shows that only a specific region of $\mathcal{W}_1$ can contain critical points with loss lower than that of the initialization points we consider. This, along with Lemma 22, implies that only a specific region of $\mathcal{W}_1$ is relevant.

**Lemma 6.** *Let $A \in \mathbb{R}^d$ be a point for which there exists $a \in \mathbb{R}$ such that $A = a\cdot\mathbf{1}$. If $a \notin [\frac{1}{d}, \frac{3}{d}]$ then either $\nabla\ell(A) \neq 0$ or $\ell(A) > 1$*

*Proof.* We begin by proving that for any $a \in \mathbb{R}$, if $a \notin (0, \frac{3}{d}]$ then $a\cdot\mathbf{1}$ must incur a loss greater than 1. If $a > \frac{3}{d}$ then it holds that $d\cdot a > 3$, hence we obtain that

$$\ell(a\cdot\mathbf{1}) = \frac{1}{2}\big((1 - d\cdot a^{L-1})^2 + (1 - d\cdot a)^2\big) \geq \frac{(1 - d\cdot a)^2}{2} > 1\,.$$

The same argument applies when $a < -\frac{1}{d}$, since in that case $d\cdot a < -1 \implies (1 - d\cdot a)^2 > 2$. Next, we show that if $a \in [-\frac{1}{d}, \frac{1}{d})$ then $\nabla\ell(a\cdot\mathbf{1}) \neq 0$. Suppose $a \in [-\frac{1}{d}, 0]$. $L-1$ is even and $d \geq 8$ hence

$$d\cdot a^{L-1} - 1 \in [-1,0) \implies (L-1)(d\cdot a^{L-1} - 1)a^{L-2} \in \left(0, \frac{L-1}{d^{L-2}}\right) \subseteq \left(0, \frac{L-1}{8^{L-2}}\right)\,.$$

The function $f(L) := \frac{L-1}{8^{L-2}}$ is decreasing for $L \geq 3$ and acheives the value $0.25$ when $L = 3$, hence since $L \geq 3$ we get $f(L) \leq 0.25$. Thus we have for any $j \in [d]$ that the gradient's $j$th entry statisfies

$$\nabla\ell(a\cdot\mathbf{1}) = \big((L-1)(d\cdot a^{L-1})a^{L-2} + (d\cdot a - 1)\big) \leq \big(0.25 + (d\cdot a - 1)\big) \leq (0.25 - 1) < 0\,.$$

Suppose $a \in (0, \frac{1}{d})$. In this case, we have that

$$a^{L-1} < \frac{1}{d} \implies d\cdot a^{L-1} < 1 \implies (L-1)(d\cdot a^{L-1} - 1)a^{L-2} < 0\,.$$

Hence, since $d\cdot a - 1 < 0$ we have for any $j \in [d]$ that the gradient's $j$th entry satisfies

$$\nabla\ell(a\cdot\mathbf{1})_j = \big((L-1)(d\cdot a^{L-1} - 1)a^{L-2} + (d\cdot a - 1)\big) < 0\,.$$

Therefore, any critical point which is not a global minimum and has value in $(0,1)$ cannot reside outside of $[\frac{1}{d}, \frac{3}{d}]$. $\qquad\square$

Having disqualified most of $\mathcal{W}_1$, we now identify the unique critical point on the non-disqualified region of $\mathcal{W}_1$ and show that it is not a global minimum.

**Lemma 7.** *There exists a unique $s \in [\frac{1}{d}, \frac{3}{d}]$ for which $\nabla \ell(s \cdot \mathbf{1}) = 0$. Additionally, $s$ satisfies*

$$\mathbf{s} := s \cdot \mathbf{1} = \underset{A \in \mathcal{W}_1}{\arg\min} \, \ell(A)$$

*and*

$$\ell(\mathbf{s}) \geq \frac{1}{8} > 0 \,.$$

*Proof.* We focus on the following function:

$$f(a) = \frac{1}{2}\left((1 - d \cdot a^{L-1})^2 + (1 - d \cdot a)^2\right) \,.$$

Note that $f(a) = \ell(a \cdot \mathbf{1})$. It holds that

$$f'(a) := (L-1)(d \cdot a^{L-1} - 1)a^{L-2} + (d \cdot a - 1) \,.$$

Note that $f'(a) = \nabla \ell(a \cdot \mathbf{1})_j$ for any $j \in [d]$, and so $f'(a) = 0$ if and only if $\nabla \ell(a \cdot \mathbf{1}) = \mathbf{0}$. We proceed to show that within $[-\frac{1}{d}, \frac{3}{d}]$, $f'(a)$ has a root and is monotonic. It holds that

$$f'(0) = d\left((L-1)(d \cdot (0)^{L-1} - 1)(0)^{L-2} + (d \cdot 0 - 1)\right) = -d < 0 \,.$$

Next, since $d \geq 8$ it holds that

$$(L-1)(1 - d \cdot (\frac{3}{d})^{L-1})(\frac{3}{d})^{L-2} \leq (L-1)(\frac{3}{d})^{L-2} \leq \frac{L-1}{2^{L-2}}(\frac{3}{4})^{L-2} =: h(L) \,.$$

$h(L)$ is a decreasing function for $L \geq 3$ and achieves the value $0.75$ when $L = 3$, hence since $L \geq 3$ we get $h(L) \leq 1$. Therefore,

$$\begin{aligned} f'(\frac{3}{d}) &= d\left((L-1)(d \cdot (\frac{3}{d})^{L-1} - 1)(\frac{3}{d})^{L-2} + (d \cdot \frac{3}{d} - 1)\right) \\ &= d\left(2 - (L-1)(1 - d \cdot (\frac{3}{d})^{L-1})(\frac{3}{d})^{L-2}\right) \\ &\geq d(2-1) \\ &> 0 \,. \end{aligned}$$

Hence by continuity, $f'(a)$ has a root within $[-\frac{1}{d}, \frac{3}{d}]$. Note that by Lemma 4, $f'(a)$ doesn't have a root within $[-\frac{1}{d}, \frac{1}{d})$, implying the root is actually achieved in $[\frac{1}{d}, \frac{3}{d}]$. Next, it holds that

$$\begin{aligned} f''(a) &= d\left((L-1)(2L-3)d \cdot a^{2L-4} - (L-1)(L-2)a^{L-3} + d\right) \\ &\geq d\left(d - (L-1)(L-2)a^{L-3}\right) \,. \end{aligned}$$

Because $d \geq 8$ and $L - 3$ is even, we have for any $a \in [-\frac{1}{d}, \frac{3}{d}]$

$$(L-1)(L-2)a^{L-3} \leq (L-1)(L-2)(\frac{3}{d})^{L-3} \leq \frac{(L-1)(L-2)}{2^{L-3}}(\frac{3}{4})^{L-3} =: g(L) \,.$$

$g(L)$ is a decreasing function for $L \geq 4$ and achieves the value $2.25$ when $L = 4$, hence since $L \geq 4$ we get $g(L) \leq 2.25$. Therefore

$$f''(a) \geq d(d - 2.25) > 0 \,,$$

implying $f'$ is monotonically increasing in $[-\frac{1}{d}, \frac{3}{d}]$. Hence, there exists a unique $s \in [\frac{1}{d}, \frac{3}{d}]$ such that $f'(s) = 0$, which implies that $\nabla \ell(s \cdot \mathbf{1}) = 0$. Note that we showed that $s$ is a minimizer of $f$ over $[-\frac{1}{d}, \frac{3}{d}]$, as $f$'s derivative is zero at $s$ and the second derivative is positive along the interval. Finally, let $a \in \mathbb{R} \setminus [-\frac{1}{d}, \frac{3}{d}]$. By Lemma 4, it holds that

$$f(a) = \ell(a \cdot \mathbf{1}) \geq 1 \,.$$

On the other hand, it also holds that

$$f(s) < f(\frac{1}{d}) = \frac{1}{2}\big((1 - d \cdot (\frac{1}{d})^{L-1})^2 + (1 - d \cdot \frac{1}{d})^2\big) \leq \frac{1}{2}.$$

Thus, $s$ is a minimizer of $f$ over $\mathbb{R}$, meaning that $\mathbf{s} := s \cdot \mathbf{1}$ is a minimizer of $\ell$ over $\mathcal{W}_1$ as required. On the other hand, since $d \geq 8$ and $L \geq 4$ it holds that

$$1 - d \cdot s^{L-1} \geq 1 - d \cdot (\frac{3}{d})^{L-1} = 1 - 3 \cdot (\frac{3}{d})^{L-2} \geq 1 - 3 \cdot (\frac{3}{8})^2 \geq \frac{1}{2}.$$

Therefore,

$$\ell(\mathbf{s}) = \frac{1}{2}\big((1 - d \cdot s^{L-1})^2 + (1 - d \cdot s)^2\big) \geq \frac{1}{2}(1 - d \cdot s^{L-1})^2 \geq \frac{1}{8} > 0$$

completing the proof. $\qquad\qquad\square$

In the last two lemmas of this section, we explicitly compute an eigendecomposition of $\ell$'s hessian in $\mathbf{s}$ and bound its eigenvalues.

**Lemma 8.** *Consider* $\mathbf{s}$ *defined in Lemma 7. An eigendecomposition of the symmetric hessian matrix* $\nabla^2\ell(\mathbf{s})$ *is the following:*

- *The eigenvector* $\mathbf{1}$ *with the eigenvalue*

$$\lambda_+ := (L - 1)\big((2L - 3)d \cdot s^{L-1} - (L - 2)\big)s^{L-3} + d.$$

- *For* $j \in \{2, \ldots, d\}$ *the eigenvector* $\mathbf{e}_1 - \mathbf{e}_j$ *with the eigenvalue*

$$\lambda_- := (L - 1)(L - 2)(d \cdot s^{L-1} - 1)s^{L-3}.$$

*Proof.* We begin by computing the hessian matrix $\nabla^2\ell(A)$ for a general $A \in \mathbb{R}^d$, which is symmetric since $\ell(A)$ is analytic. by Equation (22), for any $j \in [d]$ it holds that

$$\frac{\partial}{\partial a_j}\ell(A) = (L - 1)\bigg(\sum_{k=1}^{d} a_k^{L-1} - 1\bigg)a_j^{L-2} + \sum_{k=1}^{d} a_k - 1.$$

Therefore, for any $j \in [d]$ we have that

$$\bigg(\nabla^2\ell(A)\bigg)_{jj} = (L - 1)\bigg((2L - 3)a_j^{2L-4} + (L - 2)\sum_{k=1,k\neq j}^{d} a_k^{L-1}a_j^{L-3} - (L - 2)a_j^{L-3}\bigg) + 1.$$

Additionally, for any $j, i \in [d]$ such that $j \neq i$ we have that

$$\bigg(\nabla^2\ell(A)\bigg)_{ij} = (L - 1)^2 a_i^{L-2}a_j^{L-2} + 1.$$

Now we specialize to $A = \mathbf{s}$. For $j \in [d]$, we obtain

$$\bigg(\nabla^2\ell(\mathbf{s})\bigg)_{jj} = (L - 1)\big((2L - 3)s^{2L-4} + (L - 2)(d - 1)s^{2L-4} - (L - 2)s^{L-3}\big) + 1$$

$$= (L - 1)\big((2L - 3 + L \cdot d - 2d - L + 2)s^{2L-4} - (L - 2)s^{L-3}\big) + 1$$

$$= (L - 1)\big((L - 1 + L \cdot d - 2d)s^{2L-4} - (L - 2)s^{L-3}\big) + 1 =: \omega_1.$$

For $j, i \in [d]$ such that $j \neq i$ we obtain that

$$\bigg(\nabla^2\ell(\mathbf{s})\bigg)_{ij} = (L - 1)^2 s^{2L-4} + 1 =: \omega_2.$$

Observe that

$$\nabla^2\ell(\mathbf{s}) = (\omega_1 - \omega_2)I_d + \omega_2\mathbf{1}_{d,d}$$

where $\mathbf{1}_{d,d}$ is the $d, d$ all ones matrix. Hence, by Lemma 26 we obtain that an eigendecomposition for $\nabla^2\ell(\mathbf{s})$ is the following:

- The eigenvector $\mathbf{1}$ with the eigenvalue $\lambda_+ := \omega_1 + (d-1)\omega_2$.

- For $j \in \{2,\ldots,d\}$ the eigenvector $\mathbf{e}_1 - \mathbf{e}_j$ with the eigenvalue $\lambda_- := \omega_1 - \omega_2$.

$\lambda_+$ takes the following form:

$$
\begin{aligned}
\lambda_+ &= (L-1)\big((L-1+L\cdot d-2d)s^{2L-4} - (L-2)s^{L-3}\big) + 1 + (d-1)\big((L-1)^2 s^{2L-4} + 1\big) \\
&= (L-1)\big((L-1+L\cdot d-2d+Ld-d-L+1)s^{2L-4} - (L-2)s^{L-3}\big) + d \\
&= (L-1)\big((2L\cdot d-3d)s^{2L-4} - (L-2)s^{L-3}\big) + d \\
&= (L-1)\big((2L-3)d\cdot s^{L-1} - (L-2)\big)s^{L-3} + d\,.
\end{aligned}
$$

$\lambda_-$ takes the following form:

$$
\begin{aligned}
\lambda_- &= (L-1)\big((L-1+L\cdot d-2d)s^{2L-4} - (L-2)s^{L-3}\big) + 1 - \big((L-1)^2 s^{2L-4} + 1\big) \\
&= (L-1)\big((L-1+L\cdot d-2d-L+1)s^{2L-4} - (L-2)s^{L-3}\big) \\
&= (L-1)\big((L\cdot d-2d)s^{2L-4} - (L-2)s^{L-3}\big) \\
&= (L-1)\big((L-2)d\cdot s^{L-1} - (L-2)\big)s^{L-3} \\
&= (L-1)(L-2)\big(d\cdot s^{L-1} - 1\big)s^{L-3}\,.
\end{aligned}
$$

$\square$

We now turn to bounding $\lambda_+$ and $\lambda_-$.

**Lemma 9.** *The eigenvalues $\lambda_+$ and $\lambda_-$ from Lemma 8 statisfy*

$$
\lambda_+ \geq d - 1 > 0
$$

*and*

$$
\lambda_- \in (-1, 0)\,.
$$

*Proof.* Since $s \in [\frac{1}{d}, \frac{3}{d}]$ and since $d \geq 8$ we obtain

$$
\begin{aligned}
(L-1)\big((2L-3)d\cdot s^{L-1} - (L-2)\big)s^{L-3} &\geq -(L-1)(L-2)s^{L-3} \\
&\geq -(L-1)(L-2)(\frac{3}{d})^{L-3} \\
&\geq -\frac{(L-1)(L-2)}{2^{L-3}}(\frac{3}{4})^{L-3} =: f(L)\,.
\end{aligned}
$$

$f(L)$ is increasing for $L \geq 7$ and achieves a value that is $> -0.6$ for $L = 7$. Hence since $L \geq 7$ we get $f(L) \geq -0.6 > -1$. Therefore,

$$
\lambda_+ = (L-1)\big((2L-3)d\cdot s^{L-1} - (L-2)\big)s^{L-3} + d \geq d - 1 > 0\,.
$$

Next, since $s \in [\frac{1}{d}, \frac{3}{d}]$ and since $d \geq 8$ we obtain

$$
(L-1)(L-2)s^{L-3} \leq (L-1)(L-2)(\frac{3}{d})^{L-3} \leq \frac{(L-1)(L-2)}{2^{L-3}}(\frac{3}{4})^{L-3} =: g(L)\,.
$$

$g(L)$ is decreasing and positive for $L \geq 7$ and achieves a value that is $< 0.6$ for $L = 7$. Hence since $L \geq 7$ we get $0 \leq g(L) \leq 0.6 < 1$. Additionally, note that

$$
-1 \leq d\cdot s^{L-1} - 1 \leq 3\cdot(\frac{3}{d})^{L-2} - 1 \leq 3(\frac{3}{8})^{L-2} < 0\,.
$$

Therefore, we obtain that

$$
\lambda_- = (L-1)(L-2)s^{L-3}(d\cdot s^{L-1} - 1) \in (-1, 0)
$$

which completes our proof. $\square$

In the first half of F.3.2 we characterized the critical point $\mathbf{s}$ and established that it is the only critical point that is relevant in our case, since it is not a global minimum and since we cannot exclude the possibility that gradient flow would converge to it. In what follows, we give a closed form solution to the dynamics obtained under the linear approximation around $\mathbf{s}$ to our true dynamics. We will show that under these linearized dynamics, any gradient flow trajectory not initialized in $\mathcal{W}_1$ will escape $\mathbf{s}$ at an exponential rate.

**Lemma 10.** *The linear approximation around $\mathbf{s}$ of the gradient flow dynamics (see Equation (22)) is defined by*

$$\dot{A}^{lin}(t) := -\nabla \ell(\mathbf{s}) - \nabla^2 \ell(\mathbf{s})\big(A^{lin}(t) - \mathbf{s}\big) = -\nabla^2 \ell(\mathbf{s})\big(A^{lin}(t) - \mathbf{s}\big).$$

*The solution to the above linear differential equations system is given by*

$$A^{lin}(t) = Q \exp\Big( -t \cdot \mathrm{Diag}(\lambda_+, \lambda_-, \ldots, \lambda_-)\Big)Q^\top\big(A^{lin}(0) - \mathbf{s}\big) + \mathbf{s},$$

*where $\lambda_+$ and $\lambda_-$ are the eigenvalues $\nabla^2 \ell(\mathbf{s})$ found in Lemma 8, and $Q$ is an orthogonal matrix whose first column is $\frac{1}{\sqrt{d}}\mathbf{1}$ and the rest of its columns are an orthonormal basis of $\mathcal{W}_2$ (defined in Equation (16)).*

*Proof.* First note that the first order Taylor's expansion around $\mathbf{s}$ of $-\nabla\ell(A)$ is given by

$$-\nabla \ell(\mathbf{s}) - \nabla^2 \ell(\mathbf{s})\big(A(t) - \mathbf{s}\big).$$

Since $\mathbf{s}$ is a critical point of $\ell$ (*i.e.*, $\nabla \ell(\mathbf{s}) = \mathbf{0}$), we obtain the following linear approximation

$$\dot{A}^{lin}(t) = -\nabla^2 \ell(\mathbf{s})\big(A(t) - \mathbf{s}\big).$$

Per Lemma 8, an eigendecomposition of $\nabla^2 \ell(\mathbf{s})$ is given by the eigenvector $\mathbf{1}$ with the eigenvalue $\lambda_+$, and the eigenvectors $\{\mathbf{e}_1 - \mathbf{e}_2, \ldots, \mathbf{e}_1 - \mathbf{e}_d\}$ with the eigenvalue $\lambda_-$. Therefore, we may write $\nabla^2 \ell(\mathbf{s})$ as the orthogonal eigendecomposition

$$\nabla^2 \ell(\mathbf{s}) = Q \, \mathrm{Diag}(\lambda_+, \lambda_-, \ldots, \lambda_-)Q^\top$$

where the first column of $Q$ is $\frac{1}{\sqrt{d}}\mathbf{1}$, and the rest of its columns are an orthonormal basis of $\mathrm{Span}\{\mathbf{e}_1 - \mathbf{e}_2, \ldots, \mathbf{e}_1 - \mathbf{e}_d\} = \mathcal{W}_2$. The proof is completed by invoking Lemma 27 which yields the following solution to the linear system:

$$A^{lin}(t) = Q \exp\Big( -t \cdot \mathrm{Diag}(\lambda_+, \lambda_-, \ldots, \lambda_-)\Big)Q^\top\big(A^{lin}(0) - \mathbf{s}\big) + \mathbf{s}.$$

$\square$

The following corollary computes the solution of the linear approximation as a function of the initialization's projections onto $\mathcal{W}_1$ and $\mathcal{W}_2$ (Equation (16)):

**Corollary 2.** *Denote the projection of $A^{lin}(0)$ to $\mathcal{W}_1$ by $\beta_1 \mathbf{1}$ where $\beta_1 \in \mathbb{R}$, and the projection of $A^{lin}(0)$ to $\mathcal{W}_2$ by $\beta_2 \cdot \mathbf{v} \in \mathcal{W}_2$ where $\mathbf{v} \in \mathcal{W}_2$ is a unit vector $\beta_2 \in \mathbb{R}$. Then for any $t \geq 0$ it holds that*

$$A^{lin}(t) = \Big( \exp(-t \cdot \lambda_+)(\beta_1 - s) + s \Big)\mathbf{1} + \Big( \exp(-t \cdot \lambda_-) \cdot \beta_2 \Big)\mathbf{v}.$$

*Proof.* Plugging the projections of $A^{lin}(0)$ to $\mathcal{W}_1$ and $\mathcal{W}_2$, we can write the following:

$$A^{lin}(0) - \mathbf{s} = (\beta_1 - s)\mathbf{1} + \beta_2\mathbf{v}.$$

Hence per Lemma 27 at time $t \geq 0$ the solution $A^{lin}(t)$ takes the following form:

$$A^{lin}(t) = Q \exp\Big( -t \cdot \mathrm{Diag}(\lambda_+, \lambda_-, \ldots, \lambda_-)\Big)Q^\top\Big( (\beta_1 - s)\mathbf{1} + \beta_2\mathbf{v}\Big) + \mathbf{s}.$$

$Q$ is a projection matrix to the respective eigenspaces of $\nabla^2 \ell(\mathbf{s})$, hence since $\mathbf{1} \in \mathcal{W}_1$ and $\mathbf{v} \in \mathcal{W}_2$ we obtain

$$
\begin{aligned}
A^{lin}(t) &= \Big( \exp(-t \cdot \lambda_+)(\beta_1 - s) \Big) \mathbf{1} + \Big( \exp(-t \cdot \lambda_-) \cdot \beta_2 \Big) \mathbf{v} + \mathbf{s} \\
&= \Big( \exp(-t \cdot \lambda_+)(\beta_1 - s) + s \Big) \mathbf{1} + \Big( \exp(-t \cdot \lambda_-) \cdot \beta_2 \Big) \mathbf{v}
\end{aligned}
$$

as required. $\qquad\square$

**Remark 2.** *Note that if $\beta_2 \neq 0$ (i.e. the initialization $A^{lin}(0)$ was not in $\mathcal{W}_1$), then the solution to the system diverges from* $\mathbf{s}$*. Since $\lambda_- < 0 < \lambda_+$ we obtain*

$$
\lim_{t \to \infty} \Big( \exp(-t \cdot \lambda_+)(\beta_1 - s) + s \Big) \mathbf{1} = s \cdot \mathbf{1} = \mathbf{s}
$$

*and*

$$
\lim_{t \to \infty} \left\| \Big( \exp(-t \cdot \lambda_-) \cdot \beta_2 \Big) \mathbf{v} - \mathbf{s} \right\|_2 \to \infty .
$$

*On the other hand, if $\beta_2 = 0$ then the solution to the system converges to* $\mathbf{s}$*.*

### F.3.3 Linearization of Dynamical Systems

In F.3.2 we characterized the critical point $\mathbf{s} \in \mathbf{1}$ and established that it is the only non global minimum that we could converge to given our initialization. We would now like to show that in fact gradient flow will escape $\mathbf{s}$ and converge rapidly towards a global minimum. Corollary 2 gives some indication why this may be the case—it shows that the local linearization of the dynanics near $\mathbf{s}$ will tend to repel any trajectory which is not on the line $\mathcal{W}_1$. Intuitively one expects that once we are sufficiently close to $\mathbf{s}$, the linearized dynamics provide a sufficiently good approximation to ensure that the same conclusion will hold for the nonlinear system as well. Unfortunately, existing results from the optimization literature (*e.g.* Jin et al. [35]) give escape times which do not suffice for our purposes[12]. To obtain the required bounds on the escape time we will require some results from dynamical systems theory. Informally, the idea is that if a non linear dynamical system satisfies certain conditions on the spectrum of its linearization (these are sometimes called "non-resonance conditions"), then it is locally smoothly equivalent to its linearization. This will allow us to bound the escape time of gradient flow in terms of the closed form dynamics obtained for the linearization in Corollary 2.

We begin by defining the notions of *smooth conjugation* and *smooth linearization* of dynamical systems:

**Definition 5.** Let $f, g : \mathbb{R}^d \to \mathbb{R}^d$ be two $C^M$ vector fields with a common fixed point $\mathbf{x}_0 \in \mathbb{R}^d$, *i.e.*, $f(\mathbf{x}_0) = g(\mathbf{x}_0) = \mathbf{0}$. For any $K \in [M]$ we say that $f$ and $g$ are $C^K$-*conjugate near* $\mathbf{x}_0$ when there exist neighborhoods $\mathcal{V}_1, \mathcal{U}_1 \subseteq \mathbb{R}^d$ such that $\mathbf{x}_0 \in \mathcal{V}_1, \mathcal{U}_1$ and there exist a $C^K$-diffeomorphism $H : \mathcal{V}_1 \to \mathcal{U}_1$ satisfying the following:

- $H(\mathbf{x}_0) = \mathbf{x}_0$.

- Whenever $\mathbf{x}(t) \in \mathcal{V}_1$ is a solution of $\dot{\mathbf{x}}(t) = f(\mathbf{x}(t))$ for $t$ in some interval $\mathcal{I} \subseteq \mathbb{R}$ then $\mathbf{y}(t) = H(\mathbf{x}(t))$ is a solution of $\dot{\mathbf{y}}(t) = g(\mathbf{y}(t))$ for $t \in \mathcal{I}$.

- Whenever $\mathbf{y}(t) \in \mathcal{U}_1$ is a solution of $\dot{\mathbf{y}}(t) = g(\mathbf{y}(t))$ for $t$ in some interval $\mathcal{I} \subseteq \mathbb{R}$ then $\mathbf{x}(t) = H^{-1}(\mathbf{y}(t))$ is a solution of $\dot{\mathbf{x}}(t) = f(\mathbf{x}(t))$ for $t \in \mathcal{I}$.

The mapping $H$ is referred to as the $C^K$-*conjugation* between $\dot{\mathbf{x}}(t) = f(\mathbf{x}(t))$ and $\dot{\mathbf{y}}(t) = g(\mathbf{y}(t))$. Consider the first order Taylor's expansion of $f$ around $\mathbf{x}_0$ given by

$$
\dot{\mathbf{x}}(t) = f(\mathbf{x}(t)) = A(\mathbf{x}(t) - \mathbf{x}_0) + F(\mathbf{x}(t) - \mathbf{x}_0)
$$

---

[12]recall that our strategy is to bound the divergence between our trajectory and the reference one, and this divergence depends on the convergence time achieved by gradient flow.

where $A = Df(\mathbf{x}_0)$, $F(\mathbf{0}) = \mathbf{0}$ and $DF(\mathbf{0}) = \mathbf{0}_{d,d}$ is the $d, d$ zero matrix. The associated linear equation is given by

$$\dot{\mathbf{y}}(t) = A\big(\mathbf{y}(t) - \mathbf{x}_0\big).$$

We say that $f$ admits a $C^K$-*linearization* near $\mathbf{x}_0$ when it is $C^K$-conjugate near $\mathbf{x}_0$ with its linear approximation.

We now introduce the *Strict Hyperbolicity* property and the *Non-resonance condition* (also known as the Sternberg condition). These are sufficient conditions for a dynamical system to admit a smooth linearization and we will later show our system satisfies them.

**Definition 6.** Let $A \in \mathbb{R}^{d,d}$ be a matrix with eigenvalues $\lambda_1, \ldots, \lambda_d \in \mathbb{R}$ repeated with multiplicities. We say that $A$ is *strictly hyperbolic* when:

- For all $j \in [d]$ it holds that $\lambda_j \neq 0$.

- There exist $j_+, j_- \in [d]$ such that $\lambda_{j_+} > 0$ and $\lambda_{j_-} < 0$.

**Definition 7.** Let $A \in \mathbb{R}^{d,d}$ be a matrix with eigenvalues $\lambda_1, \ldots, \lambda_d \in \mathbb{R}$ repeated with multiplicities. For any $\mathbf{m} \in \mathbb{N}_{\geq 0}^d$ non-negative integers vector and any $\lambda \in \mathbb{R}$ we denote $\gamma(\lambda, \mathbf{m})$ as the following quantity:

$$\gamma(\lambda, \mathbf{m}) := \lambda - \sum_{k=1}^{d} m_k \cdot \lambda_k.$$

For any $N \in \mathbb{N}$ such that $N \geq 2$ we say that $A$ satisfies the *non-resonance condition* of order $N$ when for all $j \in [d]$ and all $\mathbf{m} \in \mathbb{N}_{\geq 0}^d$ such that $\sum_{k=1}^{d} m_k \in \{2, \ldots, N\}$ it holds that $\gamma(\lambda_j, \mathbf{m}) \neq 0$.

Finally, we present the property of *matrix Q-smoothness*.

**Definition 8.** Let $A \in \mathbb{R}^{d,d}$ be a matrix with eigenvalues $\lambda_1, \ldots, \lambda_d \in \mathbb{R}$ repeated with multiplicities. Suppose $A$ is strictly hyperbolic. Denote the following quantities:

$$\rho_+ := \frac{\max\{|\lambda_j| : j \in [d], \lambda_j > 0\}}{\min\{|\lambda_j| : j \in [d], \lambda_j > 0\}} \quad , \quad \rho_- := \frac{\max\{|\lambda_j| : j \in [d], \lambda_j < 0\}}{\min\{|\lambda_j| : j \in [d], \lambda_j < 0\}}.$$

Let $Q \in \mathbb{N}_{>0}$. We define the *Q-smoothness* of $A$ to be the largest integer $K \in \mathbb{N}_{\geq 0}$ for which there exist $M, N \in \mathbb{N}_{>0}$ satisfying the following:

- $Q = M + N$.

- $M - K\rho_+ \geq 0$.

- $N - K\rho_- \geq 0$.

We are now ready to present Theorem 1 of [70], which states conditions under which there exists a smooth linearization of a dynamical system.[13]

**Theorem 3** (Theorem 1 of [70] (adapted)). *Let $Q \in \mathbb{N}$ such that $Q \geq 2$. Let $f : \mathbb{R}^d \to \mathbb{R}^d$ be an analytic vector field with a fixed point $\mathbf{x}_0$. Consider the first order Taylor's expansion of $f$ around $\mathbf{x}_0$ given by*

$$\dot{\mathbf{x}}(t) = f\big(\mathbf{x}(t)\big) = A\big(\mathbf{x}(t) - \mathbf{x}_0\big) + F\big(\mathbf{x}(t) - \mathbf{x}_0\big)$$

*where $A = Df(\mathbf{x}_0)$, $F(\mathbf{0}) = \mathbf{0}$ and $DF(\mathbf{0}) = \mathbf{0}_{d,d}$. If $A$ is strictly hyperbolic (Definition 6) and satisfies the non-resonance condition of order $Q$ (Definition 7) then $f$ admits a $C^K$ linearization near $\mathbf{x}_0$ where $K$ is the Q-smoothness of $A$ (Definition 8).*

*Proof.* See proof of Theorem 1 in [70]. $\qquad\qquad\square$

Having introduced these general results on local linearization, we now show that the dynamical system induced by gradient flow admits a smooth linearization near $\mathbf{s}$. We begin by showing that $-\nabla^2 \ell(\mathbf{s})$ is strictly hyperbolic and satisfies the non-resonance condition.

---

[13]We present slightly adapted results that are specialized to our setting.

**Proposition 8.** *Consider* s *defined in Lemma 7. The hessian matrix* $-\nabla^2 \ell(\mathbf{s})$ *is strictly hyperbolic and satisfies the non-resonance condition of order* $d - 2$.

*Proof.* Per Lemma 9, it holds that

$$\lambda_+ \geq d - 1 > 0$$

and

$$-1 < \lambda_- < 0.$$

Hence by Definition 6, $\nabla^2 \ell(\mathbf{s})$ is strictly hyperbolic. Additionally, for any $m \in \{0, \ldots, d - 2\}$ we have that

$$\lambda_+ + m \cdot \lambda_- \geq d - 1 - m > 0.$$

Let $\mathbf{m} \in \mathbb{N}^d_{\geq 0}$ such that $\sum_{k=1}^d m_k \in \{2, \ldots, d - 2\}$. Per definition Definition 7 we have that

$$\gamma(\lambda_+, \mathbf{m}) = (1 - m_1)\lambda_+ - \lambda_- \sum_{k=2}^d m_k.$$

If $m_1 \in \{0, 1\}$ then since $\lambda_- < 0 < \lambda_+$ and $\sum_{k=2}^d m_k \in \{1, \ldots, d - 2\}$ we obtain

$$\gamma(\lambda_+, \mathbf{m}) \geq -\lambda_- \sum_{k=2}^d m_k > 0.$$

Otherwise, since $\sum_{k=2}^d m_k \in \{0, \ldots, d - 4\}$ we obtain by the above that

$$\gamma(\lambda_+, \mathbf{m}) \leq -\lambda_+ - \lambda_- \sum_{k=2}^d m_k \leq -(d - 1) + \sum_{k=2}^d m_k < 0.$$

Hence $\gamma(\lambda_+, \mathbf{m}) \neq 0$. Next, per Definition 7 we have that

$$\gamma(\lambda_-, \mathbf{m}) = \left(1 - \sum_{k=2}^d m_k\right)\lambda_- - m_1 \cdot \lambda_+.$$

If $m_1 = 0$ then since $1 - \sum_{k=2}^d m_k \in \{-1, \ldots, -d + 3\}$ we obtain

$$\gamma(\lambda_-, \mathbf{m}) = \left(1 - \sum_{k=2}^d m_k\right)\lambda_- > 0.$$

If $m_1 = d - 2$ then $1 - \sum_{k=2}^d m_k = 1$ and so

$$\gamma(\lambda_-, \mathbf{m}) = \lambda_- - (d - 2)\lambda_+ < 0.$$

Otherwise, since $\sum_{k=2}^d m_k - 1 \in \{0, \ldots, d - 3\}$ we obtain by the above that

$$\gamma(\lambda_-, \mathbf{m}) \leq -\lambda_+ - \left(\sum_{k=2}^d m_k - 1\right)\lambda_- \leq -(d - 1) + \sum_{k=2}^d m_k - 1 < 0.$$

Hence $\gamma(\lambda_-, \mathbf{m}) \neq 0$. Therefore by Definition 7, $-\nabla^2 \ell(\mathbf{s})$ satisfies the non-resonance condition of order $d - 2$. $\qquad\square$

Next, we turn to lower bound the $Q$-smoothness of $-\nabla^2 \ell(\mathbf{s})$.

**Proposition 9.** *For any* $Q \in \mathbb{N}_{>0}$, *the* $Q$-smoothness of $-\nabla^2 \ell(\mathbf{s})$ *is at least* $\lfloor \frac{Q}{2} \rfloor$.

*Proof.* Per Lemma 9, we have the following:

$$\rho_+ = \frac{\max\{\lambda_+\}}{\min\{\lambda_+\}} = 1 \quad, \quad \rho_- = \frac{\max\{\lambda_-\}}{\min\{\lambda_-\}} = 1.$$

Therefore per Definition 8 and since $\nabla^2 \ell(\mathbf{s})$ is strictly hyperbolic, the $Q$-smoothness of $-\nabla^2 \ell(\mathbf{s})$ is the largest $K \in \mathbb{N}_{\geq 0}$ for which there exist $M, N \in \mathbb{N}_{>0}$ such that

- $Q = M + N$.

- $M - K \geq 0$.

- $N - K \geq 0$.

One can easily verify this implies that the $Q$-smoothness of $-\nabla^2 \ell(\mathbf{s})$ is at least $\lfloor \frac{Q}{2} \rfloor$. $\qquad \square$

Finally, we are ready to prove the following proposition which shows that our dynamical system induced by gradient flow admits a linearization which is at least $C^3$.

**Proposition 10.** *The dynamical system induced by gradient flow (see Equation* (22)*) admits a linearization near* $\mathbf{s}$ *that is at least* $C^3$.

*Proof.* First note that the vector field $-\nabla \ell(A)$ which gradient flow follows is analytic. Next, per Propositions 8 and 9 it holds that $-\nabla^2 \ell(\mathbf{s})$ is strictly hyperbolic, satisfies the non-resonance condition of order at least $d - 2$, and has $Q$-smoothness of at least $\lfloor \frac{Q}{2} \rfloor$ for any $Q \in \mathbb{N}_{>0}$. Hence, by Theorem 3 the vector field $-\nabla \ell(A)$ admits a $C^{\lfloor \frac{d-2}{2} \rfloor}$-linearization near $\mathbf{s}$. The proof concludes by noting that $d \geq 8$ hence $\lfloor \frac{d-2}{2} \rfloor \geq 3$. $\qquad \square$

We denote the above linearization by $H : \mathcal{V}_1 \to \mathcal{U}_1$, where the neighborhoods $\mathcal{V}_1, \mathcal{U}_1 \subseteq \mathbb{R}^d$ are such that $\mathbf{s} \in \mathcal{V}_1, \mathcal{U}_1$. To set the stage for the rest of the proof, we prove the following proposition that considers a restriction of $H$ to a smaller domain that satisfies a few additional conditions which we will require later.

**Proposition 11.** *There exists* $r_1 > 0$ *which satisfies the following:*

1. $r_1 \leq \frac{1}{2d}$.

2. *For any* $A \in \overline{B_{r_1}}(\mathbf{s})$ *it holds that*

$$|\lambda_{min}(\nabla^2 \ell(A))| \leq 2|\lambda_-|.$$

3. $H|_{\overline{B_{r_1}}(\mathbf{s})}$ *is Lipschitz and there exist* $r_2 \in (0, r_1)$ *and* $r_3 > 0$ *such that* $H^{-1}|_{\overline{B_{r_3}}(\mathbf{s})}$ *is Lipschitz and it holds that*

$$H[\overline{B_{r_2}}(\mathbf{s})] \subseteq \overline{B_{r_3}}(\mathbf{s}) \subseteq H[\overline{B_{r_1}}(\mathbf{s})].$$

*Proof.* We show there exist three non empty intervals of the form $(0, b_i]$ for $i \in [3]$, such that if $r_1 \in (0, b_i]$ then it satisfies the corresponding requirement above. This would imply that the minimal upper limit $r_1 := \min\{b_1, b_2, b_3\}$ satisfies all requirements. The first condition is trivial, with $b_1 = \frac{1}{2d}$. Next, since $\nabla \ell(A)$ is analytic it holds that $\nabla^2 \ell(A)$ is symmetric for any $A \in \mathbb{R}^d$. Since $\lambda_- < 0$, by the continuity of the eigenvalues of $\nabla^2 \ell(A)$ around $\mathbf{s}$ there exists $b_2 > 0$ such that the second requirement is satisfied for all $(0, b_2]$. Lastly, since $H : \mathcal{V}_1 \to \mathcal{U}_1$ is $C^3$ and since $\mathcal{V}_1, \mathcal{U}_1$ are neighborhoods of $\mathbf{s}$, we can invoke Lemma 28 which states that there exists $b_3 > 0$ such that for any $b \in (0, b_3]$ the third and fourth requirements are satisfied. $\qquad \square$

### F.3.4 Movement Towards the Saddle s

Having established key properties of the loss landscape, we are ready to begin the dynamical analysis of the gradient flow trajectories over time. We first give a simple bound on the magnitude of the entries of $A(t)$.

**Lemma 11.** *Suppose we initialize at* $A(0) \in \mathcal{I}_0$ *and at* $A^{ref}(0)$, *and evolve* $A(t)$ *and* $A^{ref}(t)$ *according to Equation* (22)*. For any* $t \geq 0$ *and any* $j \in [d]$ *it holds that*

$$A_j(t), A_j^{ref}(t) \in [-3, 3].$$

*Proof.* First, we have shown in Lemma 5 that the initialization $\mathcal{I}_0$ guarantees all points encountered by gradient flow have loss no larger than 1. Assume on the contrary that there exist $t \geq 0$ and $j \in [d]$ for which

$$A_j(t) \notin [-3, 3] \,.$$

Since $L - 1$ is even, we obtain that $a_j(t)^{L-1} \geq 3^{L-1} > 3$ and that any $k \in [d], k \neq j$ satisfies $a_k(t)^{L-1} \geq 0$. Hence, we obtain that

$$
\begin{aligned}
\ell\big(A(t)\big) &= \left( \Big(1 - \sum_{k=1}^{d} a_k(t)^{L-1}\Big)^2 + \Big(1 - \sum_{k=1}^{d} a_k(t)\Big)^2 \right) \\
&\geq \left( 1 - \sum_{k=1}^{d} a_k(t)^{L-1} \right)^2 \\
&\geq (3 - 1)^2 \\
&> 1 \,.
\end{aligned}
$$

in contradiction to Lemma 22. The proof is identical when we consider the reference trajectory. $\quad\square$

The above yields the following useful corollary.

**Corollary 3.** *There exists $N > 0$ such if we initialize at $A(0) \in \mathcal{I}_0$ and at $A^{ref}(0)$, and evolve $A(t)$ and $A^{ref}(t)$ according to Equation (22) then for any $t \geq 0$ the functions $-\nabla \ell\big(A(t)\big)$ and $-\nabla \ell\big(A^{ref}(t)\big)$ are $N$-Lipschitz.*

*Proof.* As shown in Lemma 11, all points encountered by gradient flow are contained in the compact set $[-3, 3]^d$. The claim thus follows from the fact that $\ell(A)$ is analytic. $\quad\square$

We continue to prove the following lemma which analyzes the trajectories when initializing in an interval of points on the line $\mathcal{W}_1$ (Equation (16)).

**Lemma 12.** *Let $a_1, a_2 \in [-\frac{1}{d}, \frac{4}{d}] \setminus \{s\}$ such that $a_1 \neq a_2$. Suppose we initialize at $A_1(0) = a_1 \cdot \mathbf{1}$ and $A_2(0) = a_2 \cdot \mathbf{1}$, and evolve $A_1(t)$ and $A_2(t)$ according to Equation (22). It holds that:*

- *There exist functions $a_1, a_2 : \mathbb{R}_{\geq 0} \to \mathbb{R}$ such that*

$$A_1(t) = a_1(t) \cdot \mathbf{1}, \quad A_2(t) = a_2(t) \cdot \mathbf{1} \,.$$

- *For any $t \geq 0$ it holds that $a_2(t) < a_1(t) \iff a_2 < a_1$.*

- *For any $r > 0$ there exists $t_1 \geq 0$ such that for any $t \geq t_1$ it holds that $A_1(t), A_2(t) \in \overline{B_r}(\mathbf{s})$.*

*Proof.* When initializing at $A(0) = a \cdot \mathbf{1}$ and evolving $A(t)$ according to the gradient flow dynamics, all entries evolve according to the same dynamics and therefore must stay equal throughout the optimization. Concretely, all entries obey the following dynamics:

$$\dot{a}(t) = \left( (L-1)\big(1 - d \cdot a(t)^{L-1}\big)a(t)^{L-2} + \big(1 - d \cdot a(t)\big) \right) \,.$$

Hence, the first claim holds. Rewriting the above in terms of Lemma 7, we have

$$\dot{a}(t) = -\frac{1}{d} f'(a(t)) \,.$$

In Lemmas 6 and 7, we showed that the above expression is positive for $a(t) \in [-\frac{1}{d}, s)$ and equals zero at $s$. We now show that the above expression is negative for $a(t) \in (s, \frac{4}{d}]$. Indeed, since $d \geq 8$

we have that

$$-\frac{1}{d}f'\left(\frac{4}{d}\right) = (L-1)(1 - d \cdot \left(\frac{4}{d}\right)^{L-1})\left(\frac{4}{d}\right)^{L-2} + (1 - d \cdot \left(\frac{4}{d}\right))$$

$$\leq (L-1)\left(\frac{4}{d}\right)^{L-2} - 3$$

$$\leq \underbrace{\frac{L-1}{1.5^{L-2}}\left(\frac{4}{5\frac{1}{3}}\right)^{L-2}}_{=:h(L)} - 3$$

$$\leq 0.75 - 3$$

$$< 0,$$

where the second to last inequality stems from the fact that $h(L)$ is decreasing for $L \geq 4$, that $h(4) = 0.75$, and that $L \geq 7$. Next, we also have for any $a \in (s, \frac{4}{d}]$ that

$$-\frac{1}{d}f''(a) = -(L-1)(2L-3)d \cdot a^{2L-4} + (L-1)(L-2)a^{L-3} - d$$

$$\leq -d + (L-1)(L-2)a^{L-3}$$

$$\leq -d + (L-1)(L-2)\left(\frac{4}{d}\right)^{L-3}$$

$$\leq -d + \underbrace{\frac{(L-1)(L-2)}{1.5^{L-3}}\left(\frac{4}{5\frac{1}{3}}\right)^{L-3}}_{=:g(L)}$$

$$\leq -d + \frac{15}{8}$$

$$< 0,$$

where the second to last inequality stems from the fact that $g(L)$ is decreasing for $L \geq 7$, that $g(7) = \frac{15}{8}$, and that $L \geq 7$. Therefore, since $-\frac{1}{d}f'(s) = 0$ and from monotonicity we obtain that $-\frac{1}{d}f'\left(a(t)\right)$ is negative for $a(t) \in (s, \frac{4}{d}]$. We continue by noting that per Lemma 30, trajectories of the same system of ODEs with different initalizations must never meet, hence by continuity it must hold that $a_2(t) < a_1(t) \iff a_2 < a_1$ for all $t \geq 0$. Finally, since $s$ is a critical point in the interval and since $a_1(t), a_2(t)$ evolve monotonically (increase if initialized $< s$ and decrease otherwise), we get by Lemma 30 that $a_1(t)$ and $a_2(t)$ cannot reach $s$ in any finite time. However, since $s$ is the unique critical point in the interval, $a_1(t)$ and $a_2(t)$ must converge to $s$ as $t \to \infty$. Hence, there exists some time $t_1 \geq 0$ such that for any $t \geq t_1$ the entries of both $A_1(t)$ and $A_2(t)$ are within $\overline{B_r}(s)$. $\quad\square$

We use $A^Z(t)$ and $A^-(t)$ to denote the trajectories generated by initializing at $\mathbf{0}$ and $-\frac{1}{d}\cdot\mathbf{1}$ respectively and evolving according to Equation (22). Additionally, for any $r > 0$ we use $t_1(r) \geq 0$ to denote the minimal time which satisfies

$$A^Z\left(t_1(r)\right) \in \overline{B_r}(\mathbf{s}).$$

Note this means that for $r \in (0, s)$ we have $A^Z\left(t_1(r)\right) = \frac{r}{\sqrt{d}} \cdot \mathbf{1} + \mathbf{s}$.

We denote the following projections of the trajectory $A(t)$ and the reference trajectory $A^{ref}(t)$, which will be used in the rest of the proof.

**Definition 9.** Suppose we initialize at $A(0) \in \mathcal{I}_0$ and at $A^{ref}(0)$, and evolve $A(t)$ and $A^{ref}(t)$ according to Equation (22). For any $t \geq 0$, we denote the projections of $A(t)$ and $A^{ref}(t)$ to the subspace $\mathcal{W}_1$ using

$$\beta_1(t) \cdot \mathbf{1} \quad, \quad \beta_1^{ref}(t) \cdot \mathbf{1}$$

where $\beta_1(t), \beta_1^{ref}(t) \in \mathbb{R}$. Additionally, we denote the projections of $A(t)$ and $A^{ref}(t)$ to the subspace $\mathcal{W}_2$ using

$$\beta_2(t) \cdot \mathbf{v}(t) \quad, \quad \beta_2^{ref}(t) \cdot \mathbf{v}^{ref}(t)$$

where $\beta_2(t), \beta_2^{ref}(t) \in \mathbb{R}$ and $\mathbf{v}(t), \mathbf{v}^{ref}(t) \in \mathcal{W}_2$ are unit vectors. Per Equation (16), $\mathcal{W}_1$ and $\mathcal{W}_2$ are orthogonal and span $\mathbb{R}^d$, hence we may write

$$A(t) = \beta_1(t) \cdot \mathbf{1} + \beta_2(t) \cdot \mathbf{v}(t),$$
$$A^{ref}(t) = \beta_1^{ref}(t) \cdot \mathbf{1} + \beta_2^{ref}(t) \cdot \mathbf{v}^{ref}(t).$$

**Remark 3.** *Per Equation* (16), $\mathcal{W}_1$ *and* $\mathcal{W}_2$ *are orthogonal therefore since* $(\beta_1(t) \cdot \mathbf{1})^\top (\beta_2(t) \cdot \mathbf{v}(t)) = 0$ *and* $(\beta_1^{ref}(t) \cdot \mathbf{1})^\top (\beta_2(t)^{ref} \cdot \mathbf{v}^{ref}(t)) = 0$, *we obtain*

$$Dist(A(t), \mathcal{W}_1) = \|\beta_2(t) \cdot \mathbf{v}(t)\|_2 = |\beta_2(t)|,$$
$$Dist(A^{ref}(t), \mathcal{W}_1) = \|\beta_2^{ref}(t) \cdot \mathbf{v}^{ref}(t)\|_2 = |\beta_2^{ref}(t)|$$

*and*

$$Dist(A(t), \mathcal{W}_2) = \|\beta_1(t) \cdot \mathbf{1}\|_2 = \sqrt{d}|\beta_1(t)|,$$
$$Dist(A^{ref}(t), \mathcal{W}_2) = \|\beta_1^{ref}(t) \cdot \mathbf{1}\|_2 = \sqrt{d}|\beta_1^{ref}(t)|.$$

Before proving the main claim of this section, we introduce another condition on the initialization which we denote $\mathcal{I}_3$.

**Definition 10.** Let $r > 0$. We use $\mathcal{I}_3(r)$ to denote the following subset of $\mathcal{I}_0$:

$$\mathcal{I}_3(r) := \left\{ A \in \mathcal{I}_0 : \alpha \leq \frac{\min\left\{ r, \left\| A^Z(t_1(r)) - A^-(t_1(r)) \right\|_2 \right\}}{6d} \exp\left( -N \cdot t_1(r) \right), \zeta_d \leq \frac{1}{2} \right\}$$

for $N$ of Corollary 3 and for $A^Z(t)$, $A^-(t)$ and $t_1(r)$ of Lemma 12.

We are now ready to prove the main claim of this section, which states that under the above on the initalization, both the original and reference trajectories must enter a sufficiently small sphere around $\mathbf{s}$ and furthermore they arrive at points that are sufficiently faraway from $\mathcal{W}_1$.

**Proposition 12.** *Let* $r \in (0, s)$. *Suppose we initialize at* $A(0) \in \mathcal{I}_3(\frac{r}{4})$ *and at* $A^{ref}(0)$, *and evolve* $A(t)$ *and* $A^{ref}(t)$ *according to Equation* (22). *There exist constants* $D_+(r), D_-(r) > 0$ *such that:*

- $A(t_1(\frac{r}{4})), A^{ref}(t_1(\frac{r}{4})) \in \overline{B_{\frac{r}{2}}}(\mathbf{s})$.

- $|\beta_2(t_1(\frac{r}{4}))|, |\beta_2^{ref}(t_1(\frac{r}{4}))| \in [\alpha \cdot D_-(r), \alpha \cdot D_+(r)]$.

*Proof.* Consider the trajectories $A^Z(t)$ and $A^-(t)$ introduced in Lemma 12. Per Lemma 12, for any time $t \geq 0$ and any index $j \in [d]$ we have

$$a_j^-(t) < a_j^Z(t) < s.$$

We begin by showing that for $A(0) \in \mathcal{I}_3(\frac{r}{4})$, the distance between $A^Z(t_1(\frac{r}{4}))$ and $A(t_1(\frac{r}{4}))$ is at most $\frac{r}{24}$. First note that per Lemma 12, $A^Z(t)$ never leaves $\overline{B_s}(\mathbf{s}) \subseteq [-3, 3]^d$. Thus per Corollary 3, both $A^Z(t)$ and $A(t)$ are always contained in a compact domain where the vector field $-\nabla \ell(A)$ is $N$-Lipschitz. Therefore, we can invoke Lemma 29 which results in the following:

$$\|A^Z(t_1(\tfrac{r}{4})) - A(t_1(\tfrac{r}{4}))\|_2 \leq \|A^Z(0) - A(0)\|_2 \cdot \exp\left( N \cdot t_1(\tfrac{r}{4}) \right)$$
$$= \|A(0)\|_2 \cdot \exp\left( N \cdot t_1(\tfrac{r}{4}) \right)$$
$$\leq \alpha \cdot d \cdot \exp\left( N \cdot t_1(\tfrac{r}{4}) \right).$$

Per Definition 10, $\alpha$ satisfies

$$\alpha \leq \frac{\min\left\{ \frac{r}{4}, \left\| A^Z(t_1(\frac{r}{4})) - A^-(t_1(\frac{r}{4})) \right\|_2 \right\}}{6d} \exp\left( -N \cdot t_1(\tfrac{r}{4}) \right).$$

Hence, we obtain that

$$\left\| A^Z\big(t_1(\tfrac{r}{4})\big) - A\big(t_1(\tfrac{r}{4})\big) \right\|_2$$

$$\leq \frac{\min\left\{\tfrac{r}{4}, \left\| A^Z\big(t_1(\tfrac{r}{4})\big) - A^-\big(t_1(\tfrac{r}{4})\big)\right\|_2\right\}}{6d} \exp(-N \cdot t_1(\tfrac{r}{4})) \cdot d \cdot \exp(N \cdot t_1(\tfrac{r}{4}))$$

$$= \frac{\min\left\{\tfrac{r}{4}, \left\| A^Z\big(t_1(\tfrac{r}{4})\big) - A^-\big(t_1(\tfrac{r}{4})\big)\right\|_2\right\}}{6}$$

$$\leq \frac{r}{24}.$$

Therefore, using the triangle inequality we obtain

$$\left\| A\big(t_1(\tfrac{r}{4})\big) - \mathbf{s} \right\|_2 \leq \left\| A^Z\big(t_1(\tfrac{r}{4})\big) - \mathbf{s} \right\|_2 + \left\| A\big(t_1(\tfrac{r}{4})\big) - A^Z\big(t_1(\tfrac{r}{4})\big) \right\|_2 \leq \frac{r}{4} + \frac{r}{24} \leq \frac{r}{2}.$$

Hence, $A(t_1) \in \overline{B_{\frac{r}{2}}}(\mathbf{s})$. Next, by Remark 3 we obtain that

$$\left| \beta_2\big(t_1(\tfrac{r}{4})\big) \right| = \mathrm{Dist}\left( A\big(t_1(\tfrac{r}{4})\big), \mathcal{W}_1 \right).$$

By Lemma 12 we have $A^Z\big(t_1(\tfrac{r}{4})\big) \in \mathcal{W}_1$, hence

$$\left| \beta_2\big(t_1(\tfrac{r}{4})\big) \right| \leq \left\| A^Z\big(t_1(\tfrac{r}{4})\big) - A\big(t_1(\tfrac{r}{4})\big) \right\|_2 \leq \alpha \cdot d \cdot \exp\left( N \cdot t_1(\tfrac{r}{4}) \right).$$

Thus, denoting $D_+(r) := d \cdot \exp\left( N \cdot t_1(\tfrac{r}{4}) \right)$ we get the first part of the second claim. We now show that $\beta_1\big(t_1(\tfrac{r}{4})\big) \in (-\tfrac{1}{d}, s)$. Per Lemma 12, we get by definition of $t_1$ that

$$\left\| A^Z\big(t_1(\tfrac{r}{4})\big) - \mathbf{s} \right\|_2 = \frac{r}{4}$$

and so since $\|A^Z\big(t_1(\tfrac{r}{4})\big) - A\big(t_1(\tfrac{r}{4})\big)\|_2 \leq \frac{r}{24}$ and $r < s$ it must hold that $\beta_1\big(t_1(\tfrac{r}{4})\big) < s$. Since

$$\left\| A^Z\big(t_1(\tfrac{r}{4})\big) - A\big(t_1(\tfrac{r}{4})\big) \right\|_2 \leq \frac{\left\| A^Z\big(t_1(\tfrac{r}{4})\big) - A^-\big(t_1(\tfrac{r}{4})\big) \right\|_2}{6}$$

it must hold that

$$\beta_1\big(t_1(\tfrac{r}{4})\big) > a^-\big(t_1(\tfrac{r}{4})\big)$$

where $A^-\big(t_1(\tfrac{r}{4})\big) = a^-\big(t_1(\tfrac{r}{4})\big) \cdot \mathbf{1}$. Note that by Lemma 12 we obtain

$$\beta_1\big(t_1(\tfrac{r}{4})\big) > a^-\big(t_1(\tfrac{r}{4})\big) > -\frac{1}{d}$$

as $a^-(t)$ is monotonically increasing. Therefore by Lemma 12 and by continuity, there must exist some point $a \in \left( -\tfrac{1}{d}, \beta_1\big(t_1(\tfrac{r}{4})\big) \right)$ such that if we initialize $A^a(0) = a \cdot \mathbf{1}$ and evolve $A^a(t)$ according to the gradient flow dynamics, then it holds that

$$A^a\big(t_1(\tfrac{r}{4})\big) = \beta_1\big(t_1(\tfrac{r}{4})\big) \cdot \mathbf{1}.$$

Per Lemma 12, $A^a(t)$ never leaves $[-3,3]^d$ where the vector field $-\nabla \ell(A)$ is $N$-Lipschitz. Thus, invoking Lemma 29 we obtain

$$
\begin{aligned}
\left| \beta_2\big(t_1(\tfrac{r}{4})\big) \right| &= \left\| \beta_2\big(t_1(\tfrac{r}{4})\big) \cdot \mathbf{v}\big(t_1(\tfrac{r}{4})\big) \right\|_2 \\
&= \left\| \beta_2\big(t_1(\tfrac{r}{4})\big) \cdot \mathbf{v}\big(t_1(\tfrac{r}{4})\big) + \beta_1\big(t_1(\tfrac{r}{4})\big) \cdot \mathbf{1} - \beta_2\big(t_1(\tfrac{r}{4})\big) \cdot \mathbf{v}\big(t_1(\tfrac{r}{4})\big) \right\|_2 \\
&= \left\| A\big(t_1(\tfrac{r}{4})\big) - A^a\big(t_1(\tfrac{r}{4})\big) \right\|_2 \\
&\geq \left\| A(0) - A^a(0) \right\|_2 \cdot \exp\big( - N \cdot t_1(\tfrac{r}{4}) \big).
\end{aligned}
$$

As $A^a(0) \in \mathcal{W}_1$, we can lower bound the right hand side by the distance between $A(0)$ and $\mathcal{W}_1$ and obtain

$$
\left| \beta_2\big(t_1(\tfrac{r}{4})\big) \right| \geq \mathrm{Dist}\big(A(0), \mathcal{W}_1\big) \cdot \exp\big( - N \cdot t_1(\tfrac{r}{4}) \big).
$$

Next, observe that $\zeta_d \leq \tfrac{1}{2}$ since $A(0) \in \mathcal{I}_3(\tfrac{r}{4})$, hence

$$
\zeta_1 - \frac{\sum_{k=1}^d \zeta_k}{d} \geq 1 - \frac{d-1}{d} - \frac{1}{2d} = \frac{1}{2d}.
$$

Therefore,

$$
\begin{aligned}
\mathrm{Dist}\big(A(0), \mathcal{W}_1\big) &= \sqrt{ \sum_{k=1}^d \left( \alpha \cdot \zeta_k - \frac{\sum_{k=1}^d \alpha \cdot \zeta_k}{d} \right)^2 } \\
&= \alpha \sqrt{ \sum_{k=1}^d \left( \zeta_k - \frac{\sum_{k=1}^d \zeta_k}{d} \right)^2 } \\
&\geq \alpha \cdot \left| \zeta_1 - \frac{\sum_{k=1}^d \zeta_k}{d} \right| \\
&\geq \alpha \cdot \frac{1}{2d}.
\end{aligned}
$$

Hence, we meet the second part of the second claim with $D_-(r)$ defined as

$$
D_-(r) := \frac{1}{2d} \cdot \exp\big( - N \cdot t_1(\tfrac{r}{4}) \big) > 0.
$$

Note that the proof for the reference case is identical. $\qquad\square$

We note the following remark which deals with the value of points in a sufficiently small sphere around $\mathbf{s}$.

**Remark 4.** *Let $\mu > 0$. The objective $\ell$ is continuous and $\ell(\mathbf{s}) > 0$ there exists $\bar{r}(\mu) > 0$ such that any $A \in \overline{B_{\bar{r}(\mu)}}(\mathbf{s})$ satisfies*

$$
\ell(A) \leq (1 + \frac{\mu}{4}) \cdot \ell(\mathbf{s}).
$$

We conclude this section by proving the following corollary, which states that when $A(0) \in \mathcal{I}_3$, a set of additional properties are satisfied.

**Corollary 4.** *Let $\mu > 0$. Consider $\widetilde{r}(\mu) := \min\{r_2, \bar{r}(\mu)\}$ for the respective $r_2$ and $\bar{r}(\mu)$ of Proposition 11 and Remark 4. Suppose we initialize at $A(0) \in \mathcal{I}_3(\tfrac{\widetilde{r}}{4})$ and at $A^{ref}(0)$, and evolve $A(t)$ and $A^{ref}(t)$ according to Equation (22). There exist constants $D_+(\mu), D_-(\mu) > 0$ such that:*

- $A\big(t_1(\tfrac{\widetilde{r}(\mu)}{4})\big), A^{ref}\big(t_1(\tfrac{\widetilde{r}(\mu)}{4})\big) \in \overline{B_{\frac{r_2}{2}}}(\mathbf{s})$.

- $\left|\beta_2\big(t_1(\frac{\widetilde{r}(\mu)}{4})\big)\right|, \left|\beta_2^{ref}\big(t_1(\frac{\widetilde{r}(\mu)}{4})\big)\right| \in [\alpha \cdot D_-(\mu), \alpha \cdot D_+(\mu)].$

- $\ell\left(A\big(t_1(\frac{\widetilde{r}(\mu)}{4})\big)\right), \ell\left(A^{ref}\big(t_1(\frac{\widetilde{r}(\mu)}{4})\big)\right) \le (1+\frac{\mu}{4})\ell(\mathbf{s}).$

*Proof.* We consider the constants $D_+(\mu) := D_+(\widetilde{r}(\mu))$ and $D_-(\mu) := D_-(\widetilde{r}(\mu))$ from Proposition 12. Per Proposition 12 and since $A(0) \in \mathcal{I}_3(\frac{\widetilde{r}(\mu)}{4})$, we have that:

- $A\big(t_1(\frac{\widetilde{r}(\mu)}{4})\big), A^{ref}\big(t_1(\frac{\widetilde{r}(\mu)}{4})\big) \in \overline{B_{\frac{\widetilde{r}(\mu)}{2}}}(\mathbf{s}).$

- $\left|\beta_2\big(t_1(\frac{\widetilde{r}(\mu)}{4})\big)\right|, \left|\beta_2^{ref}\big(t_1(\frac{\widetilde{r}(\mu)}{4})\big)\right| \in [\alpha \cdot D_-(\mu), \alpha \cdot D_+(\mu)].$

As $\widetilde{r}(\mu) \le r_2, \overline{r}(\mu)$, we immediately obtain that

$$A\big(t_1(\frac{\widetilde{r}(\mu)}{4})\big), A^{ref}\big(t_1(\frac{\widetilde{r}(\mu)}{4})\big) \in \overline{B_{\frac{r_2}{2}}}(\mathbf{s})$$

and

$$A\big(t_1(\frac{\widetilde{r}(\mu)}{4})\big), A^{ref}\big(t_1(\frac{\widetilde{r}(\mu)}{4})\big) \in \overline{B_{\frac{\overline{r}(\mu)}{2}}}(\mathbf{s}).$$

Finally, recall Remark 4 which combined with the latter argument results in

$$\ell\left(A\big(t_1(\frac{\widetilde{r}(\mu)}{4})\big)\right), \ell\left(A^{ref}\big(t_1(\frac{\widetilde{r}(\mu)}{4})\big)\right) \le (1+\frac{\mu}{4})\ell(\mathbf{s})$$

as required. $\square$

### F.3.5 Escape From the Saddle s

In the previous section, we showed that the gradient flow trajectories must reach a sufficiently small sphere around $\mathbf{s}$. Our goal in this section is showing that not only do both trajectories escape it, but they also do it fast enough[14]. To begin this section, we prove the following three lemmas regarding the diffeomorphism $H$ from Proposition 10. The following lemma proves that $W_1$ is mapped into itself under $H$.

**Lemma 13.** *Let $A \in \overline{B_{r_2}}(\mathbf{s}) \setminus \{\mathbf{s}\}$ and denote $\widetilde{A} := H(A)$. If $A \in W_1$ then $\widetilde{A} \in W_1$.*

*Proof.* Since $A \in W_1$ there exists $a \in \overline{B_{r_2}}(\mathbf{s})$ such that $A = a \cdot \mathbf{1}$. Per Proposition 11 and Lemma 7, it holds that $r_2 < r_1 \le \frac{1}{2d}$ and $s \in [\frac{1}{d}, \frac{3}{d}]$. Thus we obtain that $a \in [0, \frac{4}{d}]$. Assume on the contrary that $\widetilde{A} \notin W_1$. On the one hand, if we initialize at $A(0) = a \cdot \mathbf{1}$ and evolve $A(t)$ according to Equation (22), then per Lemma 12 $A(t) \in \overline{B_{r_2}}(\mathbf{s})$ for all $t \ge 0$, and furthermore $\lim_{t\to\infty} A(t) = \mathbf{s}$. By continuity $H(A(t))$ converges to s as well. On the other hand, if we initialize at $\widetilde{A}(0) = \widetilde{A}$ and evolve $\widetilde{A}(t)$ according to the linear approximation around $\mathbf{s}$ of the gradient flow dynamics (see Lemma 10), then per Remark 2 the solution $\widetilde{A}(t)$ diverges away from $\mathbf{s}$ (since the projection of $\widetilde{A}$ to $W_2$ is not zero). This contradicts our assumption that H is a conjugation (see Definition 5). $\square$

The following lemma proves the existence of two points in $W_1$ that are mapped by $H$ to "opposite sides" of $\mathbf{s}$.

**Lemma 14.** *There exists $a_1, a_2 \in [s - \frac{r_2}{2\sqrt{d}}, s + \frac{r_2}{2\sqrt{d}}] \setminus \{s\}$ such that there exist $\widetilde{a}_1 \in [s - \frac{r_3}{\sqrt{d}}, s)$ and $\widetilde{a}_2 \in (s, s + \frac{r_3}{\sqrt{d}}]$ for which either*

$$H(a_1 \cdot \mathbf{1}) = \widetilde{a}_1 \cdot \mathbf{1} \quad , \quad H(a_2 \cdot \mathbf{1}) = \widetilde{a}_2 \cdot \mathbf{1}$$

*or*

$$H(a_1 \cdot \mathbf{1}) = \widetilde{a}_2 \cdot \mathbf{1} \quad , \quad H(a_2 \cdot \mathbf{1}) = \widetilde{a}_1 \cdot \mathbf{1}.$$

---

[14]recall that the divergence between the two trajectories depends on the convergence time achieved by gradient flow.

*Proof.* Consider $a_1 = s - \frac{r_2}{4\sqrt{d}}$ and $a_2 = s + \frac{r_2}{4\sqrt{d}}$. Both $a_1 \cdot \mathbf{1}$ and $a_2 \cdot \mathbf{1}$ are within $\mathcal{W}_1 \cap \overline{B_{r_2}}(\mathbf{s})$ and so by Proposition 11 and Lemma 13 it holds that $H(a_1 \cdot \mathbf{1}), H(a_2 \cdot \mathbf{1}) \in \mathcal{W}_1 \cap \overline{B_{r_3}}(\mathbf{s})$. Thus we can denote $H(a_1 \cdot \mathbf{1}) = \widetilde{a}_1 \cdot \mathbf{1}$ and $(a_2 \cdot \mathbf{1}) = \widetilde{a}_2 \cdot \mathbf{1}$ for some $\widetilde{a}_1, \widetilde{a}_2 \in [s - \frac{r_3}{\sqrt{d}}, s + \frac{r_3}{\sqrt{d}}]$. $\widetilde{a}_1, \widetilde{a}_2$ are distinct and different than $s$ since $a_1, a_2 \neq s$ are distinct and since $H$ is a homeomorphism with $H(\mathbf{s}) = \mathbf{s}$. Assume WLOG that $\widetilde{a}_1 < \widetilde{a}_2$ (otherwise we flip the indices). Assume on the contrary that $\widetilde{a}_1, \widetilde{a}_2 > s$ (the case where $\widetilde{a}_1, \widetilde{a}_2 < s$ is symmetric). Per Remark 2, if we initialize at $\widetilde{A}(0) = \widetilde{a}_2 \cdot \mathbf{1}$ and evolve $\widetilde{A}(t)$ according to the linear approximation around $\mathbf{s}$ of the gradient flow dynamics, then our trajectory (which converges to $\mathbf{s}$) must reach $\widetilde{a}_1 \cdot \mathbf{1}$ after some finite time $t_2$, *i.e.* we obtain $\widetilde{A}(t_2) = \widetilde{a}_1 \cdot \mathbf{1}$. Thus we obtain that

$$H^{-1}\big(\widetilde{A}(t_2)\big) = H^{-1}(\widetilde{a}_1 \cdot \mathbf{1}) = H^{-1}\big(H(a_1 \cdot \mathbf{1})\big) = a_1 \cdot \mathbf{1} .$$

Hence, per Proposition 10 if we initialize at $A(0) = a_2 \cdot \mathbf{1}$ and evolve $A(t)$ according to the gradient flow dynamics, we would get that

$$A(t_2) = H^{-1}\big(\widetilde{A}(t_2)\big) = a_1 \cdot \mathbf{1} .$$

The proof concludes by noting that the above is a contradiction to Lemma 12. $\qquad\square$

The following lemma proves that $W_1$ is mapped into itself under $H^{-1}$.

**Lemma 15.** *Let $\widetilde{A} \in \overline{B_{r_3}}(\mathbf{s}) \setminus \{\mathbf{s}\}$ and denote $A := H^{-1}(\widetilde{A})$. If $\widetilde{A} \in \mathcal{W}_1$ then $A \in \mathcal{W}_1$.*

*Proof.* Since $\widetilde{A} \in \mathcal{W}_1 \cap \overline{B_{r_3}}(\mathbf{s})$ there exists $\widetilde{a} \in [s - r_3, s + r_3] \setminus \{s\}$ such that $\widetilde{A} = \widetilde{a} \cdot \mathbf{1}$. Assume WLOG that $\widetilde{a} \in [s - r_3, s)$ (the opposite case is symmetric). Per Lemma 14, there exists $a' \in [s - \frac{r_2}{2\sqrt{d}}, s + \frac{r_2}{2\sqrt{d}}] \setminus \{s\}$ such that there exists $\widetilde{a}' \in [s - \frac{r_3}{\sqrt{d}}, s)$ for which

$$H(a' \cdot \mathbf{1}) = \widetilde{a}' \cdot \mathbf{1} .$$

Assume that $\widetilde{a} \leq \widetilde{a}'$. By Lemma 10, if we initialize at $\widetilde{A}(0) = \widetilde{A}$ and evolve $\widetilde{A}(t)$ according to the linear approximation around $\mathbf{s}$ of the gradient flow dynamics, then our trajectory (which converges to $\mathbf{s}$) must reach $\widetilde{a}' \cdot \mathbf{1}$ after some finite time $t_2$, *i.e.* we obtain $\widetilde{A}(t_2) = \widetilde{a}' \cdot \mathbf{1}$. Thus we obtain that

$$H^{-1}\big(\widetilde{A}(t_2)\big) = H^{-1}(\widetilde{a}' \cdot \mathbf{1}) = H^{-1}\big(H(a' \cdot \mathbf{1})\big) = a' \cdot \mathbf{1} .$$

Hence, per Proposition 10 if we initialize at $A(0) = A = H^{-1}(\widetilde{A})$ and evolve $A(t)$ according to the gradient flow dynamics, we would get that

$$A(t_2) = H^{-1}\big(\widetilde{A}(t_2)\big) = a' \cdot \mathbf{1} .$$

Invoking Lemma 2 we conclude that $A \in \mathcal{W}_1$. Assume that $\widetilde{a} > \widetilde{a}'$. By Lemma 10, if we initialize at $\widetilde{A}'(0) = \widetilde{A}'$ and evolve $\widetilde{A}'(t)$ according to the linear approximation around $\mathbf{s}$ of the gradient flow dynamics, then our trajectory (which converges to $\mathbf{s}$) must reach $\widetilde{a} \cdot \mathbf{1}$ after some finite time $t_2$, *i.e.* we obtain $\widetilde{A}'(t_2) = \widetilde{a} \cdot \mathbf{1} = \widetilde{A}$. On the one hand, note that $H^{-1}(\widetilde{A}) = A$. On the other hand, if we initialize at $A'(0) = a' \cdot \mathbf{1}$ and evolve $A'(t)$ according to the gradient flow dynamics (defined in Equation (21)), we get by Proposition 10 that $A'(t_2) = H^{-1}\big(\widetilde{A}'(t_2)\big)$. Thus, $A'(t_2) = A$. The proof concludes by invoking Lemma 12 which states that $A'(t) \in \mathcal{W}_1$ for any $t \geq 0$. $\qquad\square$

The following two lemmas give bounds on the original dynamics in terms of the linearized ones.

**Lemma 16.** *Let $A \in \overline{B_{r_2}}(\mathbf{s}) \setminus \{\mathbf{s}\}$. It holds that*

$$Dist(A, \mathcal{W}_1) \leq \widetilde{G} \cdot Dist\big(H(A), \mathcal{W}_1\big)$$

*where $\widetilde{G}$ is the Lipschitz coefficient of $H^{-1}|_{\overline{B_{r_3}}(\mathbf{s})}$.*

*Proof.* First note that per Proposition 11, $H^{-1}|_{\overline{B_{r_3}}(\mathbf{s})}$ is indeed Lipschitz. Let $\widetilde{G} > 0$ be its Lipschitz coefficient. Next, by definition of the Dist measure (Equation (15)) we have that

$$Dist\big(H(A), \mathcal{W}_1\big) = \min_{\widetilde{A} \in \mathcal{W}_1} \|H(A) - \widetilde{A}\|_2 .$$

Since $A \in \overline{B_{r_2}}(\mathbf{s})$, it holds by Proposition 11 that $H(A) \in \overline{B_{r_3}}(\mathbf{s})$. Thus, since $\overline{B_{r_3}}(\mathbf{s})$ is a ball it must hold that

$$\min_{\widetilde{A} \in \mathcal{W}_1} \|H(A) - \widetilde{A}\|_2 = \min_{\widetilde{A} \in \mathcal{W}_1 \cap \overline{B_{r_3}}(\mathbf{s})} \|H(A) - \widetilde{A}\|_2 \,.$$

As $H$ is onto $\overline{B_{r_3}}(\mathbf{s})$, by Proposition 11 there exists $A' \in \overline{B_{r_1}}(\mathbf{s})$ such that

$$H(A') \in \operatorname*{argmin}_{\widetilde{A} \in \mathcal{W}_1 \cap \overline{B_{r_3}}(\mathbf{s})} \|H(A) - \widetilde{A}\|_2 \,.$$

Hence by the Lipschitz property of of $H^{-1}$ we obtain

$$\min_{\widetilde{A} \in \mathcal{W}_1 \cap \overline{B_{r_3}}(\mathbf{s})} \|H(A) - \widetilde{A}\|_2 = \|H(A) - H(A')\|_2$$

$$\geq \frac{1}{\widetilde{G}} \left\| H^{-1}(H(A)) - H^{-1}(H(A')) \right\|_2$$

$$= \frac{1}{\widetilde{G}} \|A - A'\|_2 \,.$$

Since $A' = H^{-1}(\widetilde{A})$ for some $\widetilde{A} \in \mathcal{W}_1 \cap \overline{B_{r_3}}(\mathbf{s})$, it holds by Lemma 15 that $A' \in \mathcal{W}_1$. Thus,

$$\frac{1}{\widetilde{G}} \|A - A'\|_2 \geq \frac{1}{\widetilde{G}} \min_{A'' \in \mathcal{W}_1} \|A - A''\|_2 = \frac{1}{\widetilde{G}} \cdot \operatorname{Dist}(A, \mathcal{W}_1) \,.$$

Combining the above inequalities and multiplying by $\widetilde{G}$ we obtain overall that

$$\widetilde{G} \cdot \operatorname{Dist}(H(A), \mathcal{W}_1) \geq |A - A'\|_2 \geq \operatorname{Dist}(A, \mathcal{W}_1)$$

as required. $\qquad \square$

**Lemma 17.** *Let $A \in \overline{B_{r_1}}(\mathbf{s})$. It holds that*

$$Dist(H(A), \mathcal{W}_1) \leq G \cdot Dist(A, \mathcal{W}_1)$$

*where $G$ is the Lipschitz coefficient of $H|_{\overline{B_{r_1}}(\mathbf{s})}$.*

*Proof.* First note that per Proposition 11, $H|_{\overline{B_{r_1}}(\mathbf{s})}$ is indeed Lipschitz. Let $G > 0$ be its Lipschitz coefficient . Next, by definition of the Dist measure we have that

$$\operatorname{Dist}(A, \mathcal{W}_1) = \min_{A' \in \mathcal{W}_1} \|A - A'\|_2 \,.$$

Since $A \in \overline{B_{r_1}}(\mathbf{s})$ and $\overline{B_{r_1}}(\mathbf{s})$ is a ball, it must hold that

$$\min_{A' \in \mathcal{W}_1} \|A - A'\|_2 = \min_{A' \in \mathcal{W}_1 \cap \overline{B_{r_1}}(\mathbf{s})} \|A - A'\|_2 \,.$$

By the Lipschitz property of $H$ we obtain

$$\min_{A' \in \mathcal{W}_1 \cap \overline{B_{r_1}}(\mathbf{s})} \|A - A'\|_2 \geq \min_{A' \in \mathcal{W}_1 \cap \overline{B_{r_1}}(\mathbf{s})} \frac{1}{G} \|H(A) - H(A')\|_2 \geq \operatorname{Dist}(H(A), \mathcal{W}_1) \,.$$

where the last inequality follows from Lemma 13. Multiplying by $G$ gives the result. $\qquad \square$

Before proving the main claims of this section, we introduce another condition on the initialization which we denote $\mathcal{I}_4$.

**Definition 11.** Let $\mu > 0$. We denote $G' := \max\{1, G, \widetilde{G}\}$ for $\widetilde{G}$ and $G$ from Lemmas 16 and 17. We use $\mathcal{I}_4(\mu)$ to denote the following subset of $\mathcal{I}_0$:

$$\mathcal{I}_4(\mu) := \left\{ A \in \mathcal{I}_0 : \alpha \leq \frac{r_3}{4 \max\{2, \exp(-2\lambda_-)\} \cdot G'^2 \sqrt{d} D_+(\mu)} \right\}$$

for $r_3$, $\lambda_-$ and $D_+$ from Proposition 11, Lemma 8, , and Corollary 4 respectively.

In the next two propositions, we bound the time it takes to escape the sphere around $\mathbf{s}$ under the linearized dynamics, and prove an additional claim that will be utilized later to show that the trajectory never returns to a certain sphere around $\mathbf{s}$.

We introduce notation which will be used in both propositions; Let $\mu > 0$. Suppose we initialize at $A(0) \in \mathcal{I}_3(\frac{\tilde{r}(\mu)}{4}) \cap \mathcal{I}_4(\mu)$ (for $\tilde{r}$ of Corollary 4) and at $A^{ref}(0)$, and evolve $A(t)$ and $A^{ref}(t)$ according to Equation (22). Suppose we initialize $\widetilde{A}(0) = H\left( A\big(t_1(\frac{\tilde{r}(\mu)}{4})\big) \right)$ and at $\widetilde{A^{ref}}(0) = H\left( A^{ref}\big(t_1(\frac{\tilde{r}(\mu)}{4})\big) \right)$ (for $t_1$ of Lemma 12), and evolve $\widetilde{A}(t)$ and $\widetilde{A^{ref}}(t)$ according to the linearized dynamics (see Lemma 10). For any time $t \geq 0$, denote the representations of $\widetilde{A}(t)$ and $\widetilde{A^{ref}}(t)$ with the orthogonal subspaces $\mathcal{W}_1$ and $\mathcal{W}_2$ to be

$$\widetilde{A}(t) = \widetilde{\beta_1}(t) \cdot \mathbf{1} + \widetilde{\beta_2}(t) \cdot \widetilde{\mathbf{v}}(t)$$

and

$$\widetilde{A^{ref}}(t) = \widetilde{\beta_1^{ref}}(t) \cdot \mathbf{1} + \widetilde{\beta_2^{ref}}(t) \cdot \widetilde{\mathbf{v}^{ref}}(t)$$

where $\widetilde{\mathbf{v}}(t), \widetilde{\mathbf{v}^{ref}}(t) \in \mathcal{W}_2$ are unit vectors.

The following proposition give quantitative bounds on the rate of exponential escape from $\mathbf{s}$ of trajectories under the lineaarized dynamics.

**Proposition 13.** *There exist times* $t_2(\mu), t_2^{ref}(\mu) \geq 2$ *for which it holds*

- $\left| \widetilde{\beta_2}\big(t_2(\mu)\big) \right|, \left| \widetilde{\beta_2^{ref}}\big(t_2(\mu)\big) \right| = \frac{r_3}{2\sqrt{d}}$.

- *For* $G' := \max\{1, G, \widetilde{G}\}$ *it holds that*

$$t_2(\mu), t_2^{ref}(\mu) \in \left[ -\frac{1}{\lambda_-} \ln\left( \frac{r_3}{4G'^2\sqrt{d}\alpha D_+(\mu)} \right), -\frac{1}{\lambda_-} \ln\left( \frac{G'^2 r_3}{2\sqrt{d}\alpha D_-(\mu)} \right) \right].$$

- *For any* $t \in [0, t_2(\mu)]$ *it holds that* $\widetilde{A}(t) \in \overline{B_{r_3}}(\mathbf{s})$.

- *For any* $t \in [0, t_2^{ref}(\mu)]$ *it holds that* $\widetilde{A^{ref}}(t) \in \overline{B_{r_3}}(\mathbf{s})$.

*Proof.* We prove the argument for $\widetilde{A}$ (the proof is identical for $\widetilde{A^{ref}}$). Recall Corollary 4 which states that

$$\text{Dist}\left( A\big(t_1(\frac{\tilde{r}(\mu)}{4})\big), \mathcal{W}_1 \right) = \left| \beta_2\big(t_1(\frac{\tilde{r}(\mu)}{4})\big) \right| \in [\alpha \cdot D_-(\mu), \alpha \cdot D_+(\mu)].$$

Thus, applying Lemmas 16 and 17 we obtain

$$\frac{\alpha \cdot D_-(\mu)}{\widetilde{G}} \leq \text{Dist}\big(\widetilde{A}(0), \mathcal{W}_1\big) = |\widetilde{\beta_2}(0)| \leq G \cdot \alpha \cdot D_+(\mu).$$

Per Lemma 10, for any $t \geq 0$ the solution at time $t$ to the linear dynamics initialized at $\widetilde{A}(0)$ is given by

$$\left( \exp(-t \cdot \lambda_+)\big(\widetilde{\beta_1}(0) - s\big) + s \right)\mathbf{1} + \big( \exp(-t \cdot \lambda_-) \cdot \widetilde{\beta_2}(0)\big)\widetilde{\mathbf{v}}(0).$$

As noted in Remark 2, the coefficient $|\widetilde{\beta_1}(t) - s|$ tends to zero as $t$ grows, while the coefficient $|\widetilde{\beta_2}(t)|$ tends to $\infty$ as $t$ grows. We first bound the time $t_2(\mu)$ for which $\left| \widetilde{\beta_2}\big(t_2(\mu)\big) \right| = \frac{r_3}{2\sqrt{d}}$. Since $A(0) \in \mathcal{I}_4(\mu)$, it holds that

$$\max\{2, \exp(-2\lambda_-)\} \leq \frac{r_3}{4G'^2\sqrt{d}\alpha D_+(\mu)}.$$

Therefore since $\lambda_- < 0$ (by Lemma 9) we obtain the following positive time $t_-$:

$$t_- := -\frac{1}{\lambda_-} \ln\left(\frac{r_3}{4G'^2\sqrt{d}\alpha D_+(\mu)}\right) \geq -\frac{1}{\lambda_-}\ln\left(\exp(-2\lambda_-)\right) = 2\,.$$

Thus, at time $t_-$ the solution to the linear dynamics satisfies the following:

$$\begin{aligned}
\mathrm{Dist}(\widetilde{A}(t_-), \mathcal{W}_1) &= |\widetilde{\beta_2}(t_-)| \\
&= |\widetilde{\beta_2}(0)| \cdot \exp(-t_- \cdot \lambda_-) \\
&\leq G \cdot D_+(\mu) \cdot \alpha \cdot \exp(-t_- \cdot \lambda_-) \\
&= G \cdot D_+(\mu) \cdot \alpha \cdot \exp\left(\frac{\lambda_-}{\lambda_-}\ln\left(\frac{r_3}{4G'^2\sqrt{d}\alpha D_+(\mu)}\right)\right) \\
&= \frac{G \cdot D_+(\mu) \cdot \alpha \cdot r_3}{4G'^2\sqrt{d}\alpha D_+(\mu)} \\
&\leq \frac{r_3}{4G'\sqrt{d}} \\
&\leq \frac{r_3}{4\sqrt{d}}\,,
\end{aligned}$$

where the last two inequalities stem from the fact that $G' \geq G, 1$. Hence, $t_-$ is a lower bound on $t_2(\mu)$. On the other hand, note that

$$\frac{G'^2 \cdot r_3}{2\sqrt{d}\alpha D_-(\mu)} \geq \frac{r_3}{4G'^2\sqrt{d}\alpha D_+(\mu)}$$

and so since $\lambda_- < 0$ we obtain the following positive time $t_+$:

$$t_+ := -\frac{1}{\lambda_-}\ln\left(\frac{G'^2 \cdot r_3}{2\sqrt{d}\alpha D_-(\mu)}\right) \geq -\frac{1}{\lambda_-}\ln\left(\frac{r_3}{4G'^2\sqrt{d}\alpha D_+(\mu)}\right) = t_-\,.$$

Thus, at time $t_+$ the solution to the linear dynamics statisfies the following:

$$\begin{aligned}
|\widetilde{\beta_2}(t_+)| = |\widetilde{\beta_2}(0)| \cdot \exp(-t_+ \cdot \lambda_-) \\
&\geq \frac{\alpha \cdot D_-(\mu)}{\widetilde{G}} \cdot \exp(-t_+ \cdot \lambda_-) \\
&= \frac{\alpha \cdot D_-(\mu)}{\widetilde{G}} \cdot \exp\left(\frac{\lambda_-}{\lambda_-}\ln\left(\frac{G'^2 \cdot r_3}{2\sqrt{d}\alpha D_-(\mu)}\right)\right) \\
&= \frac{\alpha \cdot D_-(\mu)}{\widetilde{G}} \cdot \frac{G'^2 \cdot r_3}{2\sqrt{d}\alpha D_-(\mu)} \\
&\geq \frac{G' \cdot r_3}{2\sqrt{d}} \\
&\geq \frac{r_3}{2\sqrt{d}}\,,
\end{aligned}$$

where the last two inequalities stem from the fact that $G' \geq \widetilde{G}, 1$. Hence, $t_+$ is an upper bound on $t_2(\mu)$. Next, we show that $|\widetilde{\beta_1}(t_-) - s| \leq \frac{r_3}{2\sqrt{d}}$. This will allow us to claim by monotonicity that $\widetilde{A}(t_2(\mu)) \in \overline{B_{r_3}}(\mathbf{s})$, since then we'll have the following:

$$\begin{aligned}
\left\|\widetilde{A}(t_2(\mu)) - \mathbf{s}\right\|_2 &= \left\|\widetilde{\beta_1}(t_2(\mu)) \cdot \mathbf{1} + \widetilde{\beta_2}(t_2(\mu)) \cdot \mathbf{v}(t_2(\mu)) - \mathbf{s}\right\|_2 \\
&= \left\|(\widetilde{\beta_1}(t_2(\mu)) - s) \cdot \mathbf{1}\right\|_2 + \left\|\widetilde{\beta_2}(t_2(\mu)) \cdot \mathbf{v}(t_2(\mu))\right\|_2 \\
&\leq \frac{r_3}{2\sqrt{d}} \cdot \sqrt{d} + \frac{r_3}{2\sqrt{d}} \cdot 1 \\
&\leq r_3\,.
\end{aligned}$$

Since $\widetilde{A}(0) \in \overline{B_{r_3}}(\mathbf{s})$ it holds that $|\widetilde{\beta_1}(0) - s| \le \frac{r_3}{\sqrt{d}}$. Hence, since $\frac{\lambda_+}{\lambda_-} < -1$ (by Lemma 9) it holds that

$$
\begin{aligned}
|\widetilde{\beta_1}(t_-) - s| &= |\exp(-\lambda_+ \cdot t_-)(\widetilde{\beta_1}(0) - s) + s - s| \\
&\le |\widetilde{\beta_1}(0) - s| \cdot \exp(-\lambda_+ \cdot t_-) \\
&\le \frac{r_3}{\sqrt{d}} \cdot \left( \frac{r_3}{4G'^2 \sqrt{d}\alpha D_+(\mu)} \right)^{\frac{\lambda_+}{\lambda_-}} \\
&\le \frac{r_3}{2\sqrt{d}},
\end{aligned}
$$

where the last inequality stems from the fact that $2 \le \frac{r_3}{4G'^2 \sqrt{d}\alpha D_+(\mu)}$. Finally, since under the linear dynamics $|\widetilde{\beta_2}(t)|$ monotonically grows and $|\widetilde{\beta_1}(t)|$ monotonically tends to $s$, it must hold that for any $t \in [0, t_2(\mu)]$ we have $\widetilde{A}(t) \in \overline{B_{r_3}}(\mathbf{s})$. $\qquad\square$

The next proposition shows that $\widetilde{\beta_2}(t)$ must be larger than some constant throughout a time interval of length $1$ before $t_2(\mu)$.

**Proposition 14.** *For any $\tau \in [0, 1]$ it holds that*

$$
\left|\widetilde{\beta_2}\big(t_2(\mu) - \tau\big)\right|, \left|\widetilde{\beta_2^{ref}}\big(t_2(\mu) - \tau\big)\right| = \frac{r_3}{2\sqrt{d}} \cdot \exp(\lambda_- \cdot \tau).
$$

*Proof.* We prove the argument for $\widetilde{A}$ (the proof is identical for $\widetilde{A^{ref}}$). In Proposition 13 we've established that $t_2(\mu) \ge 2$. Thus, per Lemma 10, for any $\tau \in [0, 1]$ the solution at the positive time $t_2(\mu) - \tau \ge 1$ to the linear dynamics initialized at $\widetilde{A}(0)$ is given by

$$
\left( \exp\big( -(t_2(\mu) - \tau) \cdot \lambda_+ \big)\big(\widetilde{\beta_1}(0) - s\big) + s \right)\mathbf{1} + \exp\big( -(t_2(\mu) - \tau) \cdot \lambda_- \big) \cdot \widetilde{\beta_2}(0)\widetilde{\mathbf{v}}(0).
$$

Hence, since $\left|\widetilde{\beta_2}\big(t_2(\mu)\big)\right| = \frac{r_3}{2\sqrt{d}}$ we obtain that

$$
\begin{aligned}
\left|\widetilde{\beta_2}\big(t_2(\mu) - \tau\big)\right| &= |\exp\big( -(t_2(\mu) - \tau) \cdot \lambda_- \big) \cdot \widetilde{\beta_2}(0)| \\
&= |\exp\big( -t_2(\mu) \cdot \lambda_- \big) \cdot \widetilde{\beta_2}(0)| \cdot \exp(\lambda_- \cdot \tau) \\
&= \left|\widetilde{\beta_2}\big(t_2(\mu)\big)\right| \cdot \exp(\lambda_- \cdot \tau) \\
&= \frac{r_3}{2\sqrt{d}} \cdot \exp(\lambda_- \cdot \tau).
\end{aligned}
$$

$\qquad\square$

The above established that there exists a time which is $\mathcal{O}(\ln(\frac{1}{\alpha}))$ where at least one of the linearized trajectories is at a constant distant from $\mathcal{W}_1$. We complete this section by proving the following corollary, which states that the corresponding non linear dynamics trajectory must also be at a constant distance from $\mathcal{W}_1$ during a time interval of length $1$. This will eventually allow us to claim that the trajectory must remain trapped within a set where the objective $\ell$ satistfies satisfies the PL condition (see Definition 13), which in turn ensures a rapid convergence to a global minimum (discussed in the next section).

**Corollary 5.** *Let $\mu > 0$. Suppose we initialize at $A(0) \in \mathcal{I}_3(\frac{\widetilde{r}(\mu)}{4}) \cap \mathcal{I}_4(\mu)$ (for $\widetilde{r}$ of Corollary 4) and at $A^{ref}(0)$, and evolve $A(t)$ and $A^{ref}(t)$ according to Equation (22). For any $\tau \in [0, 1]$ it holds that*

$$
\left|\beta_2\big(t_1(\tfrac{\widetilde{r}(\mu)}{4}) + t_2(\mu) - \tau\big)\right| = Dist\left( A\big(t_1(\tfrac{\widetilde{r}(\mu)}{4}) + t_2(\mu) - \tau\big), \mathcal{W}_1 \right) \ge \frac{r_3 \cdot \exp(\lambda_-)}{2G \cdot \sqrt{d}}
$$

*and*

$$
\left|\beta_2^{ref}\big(t_1(\tfrac{\widetilde{r}(\mu)}{4}) + t_2^{ref}(\mu) - \tau\big)\right| = Dist\left( A^{ref}\big(t_1(\tfrac{\widetilde{r}(\mu)}{4}) + t_2^{ref}(\mu) - \tau\big), \mathcal{W}_1 \right) \ge \frac{r_3 \cdot \exp(\lambda_-)}{2G \cdot \sqrt{d}}.
$$

*Proof.* We prove the argument for $A$ (the proof is identical for $A^{ref}$). In Proposition 14 we have shown that for any $\tau \in [0,1]$ it holds that $\widetilde{A}(t_2(\mu) - \tau) \in \overline{B_{r_3}}(\mathbf{s})$, and so the mapping $H$ is a conjugation to the original dynamics. Therefore since we've initialized $\widetilde{A}(0)$ at $H\left(A(t_1(\frac{\widetilde{r}(\mu)}{4}))\right)$, it must hold that for any $\tau \in [0,1]$

$$H^{-1}\left(\widetilde{A}(t_2(\mu) - \tau)\right) = A\left(t_1(\frac{\widetilde{r}(\mu)}{4}) + t_2(\mu) - \tau\right).$$

Since $H^{-1}[\overline{B_{r_3}}] \subseteq \overline{B_{r_1}}$ we obtain that $A\left(t_1(\frac{\widetilde{r}(\mu)}{4}) + t_2(\mu) - \tau\right) \in \overline{B_{r_1}}$, and thus by Lemma 17 we obtain

$$\text{Dist}\left(A\left(t_1(\frac{\widetilde{r}(\mu)}{4}) + t_2(\mu) - \tau\right), \mathcal{W}_1\right) \geq \frac{\text{Dist}\left(\widetilde{A}(t_2(\mu) - \tau), \mathcal{W}_1\right)}{G}.$$

Note that by orthogonoality $\left|\widetilde{\beta_2}(t_2(\mu) - \tau)\right| = \text{Dist}\left(\widetilde{A}(t_2(\mu) - \tau), \mathcal{W}_1\right)$, hence plugging Proposition 14 we receive

$$\text{Dist}\left(A\left(t_1(\frac{\widetilde{r}(\mu)}{4}) + t_2(\mu) - \tau\right), \mathcal{W}_1\right) \geq \frac{\left|\widetilde{\beta_2}(t_2(\mu) - \tau)\right|}{G} = \frac{r_3 \cdot \exp(\lambda_- \cdot \tau)}{2G \cdot \sqrt{d}} \geq \frac{r_3 \cdot \exp(\lambda_-)}{2G \cdot \sqrt{d}}$$

where the last inequality is due to $\lambda_- < 0$. The proof is complete by observing that

$$\text{Dist}\left(A\left(t_1(\frac{\widetilde{r}(\mu)}{4}) + t_2(\mu) - \tau\right), \mathcal{W}_1\right) = \left|\beta_2\left(t_1(\frac{\widetilde{r}(\mu)}{4}) + t_2(\mu) - \tau\right)\right|.$$

$\square$

### F.3.6  Convergence to a Global Minimum

We begin this section by proving the following corollary regarding the difference between different coordinates of the points reached by the gradient flow trajectories. We will later show that this ensures the objective satisfies the PL condition (Definition 13).

**Corollary 6.** *Let $\mu > 0$. Suppose we initialize at $A(0) \in \mathcal{I}_3(\frac{\widetilde{r}(\mu)}{4}) \cap \mathcal{I}_4(\mu)$ (for $\widetilde{r}$ of Corollary 4) and at $A^{ref}(0)$, and evolve $A(t)$ and $A^{ref}(t)$ according to Equation (22). For any $\tau \in [0,1]$ it holds that there exist $i, j, i^{ref}, j^{ref} \in [d]$ such that*

$$\left|a_i\left(t_1(\frac{\widetilde{r}(\mu)}{4}) + t_2(\mu) - \tau\right) - a_j\left(t_1(\frac{\widetilde{r}(\mu)}{4}) + t_2(\mu) - \tau\right)\right| \geq \frac{r_3 \cdot \exp(\lambda_-)}{2G \cdot d^{1.5}}$$

*and*

$$\left|a^{ref}_{i^{ref}}\left(t_1(\frac{\widetilde{r}(\mu)}{4}) + t^{ref}_2(\mu) - \tau\right) - a^{ref}_{j^{ref}}\left(t_1(\frac{\widetilde{r}(\mu)}{4}) + t^{ref}_2(\mu) - \tau\right)\right| \geq \frac{r_3 \cdot \exp(\lambda_-)}{2G \cdot d^{1.5}}.$$

*Proof.* The claim follows from invoking Lemma 31 and plugging the lower bound on $|\beta_2|$ and $|\beta^{ref}_2|$ provided in Corollary 5. $\square$

We continue by proving that $\ell$ satisfies the PL condition (Definition 13) on the subset of points $\text{Diff}(b)^{\mathcal{C}}$ (Equation (18)):

**Lemma 18.** *Let $b > 0$. The function $\ell|_{Diff(b)^c \cap [-3,3]^d}$ satisfies the PL condition with PL coefficient*

$$\mu = \frac{1}{2} \min\left\{2d, \frac{\left((L-1)\widetilde{b}\right)^2}{4}, (\frac{\widehat{b}}{\widehat{b}+2})^2\right\},$$

*for $\widetilde{b} := (\frac{b}{2})^{L-2}$ and $\widehat{b} := (\frac{b}{6})^{L-2}$.*

*Proof.* Let $A \in \mathrm{Diff}(b)^{\mathcal{C}} \cap [-3, 3]^d$. Recalling the definition of $\mathrm{Diff}(b)^{\mathcal{C}}$ (Equation (18)) there exist $i, j \in [d]$ such that $|a_i - a_j| \geq b$. Since $a_i \neq a_j$, at least one is non-zero. Assume WLOG that $a_j \neq 0$. Hence, since $0 < L - 2 \in \mathbb{N}_{odd}$ we get per proposition 20 that

$$|a_i^{L-2} - a_j^{L-2}| \geq (\frac{b}{2})^{L-2} =: \widetilde{b}$$

and

$$|1 - \frac{a_i^{L-2}}{a_j^{L-2}}| \geq (\frac{b}{6})^{L-2} =: \widehat{b}.$$

Denote $res_s := 1 - \sum_{k=1}^d a_k$ and $res_l := 1 - \sum_{k=1}^d a_k^{L-1}$. If $res_s = 0$, then it holds that

$$\|\nabla \ell(A)\|^2 = \sum_{k=1}^d (L-1)^2 (a_k^{L-2})^2 res_l^2 = (*).$$

By the triangle inequality, either $|a_i^{L-2}| \geq \frac{\widetilde{b}}{2}$ or $|a_j^{L-2}| \geq \frac{\widetilde{b}}{2}$. In either case,

$$(*) \geq (L-1)^2 \frac{\widetilde{b}^2}{4} res_l^2 = \frac{((L-1)\widetilde{b})^2}{4} (\frac{1}{2} res_l^2 + \frac{1}{2} res_s^2) = \frac{((L-1)\widetilde{b})^2}{4} \ell(A),$$

*i.e.* the PL condition is satisfied with $\mu = \frac{1}{2} \cdot \frac{((L-1)\widetilde{b})^2}{4}$. If $res_l = 0$, then it holds that

$$\|\nabla \ell(A)\|^2 = \sum_{k=1}^d res_s^2 = d\,res_s^2 = 2d(\frac{1}{2} res_l^2 + \frac{1}{2} res_s^2) = 2d\ell(A),$$

*i.e.* the PL condition is satisfied with $\mu = \frac{1}{2} \cdot 2d$. Assume $res_l, res_s \neq 0$ and denote $\chi := \frac{-res_s}{(L-1)res_l} \neq 0$. For any $k \in [d]$ we have

$$\nabla \ell(A)_k = (L-1)a_k^{L-2} res_l + res_s = (L-1)a_k^{L-2} res_l - (L-1)res_l \cdot \chi = (L-1)(a_k^{L-2} - \chi)res_l$$

or equivalently

$$\nabla \ell(A)_k = (L-1)a_k^{L-2} res_l + res_s = -(L-1)a_k^{L-2} \frac{res_s}{(L-1)\chi} + res_s = (1 - \frac{a_k^{L-2}}{\chi})res_s$$

Squaring the above identities we obtain

$$\nabla \ell(A)_k^2 = (L-1)^2 (a_k^{L-2} - \chi)^2 res_l^2 = (1 - \frac{a_k^{L-2}}{\chi})^2 res_s^2.$$

By the triangle inequality we have that either

$$|a_i^{L-2} - \chi| \geq \frac{\widetilde{b}}{2}$$

or

$$|a_j^{L-2} - \chi| \geq \frac{\widetilde{b}}{2}.$$

Therefore, if $res_l^2 \geq res_s^2$ we get that

$$\|\nabla \ell(A)\|^2 = \sum_{k=1}^d (L-1)^2 (a_k^{L-2} - \chi)^2 res_l^2$$

$$\geq \frac{((L-1)\widetilde{b})^2}{4} res_l^2$$

$$\geq \frac{((L-1)\widetilde{b})^2}{4} (\frac{1}{2} res_l^2 + \frac{1}{2} res_s^2)$$

$$= \frac{((L-1)\widetilde{b})^2}{4} \ell(A),$$

*i.e.* the PL condition is satisfied with $\mu = \frac{1}{2}\frac{((L-1)\widetilde{b})^2}{4}$. On the other hand if $res_l^2 < res_s^2$, then by proposition 21 either

$$|1 - \frac{a_i^{L-2}}{\chi}| \geq \frac{\widehat{b}}{\widehat{b}+2}$$

or

$$|1 - \frac{a_j^{L-2}}{\chi}| \geq \frac{\widehat{b}}{\widehat{b}+2}$$

and therefore

$$\|\nabla\ell(A)\|^2 = \sum_{k=1}^{d}(1 - \frac{a_k^{L-2}}{\chi})^2 res_s^2 \geq (\frac{\widehat{b}}{\widehat{b}+2})^2 res_s^2 \geq (\frac{\widehat{b}}{\widehat{b}+2})^2(\frac{1}{2}res_l^2 + \frac{1}{2}res_s^2) = (\frac{\widehat{b}}{\widehat{b}+2})^2\ell(A),$$

*i.e.* the PL condition is satisfied with $\mu = \frac{1}{2}(\frac{\widehat{b}}{\widehat{b}+2})^2$. Overall, we get that whenever $A \in \text{Diff}(b)^\mathcal{C} \cap [-3,3]^d$, the PL condition is satisfied with $\mu = \frac{1}{2}\min\left\{2d, \frac{((L-1)\widetilde{b})^2}{4}, (\frac{\widehat{b}}{\widehat{b}+2})^2\right\}$, as required. $\qquad\square$

The above results in the following corollary regarding the PL condition satisfied by $\ell$ at a certain set of points reached by the gradient flow trajectories.

**Corollary 7.** *Let $\mu > 0$. Suppose we initialize at $A(0) \in \mathcal{I}_3(\frac{\widetilde{r}(\mu)}{4}) \cap \mathcal{I}_4(\mu)$ (for $\widetilde{r}$ of Corollary 4) and at $A^{ref}(0)$, and evolve $A(t)$ and $A^{ref}(t)$ according to Equation (22). For any $\tau \in [0,1]$ it holds that $\ell$ satisfies the PL condition (Definition 13) at the points*

$$A\big(t_1(\frac{\widetilde{r}(\mu)}{4}) + t_2(\mu) - \tau\big)$$

*and*

$$A^{ref}\big(t_1(\frac{\widetilde{r}(\mu)}{4}) + t_2^{ref}(\mu) - \tau\big)$$

*with PL coefficient*

$$\mu_1 := \frac{1}{2}\min\left\{2d, \frac{((L-1)\widetilde{b})^2}{4}, (\frac{\widehat{b}}{\widehat{b}+2})^2\right\},$$

*for*

$$\widetilde{b} := \left(\frac{r_3 \cdot \exp(\lambda_-)}{4G \cdot d^{1.5}}\right)^{L-2} \quad , \quad \widehat{b} := \left(\frac{r_3 \cdot \exp(\lambda_-)}{12G \cdot d^{1.5}}\right)^{L-2}$$

*Proof.* The claim follows from invoking Lemma 18 and plugging the bound on the coordinate difference provided in Corollary 6. $\qquad\square$

For the rest of the proof, we let $\mu = \mu_1$ from Corollary 7. We are now ready to prove the following proposition, which states that at time $t_1(\frac{\widetilde{r}(\mu_1)}{4}) + t_2(\mu_1)$, the trajectory is at a point whose value improves upon the value of $\ell$ at $\mathbf{s}$ by a constant:

**Proposition 15.** *Consider $\mu_1$ from Corollary 7. Suppose we initialize at $A(0) \in \mathcal{I}_3(\frac{\widetilde{r}(\mu_1)}{4}) \cap \mathcal{I}_4(\mu_1)$ (for $\widetilde{r}$ of Corollary 4) and at $A^{ref}(0)$, and evolve $A(t)$ and $A^{ref}(t)$ according to Equation (22). it holds that*

$$\ell(\mathbf{s}) - \ell\bigg(A\big(t_1(\frac{\widetilde{r}(\mu_1)}{4}) + t_2(\mu_1)\big)\bigg) \geq \min\left\{\frac{\ell(\mathbf{s})}{2}, \frac{3\mu_1 \cdot \ell(\mathbf{s})}{4}\right\}$$

*and*

$$\ell(\mathbf{s}) - \ell\bigg(A^{ref}\big(t_1(\frac{\widetilde{r}(\mu_1)}{4}) + t_2^{ref}(\mu_1)\big)\bigg) \geq \min\left\{\frac{\ell(\mathbf{s})}{2}, \frac{3\mu_1 \cdot \ell(\mathbf{s})}{4}\right\}.$$

*Proof.* We prove the argument for $A$ (the proof is identical for $A^{ref}$). First, suppose that $\ell\left(A\big(t_1(\frac{\widetilde{r}(\mu_1)}{4}) + t_2(\mu_1)\big)\right) \leq \frac{\ell(\mathbf{s})}{2}$. Then it holds that

$$\ell(\mathbf{s}) - \ell\left(A\big(t_1(\frac{\widetilde{r}(\mu_1)}{4}) + t_2(\mu_1)\big)\right) \geq \ell(\mathbf{s}) - \frac{\ell(\mathbf{s})}{2} = \frac{\ell(\mathbf{s})}{2} \geq \min\left\{\frac{\ell(\mathbf{s})}{2}, \frac{3\mu_1 \cdot \ell(\mathbf{s})}{4}\right\}.$$

Next, suppose that $\ell\left(A\big(t_1(\frac{\widetilde{r}(\mu_1)}{4}) + t_2(\mu_1)\big)\right) > \frac{\ell(\mathbf{s})}{2}$. Thus per Lemma 22, for any $\tau \in [0,1]$ it holds that

$$\ell\left(A\big(t_1(\frac{\widetilde{r}(\mu_1)}{4}) + t_2(\mu_1) - \tau\big)\right) \geq \ell\left(A\big(t_1(\frac{\widetilde{r}(\mu_1)}{4}) + t_2(\mu_1)\big)\right) > \frac{\ell(\mathbf{s})}{2}.$$

Next, per Corollary 7 for any $\tau \in [0,1]$ it also holds that $\ell$ satisfies the PL condition in the point $A\big(t_1(\frac{\widetilde{r}(\mu_1)}{4}) + t_2(\mu_1) - \tau\big)$ with PL coefficient $\mu_1$. Thus, by Lemma 32 it holds that the difference

$$\ell\left(A\big(t_1(\frac{\widetilde{r}(\mu_1)}{4}) + t_2(\mu_1) - 1\big)\right) - \ell\left(A\big(t_1(\frac{\widetilde{r}(\mu_1)}{4}) + t_2(\mu_1)\big)\right)$$

is lower bound by

$$2\big(t_1(\frac{\widetilde{r}(\mu_1)}{4}) + t_2(\mu_1) - t_1(\frac{\widetilde{r}(\mu_1)}{4}) - t_2(\mu_1) + 1\big) \cdot \mu_1 \cdot \frac{\ell(\mathbf{s})}{2} = \mu_1 \cdot \ell(\mathbf{s}).$$

On the other hand, recall that by Corollary 4 and since gradient flow is non-increasing we have that

$$\ell\left(A\big(t_1(\frac{\widetilde{r}(\mu_1)}{4}) + t_2(\mu_1) - 1\big)\right) \leq \ell\left(A\big(t_1(\frac{\widetilde{r}(\mu_1)}{4})\big)\right) \leq (1 + \frac{\mu_1}{4})\ell(\mathbf{s}).$$

Thus, we obtain the following:

$$(1 + \frac{\mu_1}{4})\ell(\mathbf{s}) - \ell\left(A\big(t_1(\frac{\widetilde{r}(\mu_1)}{4}) + t_2(\mu_1)\big)\right) \geq \mu_1 \cdot \ell(\mathbf{s}).$$

Reorganizing thus yields

$$\ell(\mathbf{s}) - \ell\left(A\big(t_1(\frac{\widetilde{r}(\mu_1)}{4}) + t_2(\mu_1)\big)\right) \geq \frac{3\mu_1 \cdot \ell(\mathbf{s})}{4}$$

as required. $\qquad\square$

We continue to prove the following proposition, which states that after time $t_1(\frac{\widetilde{r}(\mu_1)}{4}) + t_2(\mu_1)$, the points reached by the gradient flow trajectory are ones where $\ell$ satisfies the PL condition with a certain PL coefficient.

**Proposition 16.** *Consider $\mu_1$ from Corollary 7. Suppose we initialize at $A(0) \in \mathcal{I}_3(\frac{\widetilde{r}(\mu_1)}{4}) \cap \mathcal{I}_4(\mu_1)$ (for $\widetilde{r}$ of Corollary 4) and at $A^{ref}(0)$, and evolve $A(t)$ and $A^{ref}(t)$ according to Equation (22). There exists $\nu > 0$ such that for any time $t$ it holds that:*

- *If $t \geq t_1(\frac{\widetilde{r}(\mu_1)}{4}) + t_2(\mu_1)$ then $A(t) \in Diff(\nu)^{\mathcal{C}}$.*

- *If $t \geq t_1(\frac{\widetilde{r}(\mu_1)}{4}) + t_2^{ref}(\mu_1)$ then $A^{ref}(t) \in Diff(\nu)^{\mathcal{C}}$.*

*Additionally, it holds that $\ell$ satistfies the PL condition at $Diff(\nu)^{\mathcal{C}}$ with coefficient $\mu_2$ for*

$$\mu_2 := \frac{1}{2}\min\left\{2d, \frac{\big((L-1)\widetilde{\nu}\big)^2}{4}, \big(\frac{\widehat{\nu}}{\widehat{\nu}+2}\big)^2\right\}$$

*where*

$$\widetilde{\nu} := \big(\frac{\nu}{2}\big)^{L-2} \quad , \quad \widehat{\nu} := \big(\frac{\nu}{6}\big)^{L-2}$$

*Proof.* We prove the argument for $A$ (the proof is identical for $A^{ref}$). Let $t \geq t_1(\frac{\widetilde{r}(\mu_1)}{4}) + t_2^{ref}(\mu_1)$. Denote for any $\psi \geq 0$ the function

$$f(\psi) := \min_{A \in \mathrm{Diff}(\psi) \cap [-3,3]^d} \ell(A)$$

for $\mathrm{Diff}(\psi)$ defined in Equation (17). Per Lemma 7 and since $\mathbf{s} \in [-3,3]^d$ it holds that

$$\ell(\mathbf{s}) = \min_{A \in \mathcal{W}_1} \ell(A) = \min_{A \in \mathcal{W}_1 \cap [-3,3]^d} \ell(A) = \min_{A \in \mathrm{Diff}(0) \cap [-3,3]^d} \ell(A) = f(0) \,.$$

Invoking Lemma 33, it holds that $f$ is right side continuous in $0$. Hence by continuity, we obtain that there exists $\nu$ such that for any $\psi \in [0, \nu]$ it holds that

$$f(\psi) \geq f(0) - \frac{1}{2} \min\left\{ \frac{\ell(\mathbf{s})}{2}, \frac{3\mu_1 \cdot \ell(\mathbf{s})}{4} \right\}$$

$$= \ell(\mathbf{s}) - \frac{1}{2} \min\left\{ \frac{\ell(\mathbf{s})}{2}, \frac{3\mu_1 \cdot \ell(\mathbf{s})}{4} \right\}$$

$$> \ell(\mathbf{s}) - \min\left\{ \frac{\ell(\mathbf{s})}{2}, \frac{3\mu_1 \cdot \ell(\mathbf{s})}{4} \right\} \,.$$

On the other hand, per Proposition 15 and since the objective is non-increasing under gradient flow (see Lemma 22), we obtain the following:

$$\ell(\mathbf{s}) - \ell\big(A(t)\big) \geq \ell(\mathbf{s}) - \ell\left( A\big(t_1(\frac{\widetilde{r}(\mu_1)}{4}) + t_2(\mu_1)\big) \right) \geq \min\left\{ \frac{\ell(\mathbf{s})}{2}, \frac{3\mu_1 \cdot \ell(\mathbf{s})}{4} \right\} \,.$$

Rearranging we thus obtain

$$f(\psi) > \ell(\mathbf{s}) - \min\left\{ \frac{\ell(\mathbf{s})}{2}, \frac{3\mu_1 \cdot \ell(\mathbf{s})}{4} \right\} \geq \ell\big(A(t)\big) \,.$$

Note that per Lemma 33, $f$ is non increasing w.r.t $\psi$, thus it must hold that $A(t) \notin \mathrm{Diff}(\psi)$ (since it is in $[-3,3]^d$). As this holds for any $\psi \in [0,\nu]$, we obtain that $A(t) \in \mathrm{Diff}(\nu)^{\mathcal{C}}$. Hence, the argument follows from Lemma 18. $\qquad\square$

The final proposition of this section proves that the gradient flow trajectory converges to a global minimum $\widehat{A_2}$.

**Proposition 17.** *Consider $\mu_1$ from Corollary 7. Suppose we initialize at $A(0) \in \mathcal{I}_3(\frac{\widetilde{r}(\mu_1)}{4}) \cap \mathcal{I}_4(\mu_1)$ (for $\widetilde{r}$ of Corollary 4) and evolve $A(t)$ according to Equation (22). There exists $\widehat{A_2} \in \mathbb{R}^d$ such that*

$$\lim_{t \to \infty} A(t) = \widehat{A_2} \quad \text{and} \quad \ell(\widehat{A_2}) = 0 \,.$$

*Proof.* Per Corollary 7, there exists $\nu > 0$ such that for any $t \geq t_1(\frac{\widetilde{r}(\mu_1)}{4}) + t_2(\mu_1)$ it holds that $A(t) \in \mathrm{Diff}(\nu)^{\mathcal{C}}$ where $\ell$ satisfies the PL condition with coefficient $\mu_2$ (defined in Corollary 7). Next, note that per Lemma 11 and Corollary 3, $A(t)$ is always contained in $[-3,3]^d$, where $\ell$'s gradient is $N$ Lipschitz. Therefore, the claim follows from Lemma 25. $\qquad\square$

### F.3.7 Overall Divergence From Reference Trajecory

In this section we show that one can choose a set of initializations such that the divergence between $\widehat{A_2}$ and some point on the reference trajectory is arbitrarily small. This shows that $\widehat{A_2}$ must not recover the teacher (per Lemma 3). We begin by proving the following lemma which gives explicit times at which the gradient flow trajectories reach points of arbitrary small value.

**Lemma 19.** *Let $\eta_2 \in (0,1)$. Consider $\mu_1$ from Corollary 7. Suppose we initialize at $A(0) \in \mathcal{I}_3(\frac{\widetilde{r}(\mu_1)}{4}) \cap \mathcal{I}_4(\mu_1)$ (for $\widetilde{r}$ of Corollary 4) and evolve $A(t)$ according to Equation (22). Denote $t_2^*(\mu_1) := \min\left\{ t_2(\mu_1), t_2^{ref}(\mu_1) \right\}$. Denote the time $t_3(\eta_2) \geq 0$ to be*

$$t_3(\eta_2) := -\frac{\ln(\eta_2)}{2\mu_2} - \frac{1}{\lambda_-} \ln\left( \frac{G'^2 r_3}{2\sqrt{d}\alpha D_-(\mu_1)} \right) + \frac{1}{\lambda_-} \ln\left( \frac{r_3}{4G'^2 \sqrt{d}\alpha D_+(\mu_1)} \right) \,.$$

*It holds that*

$$\ell\left(A\big(t_1(\frac{\widetilde{r}(\mu_1)}{4}) + t_2^*(\mu_1) + t_3(\eta_2)\big)\right) \le \eta_2.$$

*Proof.* First, note that since $\lambda_- < 0 < G'$ and $0 < D_-(\mu_1) < D_+(\mu_1)$ we obtain that

$$t_3(\eta_2) = -\frac{\ln(\eta_2)}{2\mu_2} + \frac{1}{\lambda_-}\ln\left(\frac{D_-(\mu_1)}{2G'^4 D_+(\mu_1)}\right) \ge -\frac{\ln(\eta_2)}{2\mu_2}.$$

The right term is positive as $0 < \eta_2 < 1$ and $\mu_2 > 0$. Next per Proposition 13 it holds that

$$|t_2(\mu_1) - t_2^{ref}(\mu_1)| \le -\frac{1}{\lambda_-}\ln\left(\frac{G''^2 r_3}{2\sqrt{d}\alpha D_-(\mu)}\right) + \frac{1}{\lambda_-}\ln\left(\frac{r_3}{4G'^2\sqrt{d}\alpha D_+(\mu)}\right)$$

$$= \frac{1}{\lambda_-}\ln\left(\frac{D_-(\mu_1)}{2G'^4 D_+(\mu_1)}\right)$$

which results in

$$t_1(\frac{\widetilde{r}(\mu_1)}{4}) + t_2^*(\mu_1) + t_3(\eta_2) \ge t_1(\frac{\widetilde{r}(\mu_1)}{4}) + t_2(\mu_1) - \frac{\ln(\eta_2)}{2\mu_2}.$$

Hence, per Lemma 22 we obtain that

$$\ell\left(A\big(t_1(\frac{\widetilde{r}(\mu_1)}{4}) + t_2^*(\mu_1) + t_3(\eta_2)\big)\right) \le \ell\left(A\big(t_1(\frac{\widetilde{r}(\mu_1)}{4}) + t_2(\mu_1) - \frac{\ln(\eta_2)}{2\mu_2}\big)\right).$$

Per Proposition 16 there exists $\nu > 0$ such that for any $t \ge t_1(\frac{\widetilde{r}(\mu_1)}{4}) + t_2(\mu_1)$ it holds that $A(t) \in \text{Diff}(\nu)^{\mathcal{C}}$ where $\ell$ satisfies the PL condition with coefficient $\mu_2$ (defined in Corollary 7). Thus, per Lemma 24 it holds that

$$\ell\left(A\big(t_1(\frac{\widetilde{r}(\mu_1)}{4}) + t_2(\mu_1) - \frac{\ln(\eta_2)}{2\mu_2}\big)\right) \le \ell\left(A\big(t_1(\frac{\widetilde{r}(\mu_1)}{4}) + t_2(\mu_1)\big)\right) \cdot \exp(2\mu_2 \cdot \frac{\ln(\eta_2)}{2\mu_2})$$

$$\le \ell(A(0)) \cdot \eta_2$$

where the second to last inequality stems from Lemma 22. The proof follows by noting that at initialization $\ell$'s value is no more than 1. $\square$

We continue to the following lemma which proves that there exist times at which the distance between the gradient flow trajectory and $\widehat{A_2}$ (defined in Proposition 17) is arbitrarily small.

**Lemma 20** (Distance between gradient flow trajectory and $\widehat{A_2}$). *Let $\delta \in (0,1)$. Consider $\mu_1$ from Corollary 7. Suppose we initialize at $A(0) \in \mathcal{I}_3(\frac{\widetilde{r}(\mu_1)}{4}) \cap \mathcal{I}_4(\mu_1)$ (for $\widetilde{r}$ of Corollary 4). Denote $t_2^*(\mu_1) := \min\{t_2(\mu_1), t_2^{ref}(\mu_1)\}$. Then there exists $\eta_{2,\delta} > 0$ such that*

$$\left\|\widehat{A_2} - A\big(t_1(\frac{\widetilde{r}(\mu_1)}{4}) + t_2^*(\mu_1) + t_3(\eta_{2,\delta})\big)\right\|_2 \le \delta.$$

*Proof.* Denote $\mathcal{A}^* := \left\{A \in \mathbb{R}^d : \ell(A) = 0\right\}$. Per Corollary 7, there exists $\mu_2 > 0$ such that for any $t \ge t_1(\frac{\widetilde{r}(\mu_1)}{4}) + t_2(\mu_1)$ it holds that $\ell$ satisfies the PL condition in $A(t)$ with PL coefficient $\mu_2$. Next, note that per Lemma 11 and Corollary 3 it holds that $\ell$'s gradient is $N$ Lipschitz in $A(t)$. Therefore by Lemma 25 for any $t \ge 0$ it holds that

$$\left\|\widehat{A_2} - A\big(t_1(\frac{\widetilde{r}(\mu_1)}{4}) + t_2(\mu_1) + t\big)\right\|_2 \le \sqrt{\frac{N}{\mu_2}}\text{Dist}\left(A\big(t_1(\frac{\widetilde{r}(\mu_1)}{4}) + t_2(\mu_1) + t\big), \mathcal{A}^*\right).$$

Additionally, note that since $\ell$ is continuous and non-negative, when considering its restriction to $[-3,3]^d$ we obtain that its 1 sub-level set is compact. Therefore we obtain by Lemma 35 that there

exists $\eta_{2,\delta} \in (0,1)$ such that for any $A \in [-3,3]^d$ if $\ell(A) \leq \eta_{2,\delta}$ then $\text{Dist}(A, \mathcal{A}^*) \leq \sqrt{\frac{\mu_2}{N}}\delta$. It was shown in Lemma 19 that

$$t_1(\frac{\widetilde{r}(\mu_1)}{4}) + t_2^*(\mu_1) + t_3(\eta_{2,\delta}) \geq t_1(\frac{\widetilde{r}(\mu_1)}{4}) + t_2(\mu_1)$$

and so we obtain that

$$\left\| \widehat{A_2} - A\big(t_1(\frac{\widetilde{r}(\mu_1)}{4}) + t_2^*(\mu_1) + t_3(\eta_{2,\delta})\big) \right\|_2$$
$$\leq \sqrt{\frac{N}{\mu_2}} \text{Dist}\left( A\big(t_1(\frac{\widetilde{r}(\mu_1)}{4}) + t_2^*(\mu_1) + t_3(\eta_{2,\delta})\big), \mathcal{A}^* \right).$$

It was also shown that

$$\ell\left( A\big(t_1(\frac{\widetilde{r}(\mu_1)}{4}) + t_2^*(\mu_1) + t_3(\eta_{2,\delta})\big) \right) \leq \eta_{2,\delta}$$

and so we obtain

$$\left\| \widehat{A_2} - A\big(t_1(\frac{\widetilde{r}(\mu_1)}{4}) + t_2^*(\mu_1) + t_3(\eta_{2,\delta})\big) \right\|_2 \leq \sqrt{\frac{N}{\mu_2}\frac{\mu_2}{N}}\delta = \delta.$$

as required. $\qquad\square$

Before proving the last proposition of this section, we introduce another condition on the initialization which we denote $\mathcal{I}_5$.

**Definition 12.** Let $\delta, \eta > 0$. We use $\mathcal{I}_5(\delta, \eta)$ to denote the following subset of $\mathcal{I}_0$:

$$\mathcal{I}_5(\delta, \eta) := \left\{ A \in \mathcal{I}_0 : (1 - \zeta_2) \leq \frac{\delta}{\left( \frac{G'^2 r_3}{2\sqrt{d}D_-(\mu_1)} \right)^2 \exp\left( N \cdot \big(t_1(\frac{\widetilde{r}(\mu_1)}{4}) + t_3(\eta)\big) \right)} \cdot \alpha \right\}$$

for $r_3$, $D_-$, $G'$ and $\mu_1$ from Proposition 11, Corollaries 4 and 7, , and Definition 11 respectively.

We proceed to proving the following proposition which upper bounds the divergence between $A(t)$ and $A^{ref}(t)$.

**Proposition 18.** Let $\delta > 0$. Consider $\mu_1$ from Corollary 7. Suppose we initialize at $A(0) \in \mathcal{I}_3(\frac{\widetilde{r}(\mu_1)}{4}) \cap \mathcal{I}_4(\mu_1) \cap \mathcal{I}_5(\frac{\delta}{2}, \eta_{2,\delta})$ (for $\eta_{2,\delta}$ of Lemma 20) and at $A^{ref}(0)$, and evolve $A(t)$ and $A^{ref}(t)$ according to Equation (22). Denote $t_2^*(\mu_1) := \min\left\{ t_2(\mu_1), t_2^{ref}(\mu_1) \right\}$. It holds that

$$\left\| A\big(t_1(\frac{\widetilde{r}(\mu_1)}{4}) + t_2^*(\mu_1) + t_3(\eta_{2,\delta})\big) - A^{ref}\big(t_1(\frac{\widetilde{r}(\mu_1)}{4}) + t_2^*(\mu_1) + t_3(\eta_{2,\delta})\big) \right\|_2 \leq \frac{\delta}{2}$$

for $\eta_{2,\delta}$ described in Lemma 20.

*Proof.* Per Lemma 11 and Corollary 3 for any $t \geq 0$ it holds that $A(t)$, $A^{ref}(t)$ are contained in $[-3,3]^d$, where $\ell$'s gradient is $N$ Lipschitz . Thus per Lemma 29 it holds that

$$\left\| A\big(t_1(\frac{\widetilde{r}(\mu_1)}{4}) + t_2^*(\mu_1) + t_3(\eta_{2,\delta})\big) - A^{ref}\big(t_1(\frac{\widetilde{r}(\mu_1)}{4}) + t_2^*(\mu_1) + t_3(\eta_{2,\delta})\big) \right\|_2$$
$$\leq \left\| A\big(t_1(\frac{\widetilde{r}(\mu_1)}{4}) + t_2^*(\mu_1)\big) - A^{ref}\big(t_1(\frac{\widetilde{r}(\mu_1)}{4}) + t_2^*(\mu_1)\big) \right\|_2 \cdot \exp\left( N \cdot t_3(\eta_{2,\delta}) \right).$$

Next, since $t_2^*(\mu_1) \leq t_2(\mu_1), t_2^{ref}(\mu_1)$ we obtain by Proposition 13 that for any $t \in [t_1(\frac{\widetilde{r}(\mu_1)}{4}), t_1(\frac{\widetilde{r}(\mu_1)}{4}) + t_2^*(\mu_1)]$ it holds that

$$H\big(A(t)\big), H\big(A^{ref}(t)\big) \in \overline{B_{r_3}}(\mathbf{s}).$$

Invoking Proposition 11, the above results in

$$A(t), A^{ref}(t) \in \overline{B_{r_1}}(\mathbf{s}).$$

By definitions of $r_1$ and $\overline{B_{r_1}}(\mathbf{s})$ (Proposition 11), the above yields the following for any $t$ in the interval $[t_1(\frac{\widetilde{r}(\mu_1)}{4}), t_1(\frac{\widetilde{r}(\mu_1)}{4}) + t_2^*(\mu_1)]$ and $k \in [0, 1]$:

$$\left| \lambda_{\min}\left( \nabla^2 \ell\big(k \cdot A(t) + (1-k) \cdot A^{ref}(t)\big) \right) \right| \leq 2|\lambda_-|.$$

Next, it holds that $\lambda_{\min}(\nabla^2\ell(A)) = -\lambda_{\max}(-\nabla^2\ell(A))$ for any $A \in \mathbb{R}^d$. Therefore, invoking Lemma 34 and plugging the above we obtain that

$$\left\| A\big(t_1(\frac{\widetilde{r}(\mu_1)}{4}) + t_2^*(\mu_1)\big) - A^{ref}\big(t_1(\frac{\widetilde{r}(\mu_1)}{4}) + t_2^*(\mu_1)\big) \right\|_2$$

$$\leq \exp\left( \int_0^{t_2^*(\mu_1)} 2|\lambda_-| d\tau \right) \cdot \left\| A\big(t_1(\frac{\widetilde{r}(\mu_1)}{4})\big) - A^{ref}\big(t_1(\frac{\widetilde{r}(\mu_1)}{4})\big) \right\|_2$$

$$= \exp(2t_2^*(\mu_1)|\lambda_-|) \cdot \left\| A\big(t_1(\frac{\widetilde{r}(\mu_1)}{4})\big) - A^{ref}\big(t_1(\frac{\widetilde{r}(\mu_1)}{4})\big) \right\|_2.$$

Note that $\lambda_- < 0$ and so $|\lambda_-| = -\lambda_-$. Thus, recalling Proposition 13 we upper bound $t_2^*(\mu_1)$ and obtain that

$$\left\| A\big(t_1(\frac{\widetilde{r}(\mu_1)}{4}) + t_2^*(\mu_1)\big) - A^{ref}\big(t_1(\frac{\widetilde{r}(\mu_1)}{4}) + t_2^*(\mu_1)\big) \right\|_2$$

$$\leq \exp\left( -\frac{2}{\lambda_-} \ln\left( \frac{G'^2 r_3}{2\sqrt{d}\alpha D_-(\mu_1)} \right) |\lambda_-| \right) \cdot \left\| A\big(t_1(\frac{\widetilde{r}(\mu_1)}{4})\big) - A^{ref}\big(t_1(\frac{\widetilde{r}(\mu_1)}{4})\big) \right\|_2$$

$$= \left( \frac{G'^2 r_3}{2\sqrt{d}\alpha D_-(\mu_1)} \right)^2 \cdot \left\| A\big(t_1(\frac{\widetilde{r}(\mu_1)}{4})\big) - A^{ref}\big(t_1(\frac{\widetilde{r}(\mu_1)}{4})\big) \right\|_2.$$

Applying Lemma 29 once more, we obtain that

$$\left\| A\big(t_1(\frac{\widetilde{r}(\mu_1)}{4})\big) - A^{ref}\big(t_1(\frac{\widetilde{r}(\mu_1)}{4})\big) \right\|_2 \leq \|A(0) - A^{ref}(0)\|_2 \cdot \exp\left( N \cdot t_1(\frac{\widetilde{r}(\mu_1)}{4}) \right).$$

Finally, by Equation (23) and Definition 4 we have at initalization that

$$\|A(0) - A^{ref}(0)\|_2 = |a_2(0) - a_2^{ref}(0)| = \alpha \cdot (1 - \zeta_2).$$

Altogether we obtain the following bound on the divergence:

$$\left\| A\big(t_1(\frac{\widetilde{r}(\mu_1)}{4}) + t_2^*(\mu_1) + t_3(\eta_{2,\delta})\big) - A^{ref}\big(t_1(\frac{\widetilde{r}(\mu_1)}{4}) + t_2^*(\mu_1) + t_3(\eta_{2,\delta})\big) \right\|_2$$

$$\leq \alpha \cdot (1 - \zeta_2) \left( \frac{G'^2 r_3}{2\sqrt{d}\alpha D_-(\mu_1)} \right)^2 \exp\left( N \cdot \big(t_1(\frac{\widetilde{r}(\mu_1)}{4}) + t_3(\eta_{2,\delta})\big) \right).$$

The proof concludes by recalling that the initialization satisfies $A(0) \in \mathcal{I}_5(\frac{\delta}{2}, \eta_{2,\delta})$ and so since $\alpha > 0$ we can rewrite and obtain that

$$\alpha \cdot (1 - \zeta_2) \leq \frac{\frac{\delta}{2}}{\left( \frac{G'^2 r_3}{2\sqrt{d}\alpha D_-(\mu_1)} \right)^2 \exp\left( N \cdot \big(t_1(\frac{\widetilde{r}(\mu_1)}{4}) + t_3(\eta_{2,\delta})\big) \right)}$$

and so

$$\alpha \cdot (1 - \zeta_2) \left( \frac{G'^2 r_3}{2\sqrt{d}\alpha D_-(\mu_1)} \right)^2 \exp\left( N \cdot \big(t_1(\frac{\widetilde{r}(\mu_1)}{4}) + t_3(\eta_{2,\delta})\big) \right) \leq \frac{\delta}{2}.$$

$\square$

We finish this section by proving the following corollary bounding the distance between $\widehat{A_2}$ (the point to which the gradient flow trajecory converges to) and $A^{ref}\big(t_1(\frac{\widetilde{r}(\mu_1)}{4}) + t_2^*(\mu_1) + t_3(\eta_{2,\delta})\big)$, a point which by Lemma 3 is far away from the teacher.

**Corollary 8.** *Let $\delta > 0$. Consider $\mu_1$ from Corollary 7. Suppose we initialize at $A(0) \in \mathcal{I}_3(\frac{\widetilde{r}(\mu_1)}{4}) \cap \mathcal{I}_4(\mu_1) \cap \mathcal{I}_5(\frac{\delta}{2}, \eta_{2,\delta})$ (for $\eta_{2,\delta}$ of Lemma 20) and at $A^{ref}(0)$, and evolve $A(t)$ and $A^{ref}(t)$ according to Equation (22). Denote $t_2^*(\mu_1) := \min\left\{t_2(\mu_1), t_2^{ref}(\mu_1)\right\}$. It holds that*

$$\left\| \widehat{A_2} - A^{ref}\big(t_1(\frac{\widetilde{r}(\mu_1)}{4}) + t_2^*(\mu_1) + t_3(\eta_{2,\delta})\big) \right\|_2 \leq \delta$$

*where $A(t)$ converges to $\widehat{A_2}$ (see Proposition 17).*

*Proof.* Per Lemma 20, it holds that

$$\left\| \widehat{A_2} - A\big(t_1(\frac{\widetilde{r}(\mu_1)}{4}) + t_2^*(\mu_1) + t_3(\eta_{2,\delta})\big) \right\|_2 \leq \frac{\delta}{2}.$$

Per Proposition 18, it holds that

$$\|A\big(t_1(\frac{\widetilde{r}(\mu_1)}{4}) + t_2^*(\mu_1) + t_3(\eta_{2,\delta})\big) - A^{ref}\big(t_1(\frac{\widetilde{r}(\mu_1)}{4}) + t_2^*(\mu_1) + t_3(\eta_{2,\delta})\big)\|_2 \leq \frac{\delta}{2}.$$

The claim thus follows from the triangle inequality. $\qquad\square$

Let $\mathcal{I}_2(\delta_2)$ be the initialization subset defined above, *i.e.*

$$I_2(\delta_2) := \mathcal{I}_3(\frac{\widetilde{r}(\mu_1)}{4}) \cap \mathcal{I}_4(\mu_1) \cap \mathcal{I}_5(\frac{\delta_2}{2}, \eta_{2,\delta_2})$$

for $\delta_2$ of Lemma 3. Invoking the lemma we obtain that for any $L' \geq L + 2$ it holds that

$$Gen_{L'}(\widehat{A_2}) \geq \frac{1}{2} \min\left\{0.1, 1/(9d) \cdot (1 - (0.6)^{1/(L-1)})\right\}$$

which concludes our proof of the fact that gradient flow under $\mathcal{S}_2$ converges to a non-generalizing solution when initialized at $\mathcal{I}_2(\delta)$.

## F.4 Initialization Subsets Intersect

In Proposition 7 we showed that when initializing at $\mathcal{I}_1(\epsilon)$ gradient flow converges to a point $\widehat{A_1}$ which satisfies $Gen_{L'}(\widehat{A_1}) \leq \epsilon$. In Corollary 8 we showed that when initializing at $\mathcal{I}_2(\delta_2)$ gradient flow converges to a point $\widehat{A_2}$ which satisfies $Gen_{L'}(\widehat{A_2}) \geq \frac{1}{2} \min\left\{0.1, 1/(9d) \cdot (1 - (0.6)^{1/(L-1)})\right\}$.

In this section we show that not only do the initialization subsets $\mathcal{I}_1(\epsilon)$ and $\mathcal{I}_2(\delta_2)$ intersect but also that their intersection contains an open subset. For convenience of the reader, we rewrite the full requirements as they appear in the statements of Appendices F.1 to F.3 and state their respective arguments. The base initialization set (Equation (23)) we consider is

$$\mathcal{I}_0 = \left\{ \alpha \cdot (\zeta_1, \ldots, \zeta_d)^\top \in \mathbb{R}^d : \alpha \in (0, \frac{1}{2d}), 1 = \zeta_1 > \zeta_2 > \cdots > \zeta_d > 0 \right\}.$$

$\mathcal{I}_1(\epsilon)$ (Definition 3) was defined as

$$\mathcal{I}_1(\epsilon) = \left\{ A \in \mathcal{I}_0 : \forall j \in \{2, \ldots, d\}. \ \alpha \leq \left( \frac{1 - (1 - \eta_{1,\delta_1})^{L-1} - \eta_{1,\delta_1}}{d - 1} \right)^{\frac{1}{L-1}} \frac{1}{\zeta_j} (1 - \zeta_j^{L-3})^{\frac{1}{L-3}} \right\}$$

for $\eta_{1,\delta_1}$ of Remark 1. $\mathcal{I}_3(\frac{\widetilde{r}(\mu_1)}{4})$ (Definition 10) was defined as

$$\mathcal{I}_3(\frac{\widetilde{r}(\mu_1)}{4}) := \left\{ A \in \mathcal{I}_0 : \alpha \leq \frac{\min\left\{ \frac{\widetilde{r}(\mu_1)}{4}, \left\| A^Z\big(t_1(\frac{\widetilde{r}(\mu_1)}{4})\big) - A^-\big(t_1(\frac{\widetilde{r}(\mu_1)}{4})\big) \right\|_2 \right\}}{6d} e^{-N \cdot t_1(\frac{\widetilde{r}(\mu_1)}{4})}, \ \zeta_d \leq \frac{1}{2} \right\}$$

for $\widetilde{r}(\cdot)$ and $\mu_1$ of Corollaries 4 and 7 respectively. $\mathcal{I}_4(\mu_1)$ (Definition 11) was defined as

$$\mathcal{I}_4(\mu_1) = \left\{ A \in \mathcal{I}_0 : \alpha \leq \frac{r_3}{4 \max\{2, \exp(-2\lambda_-)\} \cdot G'^2 \sqrt{d} D_+(\mu_1)} \right\}$$

$\mathcal{I}_5(\delta_2, \eta_{2,\delta_2})$ (Definition 12) was defined as

$$\mathcal{I}_5(\delta_2, \eta_{2,\delta_2}) = \left\{ A \in \mathcal{I}_0 : (1 - \zeta_2) \leq \frac{\delta_2}{\left(\frac{G'^2 r_3}{2\sqrt{d} D_-(\mu_1)}\right)^2 \exp\left(N \cdot \left(t_1(\frac{\widetilde{r}(\mu_1)}{4}) + t_3(\eta_{2,\delta_2})\right)\right)} \cdot \alpha \right\}$$

for $\eta_{2,\delta_2}$ of Lemma 20. We begin by observing the following simplication:

$$\mathcal{I}_1(\epsilon) \cap \mathcal{I}_0 = \mathcal{I}_0 \cap \left\{ A \in \mathcal{I}_0 : \alpha \leq \left(\frac{1 - (1 - \eta_{1,\delta_1})^{L-1} - \eta_{1,\delta_1}}{d - 1}\right)^{\frac{1}{L-1}} \frac{1}{\zeta_2}(1 - \zeta_2^{L-3})^{\frac{1}{L-3}} \right\}$$

since the right hand side is monotonically decreasing in $\zeta$ and since $\zeta_2 > \zeta_j$ for any $j \in \{3, \ldots d\}$. Next, we also require that $\frac{1}{2} \geq \zeta_3$. Note that this requirement satisfies the requirement of $\mathcal{I}_3$ on the magnitude of $\zeta_d$ (since $\zeta_3 > \zeta_4 > \cdots > \zeta_d$). Moreover, note that $\mathcal{I}_3$ and $\mathcal{I}_4$ impose upper bounds on $\alpha$ which are not related to $\zeta_2, \ldots, \zeta_d$. Therefore, there exists some $\alpha^* > 0$ such that if $\alpha \in (0, \alpha^*)$ then all of these conditions are satisfied. Moving on to the conditions that involve $\alpha$ and $\zeta_2$ we first observe that there exists constants $S, T > 0$ such that

$$\alpha \leq \frac{S}{\zeta_2}(1 - \zeta_2^{L-3})^{\frac{1}{L-3}}$$

is equivalent to the condition from $\mathcal{I}_1$ and

$$(1 - \zeta_2) \leq T\alpha$$

is equivalent to the condition from $\mathcal{I}_5$. Invoking Lemma 36 we obtain that there exist constants $q_1, w_1 \in (0, 1)$ and $q_2, w_2 \in (\frac{1}{2}, 1)$ such that taking $\alpha \in (q_1, w_1)$ and $\zeta_2 \in (q_2, w_2)$ satisfies the two conditions involving $\alpha$ and $\zeta_2$. The above discussion is summarized in the following proposition.

**Proposition 19.** *For any $\epsilon > 0$ there exist constants $q_1, w_1 \in (0, 1)$ and $q_2, w_2 \in (\frac{1}{2}, 1)$ such that the set*

$$\left\{ \alpha \cdot (1, \zeta_2, \ldots, \zeta_d)^\top : (\alpha, \zeta_2) \in (q_1, w_1) \times (q_2, w_2), \frac{1}{2} \geq \zeta_3 > \cdots > \zeta_d \right\}$$

*is contained in the intersection of initialization subsets given by*

$$\mathcal{I}_1(\epsilon) \cap \mathcal{I}_2(\delta_2).$$

Now that we have characterized a set of initializations which satisfy all our requirements, we will show that this set contains an open subset.

**Lemma 21.** *The set of initializations satisfying all our requirements, namely*

$$\left\{ \alpha \cdot (1, \zeta_2, \ldots, \zeta_d)^\top : (\alpha, \zeta_2) \in (q_1, w_1) \times (q_2, w_2), \frac{1}{2} \geq \zeta_3 > \cdots > \zeta_d \right\}$$

*contains an open set, namely a set of the form $(a_1, b_1) \times (a_2, b_2) \times \cdots \times (a_d, b_d)$.*

*Proof.* We begin by restricting $\zeta_3, \ldots, \zeta_d$ by requiring that for $3 \leq j \leq d$, $\zeta_j \in [e_j, f_j]$ where the closed intervals $\{[e_j, f_j]\}_{2 \leq j \leq d}$ satisfy

$$\frac{1}{2} > f_2 > e_2 > f_3 > e_3 > \cdots > f_d > e_d > 0.$$

We can restrict $\alpha$ further, by requiring that $\alpha \in (a_1, b_1)$, where $a_1, b_1$ are chosen such that for all $2 \leq j \leq d$ we have

$$a_1 f_j > b_1 e_j.$$

We now claim that for all $2 \leq j \leq d$

$$(b_1 e_j, a_1 f_j) \subseteq \bigcap_{\alpha \in (a_1, b_1)} [\alpha e_j, \alpha f_j].$$

Indeed, for any $\alpha \in (a_1, b_1)$ we have $b_1 e_j > \alpha e_j$ and $a_1 f_j < \alpha f_j$. It follows that we can take $a_j = b_1 e_j, b_j = a_1 f_j$ and obtain that the set $(a_1, b_1) \times (a_2, b_2) \times \cdots \times (a_d, b_d)$ is contained within

$$\left\{ \alpha(1, \zeta_2, \ldots, \zeta_d) : (\alpha, \zeta_2) \in (q_1, w_1) \times (q_2, w_2), \frac{1}{2} > \zeta_3 > \cdots > \zeta_d \right\}$$

as required. $\qquad\square$

## G  Proof of Theorem 2

The first part of the Theorem is a special case of Proposition 2. We present below the proof for the second part.

For $k \in \mathbb{N}$ and $A \in \mathbb{R}^{d,d}$, we denote by $Gen_k(A)$ the generalization error over sequence length $k$ (Definition 1). Note that $B$ and $C$ are omitted from this notation, as they are fixed to the values $B = \mathbf{1}, C = \mathbf{1}^\top$ throughout the proof. With slight abuse of notation, we also denote $\ell_{\mathcal{S}}(A) := \ell_{\mathcal{S}}(A, B, C)$.

Our proofs considers an augmented objective, given for the training set $\mathcal{S}''$ obtained by scaling all sequences by $\frac{1}{c}$:

$$\mathcal{S}'' = \left\{ \left( \frac{1}{c}\mathbf{x}, \frac{1}{c}y \right) : (\mathbf{x}, \mathbf{y}) \in \mathcal{S}' \right\} = \left\{ \left( \frac{1}{c}\mathbf{x}, \frac{1}{c}y \right) : (\mathbf{x}, \mathbf{y}) \in \mathcal{S} \right\} \cup \{(\mathbf{e}_{\kappa-1}, 1)\}.$$

Since SSMs realize linear mappings, for any $A \in \mathbb{R}^{d,d}$ we have that

$$\ell_{\mathcal{S}''}(A) = \frac{1}{c^2}\ell_{\mathcal{S}'}(A) = \frac{n}{(n+1)c^2}\ell_{\mathcal{S}}(A) + \frac{1}{n+1}\ell_{\{(\mathbf{e}_{\kappa-1}, 1)\}}(A).$$

We prove the required statement for the gradient flow dynamics (Equation (4)) obtained for the set $\mathcal{S}''$, from which it follow that it also holds when we revert back to the set $\mathcal{S}'$, since the two systems of ODEs differ only by a positive scale.

First, consider the objective when the training set consists solely of the sequence

$$\left( \frac{1}{c}\mathbf{x}^\dagger, \phi_{A^*, B^*, C^*} \left( \frac{1}{c}\mathbf{x}^\dagger \right) \right) = (\mathbf{e}_{\kappa-1}, 1) .$$

The corresponding training loss (Equation (3)) is given by

$$\ell_{\{(\mathbf{e}_{\kappa-1}, 1)\}}(A) = \left( 1 - \sum_{l=0}^{\kappa-1} \sum_{i=1}^{d} a_i^l [\mathbf{e}_{\kappa-1}]_{\kappa-l} \right)^2 = \left( 1 - \sum_{i=1}^{d} a_i \right)^2 = \left( 1 - \sum_{i=1}^{d} a_i \right)^2,$$

and the corresponding gradient flow dynamics (Equation (4)) for $\tau \in \mathbb{R}_{\geq 0}$ are given by

$$\forall i \in [d], \ \dot{a}_i(\tau) = -2 \left( \sum_{i=1}^{d} a_i(\tau) - 1 \right).$$

For any initialization of the student SSM $A_0 = (\alpha_1, \ldots, \alpha_d)$, it is straightforward to show that the solution to the above system of ODEs $A(\tau)$ is given by

$$\forall i \in [d], \tau \in \mathbb{R}_{\geq 0}, \ a_i(\tau) = \frac{1 + \sum_{j=1}^{d} \alpha_i - \alpha_j}{d} - \frac{1 - \sum_{j=1}^{d} \alpha_j}{d} \cdot \exp(-2 \cdot d \cdot \tau).$$

Specifically, as $\tau \to \infty$ the solution $A(\tau)$ converges exponentially fast to the point $\tilde{A}_0$ given by

$$\forall i \in [d], \ \tilde{\alpha}_i = \frac{1 + \sum_{j=1}^{d} \alpha_i - \alpha_j}{d},$$

which satisfies

$$\ell_{\{(\mathbf{e}_{\kappa-1},1)\}}(\tilde{A}_0) = \left(1 - \sum_{i=1}^{d} \frac{1 + \sum_{j=1}^{d} \alpha_i - \alpha_j}{d}\right)^2 = 0\,.$$

Consider the open set $\mathcal{I}$ of initializations for the student SSM given by

$$\mathcal{I} := \left(-\frac{1}{d^3}, \frac{1}{d^3}\right)^d$$

Note that for any $\epsilon > 0$, the corresponding set of initializations from Proposition 2 and the set $\mathcal{I}$ intersect, and moreover the intersection contains an open set sufficient for our needs. We next prove that for any initialization $A_0 = (\alpha_1, \ldots, \alpha_d) \in \mathcal{I}$, the point $\tilde{A}_0$ that the system converges to as $\tau \to \infty$ satisfies $Gen_k(\tilde{A}_0) > 1 - \frac{4}{d}$. For any index $i \in [d]$ it holds that

$$\forall j \in [d]\,, |\alpha_i - \alpha_j| < \frac{1}{d^2} \implies \left|\sum_{j=1}^{d} \alpha_i - \alpha_j\right| \leq \sum_{j=1}^{d} |\alpha_i - \alpha_j| < \frac{1}{d}\,.$$

The latter implies that

$$\frac{1}{d^2} \leq \frac{1}{d} - \frac{1}{d^2} < \frac{1 + \sum_{j=1}^{d} \alpha_i - \alpha_j}{d} < \frac{1}{d} + \frac{1}{d^2} < \frac{2}{d}\,,$$

which results in

$$\left(\frac{1 + \sum_{j=1}^{d} \alpha_i - \alpha_j}{d}\right)^2 \in \left(\frac{1}{d^4}, \frac{4}{d^2}\right)\,.$$

Overall, since $d \geq 5$ and $k \geq 3$ we obtain that

$$\begin{aligned}
Gen_k(\tilde{A}_0) &= \max_{k' \in \{0,1,\ldots,k-1\}} \left|1 - \sum_{i=1}^{d} \tilde{\alpha}_i^{k'}\right| \\
&\geq \left|1 - \sum_{i=1}^{d} \tilde{\alpha}_i^2\right| \\
&= \left|1 - \sum_{i=1}^{d} \left(\frac{1 + \sum_{j=1}^{d} \alpha_i - \alpha_j}{d}\right)^2\right| \\
&> 1 - \frac{4d}{d^2} \\
&= 1 - \frac{4}{d}\,.
\end{aligned}$$

By the continuity of $\ell_{\{(\mathbf{e}_{\kappa-1},1)\}}(A)$ and $Gen_k(A)$, the above implies that for any initialization $A_0 \in \mathcal{I}$ there exists a time $t \in \mathbb{R}_{>0}$ such that when initializing at $A(0) = A_0$ and evolving $A(\tau)$ according to the above dynamics it holds that

$$\ell_{\{(\mathbf{e}_{\kappa-1},1)\}}(A(t)) \leq \frac{\delta}{4}$$

and

$$Gen_k(A(t)) \geq \frac{3}{4} - \frac{3}{d}\,.$$

Now consider the objective for the full set $\mathcal{S}''$

$$\ell_{\mathcal{S}''}(A) = \frac{n}{(n+1)c^2}\ell_{\mathcal{S}}(A) + \frac{1}{n+1}\ell_{\{(\mathbf{e}_{\kappa-1},1)\}}(A)\,,$$

whose corresponding gradient flow dynamics (Equation (4)) for $\tau \in \mathbb{R}_{\geq 0}$ are given by

$$\forall i \in [d]\,, \; \dot{a}_i(\tau) = -\frac{n}{(n+1)c^2}\frac{\partial}{\partial a_i(\tau)}\ell_{\mathcal{S}}(A(\tau)) - \frac{2}{n+1}\left(\sum_{i=1}^{d} a_i(\tau) - 1\right).$$

Observe that the left terms of both the objective and the dynamics all vanish as $c \to \infty$. Additionally, note that when the left terms of the dynamics vanish, the dynamics differ by a positive scale from those discussed above when the training set consists solely of the sequence $(\mathbf{e}_{\kappa-1}, 1)$. Therefore by continuity, for any initialization $A_0 \in \mathcal{I}$ and any time $t' \in \mathbb{R}_{>0}$ there exists $c \in \mathbb{R}_{>0}$ such that when initializing $A(0) = A_0 = A'(0)$, evolving $A(\tau)$ according to the dynamics given when the left terms vanish, and evolving $A'(\tau)$ according to the full dynamics, it holds that $\|A(t') - A'(t')\|_2$ is sufficiently small to ensure that both

$$\left|\frac{1}{n+1}\ell_{\{(\mathbf{e}_{\kappa-1},1)\}}(A(t')) - \ell_{\mathcal{S}''}(A'(t'))\right| \leq \frac{\delta}{4}$$

and

$$|Gen_k(A(t')) - Gen_k(A'(t'))| \leq \frac{1}{4} - \frac{1}{d}.$$

The proof follows by invoking the last claim for the time $t$ obtained previously. $\qquad\square$

## H   Auxiliary Theorems and Lemmas

In this section we provide additional Theorems and Lemmas used throughout our proofs.

**Lemma 22.** *Let $f : \mathbb{R}^d \to \mathbb{R}$ be some differentiable function. Suppose we optimize over $f$ by initializing $\mathbf{x}(0) := \mathbf{x_0}$ for some $\mathbf{x_0} \in \mathbb{R}^d$ and updating using gradient flow, i.e.:*

$$\dot{\mathbf{x}}(t) := \frac{d}{dt}\mathbf{x}(t) = -\nabla f(\mathbf{x}(t)).$$

*Then the objective is non-increasing w.r.t time, i.e. for any $t \geq 0$ it holds that*

$$\frac{d}{dt}f(\mathbf{x}(t)) \leq 0.$$

*Proof.* Applying the chain rule, we obtain the following:

$$\frac{d}{dt}f(\mathbf{x}(t)) = \nabla f(\mathbf{x}(t))^\top \frac{d}{dt}\mathbf{x}(t) = f(\mathbf{x}(t))^\top(-f(\mathbf{x}(t))) = -\|f(\mathbf{x}(t))\|_2^2 \leq 0.$$

$\qquad\square$

**Lemma 23.** *Let $f : \mathbb{R}^d \to \mathbb{R}$ be some continuously differentiable function, which is also coercive, namely*

$$\lim_{\|x\|\to\infty} f(x) = \infty.$$

*Suppose we optimize over $f$ by initializing $\mathbf{x}(0) := \mathbf{x_0}$ for some $\mathbf{x_0} \in \mathbb{R}^d$ and updating using gradient flow, i.e.:*

$$\dot{\mathbf{x}}(t) := \frac{d}{dt}\mathbf{x}(t) = -\nabla f(\mathbf{x}(t)). \tag{24}$$

*Then there exists a global solution to the above ODE, namely a curve $\mathbf{x}(t)$ which satisfies the above equation for all $t \geq 0$.*

*Proof.* By Lemma 22 and the coercivity of $f$, the trajectories of gradient flow cannot escape from some compact set $K := K(\mathbf{x}_0)$. Because $f$ is continuously differentiable $\nabla f$ has some finite Lipschitz constant on $K$. Existence of the solution for all $t \geq 0$ now follows from the Picard–Lindelöf theorem (see [80]). $\qquad\square$

**Theorem 4.** *Let $\mathcal{V} \subseteq \mathbb{R}^d$ be an open set. Let $f : \mathcal{V} \to \mathbb{R}$ be a non-negative differentiable function satisfying the following conditions:*

- *The set $X^* := \{\mathbf{x} \in \mathcal{V} : f(\mathbf{x}) = 0\}$ is not empty.*

- *There exists $\mu > 0$ such that for any $\mathbf{x} \in \mathcal{V}$ it holds that*

$$\|\nabla f(\mathbf{x})\|_2^2 \geq 2\mu f(\mathbf{x}).$$

- *There exists $M > 0$ such that $\nabla f(\mathbf{x})$ is $M$-Lipschitz in $\mathcal{V}$.*

*Suppose we optimize over $f$ by initializing $\mathbf{x}(0) := \mathbf{x}_0$ for some $\mathbf{x}_0 \in \mathcal{V}$ and evolving via gradient flow, i.e. via the update rule*

$$\dot{\mathbf{x}}(t) := \frac{d}{dt}\mathbf{x}(t) = -\nabla f(\mathbf{x(t)}).$$

*Assume the set $\mathcal{V}$ is not escaped, i.e. for any time $t \geq 0$ it holds that $\mathbf{x}(t) \in \mathcal{V}$. Then it holds that*

$$\int_0^\infty \|\dot{\mathbf{x}}(t)\|_2 dt = \int_0^\infty \|\nabla f(\mathbf{x}(t))\|_2 dt \leq \sqrt{\frac{M}{\mu}} Dist(\mathbf{x_0}, X^*).$$

*Proof.* The theorem is a restatement of theorem 9 in [29]. $\qquad\square$

**Definition 13.** Let $\mathcal{V} \subseteq \mathbb{R}^d$ be an open set. Let $f : \mathcal{V} \to \mathbb{R}$ be a differentiable function. We say that $f$ satisfies the *Polyak-Lojasiewicz condition* with coefficient $\mu > 0$ at $\mathbf{x} \in \mathcal{V}$

$$\|\nabla f(\mathbf{x})\|_2^2 \geq 2\mu(f(\mathbf{x}) - \min_{\mathbf{y} \in \mathbb{R}^d} f(\mathbf{y})).$$

If the above holds for all $\mathbf{x} \in \mathcal{V}$ we say that $f$ satisfies the PL condition in $\mathcal{V}$.

**Lemma 24.** *Let $\mathcal{V} \subseteq \mathbb{R}^d$ be an open set. Let $f : \mathcal{V} \to \mathbb{R}$ be a non-negative differentiable function satisfying the following conditions:*

- *The set $\mathbf{x}^* := \{\mathbf{x} \in \mathcal{V} : f(\mathbf{x}) = 0\}$ is not empty.*

- *PL condition (Definition 13) - there exists $\mu > 0$ such that for any $\mathbf{x} \in \mathcal{V}$ it holds that*

$$\|\nabla f(\mathbf{x})\|^2 \geq 2\mu f(\mathbf{x}).$$

*Suppose we optimize over $f$ by initializing $\mathbf{x}(0) := \mathbf{x}_0$ for some $\mathbf{x}_0 \in \mathcal{V}$ and evolving via gradient flow, i.e. via the update rule*

$$\dot{\mathbf{x}}(t) := \frac{d}{dt}\mathbf{x}(t) = -\nabla f(\mathbf{x(t)}).$$

*Assume the set $\mathcal{V}$ is not escaped, i.e. for any time $t \geq 0$ it holds that $\mathbf{x}(t) \in \mathcal{V}$. Then for any $t \geq 0$ it holds that*

$$f(\mathbf{x}(t)) \leq f(\mathbf{x}(0)) \cdot \exp(-2\mu \cdot t).$$

*Namely, it holds that*

$$\lim_{t \to \infty} f(\mathbf{x}(t)) = 0.$$

*Proof.* Let $t \geq 0$. By the chain rule, it holds that

$$\frac{d}{dt}f(\mathbf{x}(t)) = \nabla f(\mathbf{x}(t))^\top \frac{d}{dt}\mathbf{x}(t) = f(\mathbf{x}(t))^\top(-f(\mathbf{x}(t))) = -\|f(\mathbf{x}(t))\|_2^2.$$

By the PL condition and since $\mathcal{V}$ is not escaped, we have that

$$\frac{d}{dt}f(\mathbf{x}(t)) = -\|\nabla f(\mathbf{x}(t))\|_2^2 \leq -2\mu f(\mathbf{x}(t)).$$

Therefore, by Grönwall's inequality [21] we have that

$$f(\mathbf{x}(t)) \leq f(\mathbf{x}(0)) \cdot \exp\left(\int_0^t -2\mu d\tau\right) = f(\mathbf{x}(0)) \cdot \exp(-2\mu \cdot t).$$

Taking the limit as $t \to \infty$ completes the proof. $\qquad\square$

**Lemma 25.** *Let $\mathcal{V} \subseteq \mathbb{R}^d$ be an open set. Let $f : \mathcal{V} \to \mathbb{R}$ be a non-negative differentiable function satisfying the following conditions:*

- *The set $X^* := \{\mathbf{x} \in \mathcal{V} : f(\mathbf{x}) = 0\}$ is not empty.*

- *$f$ satisfies the PL condition with coefficient $\mu > 0$ (see Definition 13).*

- *Lipschitz gradient - there exists $M > 0$ such that $\nabla f(\mathbf{x})$ is $M$-Lipschitz in $\mathcal{V}$.*

*Suppose we optimize over $f$ by initializing $\mathbf{x}(0) := \mathbf{x}_0$ for some $\mathbf{x}_0 \in \mathcal{V}$ and evolving via gradient flow, i.e. via the update rule*

$$\dot{\mathbf{x}}(t) := \frac{d}{dt}\mathbf{x}(t) = -\nabla f(\mathbf{x}(\mathbf{t})).$$

*Assume the set $\mathcal{V}$ is not escaped, i.e. for any time $t \geq 0$ it holds that $\mathbf{x}(t) \in \mathcal{V}$. Then the limit $\lim_{t\to\infty} \mathbf{x}(t) = \mathbf{x}^*$ exists and satisfies $f(\mathbf{x}^*) = 0$ and*

$$\|\mathbf{x}^* - \mathbf{x}_0\|_2 \leq \sqrt{\frac{M}{\mu}} Dist(\mathbf{x}_0, X^*).$$

*Proof.* Let $\epsilon > 0$. By theorem 4, it holds that

$$\int_0^\infty \|\dot{\mathbf{x}}(\tau)\|_2 d\tau \leq \sqrt{\frac{M}{\mu}} \mathrm{Dist}(\mathbf{x}_0, X^*)$$

which is finite since $X^*$ is not empty. Hence, there exists $t^* \geq 0$ such that for any $t \geq t^*$ it holds that

$$\int_t^\infty \|\dot{\mathbf{x}}(\tau)\|_2 d\tau \leq \epsilon.$$

Therefore, for any $t_2 \geq t_1 \geq t^*$ it holds by the fundamental theorem of calculus and by the triangle inequality that

$$\begin{aligned}
\|\mathbf{x}(t_2) - \mathbf{x}(t_1)\|_2 &= \left\| \mathbf{x}_0 + \int_0^{t_2} \dot{\mathbf{x}}(\tau)d\tau - \mathbf{x}_0 - \int_0^{t_1} \dot{\mathbf{x}}(\tau)d\tau \right\|_2 \\
&= \left\| \int_{t_1}^{t_2} \dot{\mathbf{x}}(\tau)d\tau \right\|_2 \\
&\leq \int_{t_1}^{t_2} \|\dot{\mathbf{x}}(\tau)\|_2 d\tau \\
&\leq \int_{t_1}^\infty \|\dot{\mathbf{x}}(\tau)\|_2 d\tau \\
&\leq \epsilon.
\end{aligned}$$

Thus, the Cauchy convergence criterion is met and so the limit $\lim_{t\to\infty} \mathbf{x}(t) = \mathbf{x}^*$ exists. Plugging $f$'s continuity and lemma 24 yields the following

$$f(\mathbf{x}^*) = f\left( \lim_{t\to\infty} \mathbf{x}(t) \right) = \lim_{t\to\infty} f\left( \mathbf{x}(t) \right) = 0.$$

Finally, by continuity and by the triangle inequality it holds that

$$\begin{aligned}
\|\mathbf{x}^* - \mathbf{x}_0\|_2 &= \left\| \mathbf{x}_0 + \int_0^\infty \dot{\mathbf{x}}(\tau)d\tau - \mathbf{x}_0 \right\|_2 \\
&= \left\| \int_0^\infty \dot{\mathbf{x}}(\tau)d\tau \right\|_2 \\
&\leq \int_0^\infty \|\dot{\mathbf{x}}(t)\|_2 \\
&\leq \sqrt{\frac{M}{\mu}} \mathrm{Dist}(\mathbf{x}_0, X^*)
\end{aligned}$$

as required. □

**Lemma 26.** *Let $a, b \in \mathbb{R}$. An eigendecomposition of the matrix $(a - b)I_d + b\mathbf{1}_{d,d}$, where $\mathbf{1}_{d,d}$ is the $d, d$ all ones matrix, is the following:*

- *The eigenvector $\mathbf{1}$ with the eigenvalue $a + (d - 1)b$.*

- *For $j \in \{2, \dots, d\}$ the eigenvector $\mathbf{e}_1 - \mathbf{e}_j$ with the eigenvalue $a - b$.*

*Proof.* First, it holds that

$$[(a - b)I_d + b\mathbf{1}_{d,d}]\mathbf{1} = (a - b)\mathbf{1} + b \cdot d\mathbf{1} = (a + (d - 1)b)\mathbf{1}$$

hence $\mathbf{1}$ is an eigenvector with the eigenvalue $a + (d - 1)b$. Next, note that for any $j \in \{2, \dots, d\}$ we have

$$[(a - b)I_d + b\mathbf{1}_{d,d}](\mathbf{e}_1 - \mathbf{e}_j) = (a - b)\mathbf{e}_1 + b\mathbf{1} - (a - b)\mathbf{e}_j - b\mathbf{1} = (a - b)(\mathbf{e}_1 - \mathbf{e}_j)$$

hence $\mathbf{e}_1 - \mathbf{e}_j$ is an eigenvector with the eigenvalue $a - b$. Finally, note that the set $\{\mathbf{1}, \mathbf{e}_1 - \mathbf{e}_2, \dots, \mathbf{e}_1 - \mathbf{e}_d\}$ is linearly independent and thus spans $\mathbb{R}^d$. Therefore, the above eigenvectors and eigenvalues constitute and eigendecomposition of $(a - b)I_d + b\mathbf{1}_{d,d}$. $\quad\square$

**Lemma 27.** *Let $W \in \mathbb{R}^{d,d}$ be a symmetric matrix and $\mathbf{b} \in \mathbb{R}^d$ be a vector. The solution of the linear dynamical system*

$$\dot{\mathbf{y}}(t) = -W(\mathbf{y}(t) - \mathbf{b})$$

*is given by*

$$\mathbf{y}(t) = \exp(-W)(\mathbf{y}(0) - \mathbf{b}) = Q\exp(-t \cdot \Lambda)Q^\top(\mathbf{y}(0) - \mathbf{b}) + \mathbf{b}$$

*where $W = Q\Lambda Q^\top$ is any orthogonal eigendecomposition of $W$.*

*Proof.* Using the change of variables $\mathbf{z}(t) = \mathbf{y}(t) - \mathbf{b}$, the given system simplifies to

$$\dot{\mathbf{z}}(t) = -W\mathbf{z}(t)$$

whose solution is given by

$$\mathbf{z}(t) = \exp(-t \cdot W)\mathbf{z}(0).$$

Reversing the change of variables and reorganizing yields

$$\mathbf{y}(t) = \exp(-t \cdot W)(\mathbf{y}(0) - \mathbf{b}) + \mathbf{b}.$$

Let $W = Q\Lambda Q^\top$ be an orthogonal eigendecomposition of the symmetric $W$. Then we have by the definition of matrix exponential that

$$\mathbf{y}(t) = Q\exp(-t \cdot \Lambda)Q^\top(\mathbf{y}(0) - \mathbf{b}) + \mathbf{b}$$

as required. $\quad\square$

**Lemma 28.** *Let $\mathbf{x}_0 \in \mathbb{R}^d$. Let $\mathcal{V}_1, \mathcal{U}_1 \subseteq \mathbb{R}^d$ be neighborhoods of $\mathbf{x}_0$. Let $H : \mathcal{V}_1 \to \mathcal{U}_1$ be a $C^3$ diffeomorphism. There exists $r > 0$ such that for any $r_1 \in (0, r]$ there exist $r_2 \in (0, r_1)$ and $r_3 > 0$ for which*

- $H[\overline{B_{r_2}}(\mathbf{s})] \subseteq \overline{B_{r_3}}(\mathbf{s}) \subseteq H[\overline{B_{r_1}}(\mathbf{s})].$

- $H|_{\overline{B_{r_1}}(\mathbf{s})}$ *is Lipschitz.*

- $H^{-1}|_{\overline{B_{r_3}}(\mathbf{s})}$ *is Lipschitz.*

*Proof.* $\mathcal{V}_1$ and $\mathcal{U}_1$ are neighborhoods of $\mathbf{x}_0$ and so there exist $r^{'}, r^{''} > 0$ for which $\overline{B_{r^{'}}}(\mathbf{x}_0) \subseteq \mathcal{V}_1$ and $\overline{B_{r^{''}}}(\mathbf{x}_0) =: \mathcal{U}_2 \subseteq \mathcal{U}_1$. Hence, by $H$'s continuity there exists some small enough $r > 0$ for which $\overline{B_r}(\mathbf{x}_0) \subseteq \mathcal{V}_1$ is mapped by $H$ to $H[\overline{B_r}(\mathbf{x}_0)] \subseteq \mathcal{U}_2$. Fix $r_1 \in (0, r]$. Then it holds that $\overline{B_{r_1}}(\mathbf{x}_0)$ satisfies

$$\overline{B_{r_1}}(\mathbf{x}_0) \subseteq \overline{B_r}(\mathbf{x}_0) \subseteq \mathcal{V}_1$$

and is mapped by $H$ to
$$H\big[\overline{B_{r_1}}(\mathbf{x}_0)\big] \subseteq H\big[\overline{B_r}(\mathbf{x}_0)\big] \subseteq \mathcal{U}_2 \,.$$
Since $\overline{B_{r_1}}(\mathbf{x}_0)$ is a compact ball and since $H$ is $C^3$, we obtain that $H$ is Lipschitz over $\overline{B_{r_1}}(\mathbf{x}_0)$, *i.e.* $H|_{\overline{B_{r_1}}(\mathbf{x}_0)}$ is Lipschitz. Similarly, we obtain that $H^{-1}$ is Lipschitz over $\mathcal{U}_2$, *i.e.* $H^{-1}|_{\mathcal{U}_2}$ is Lipschitz. Therefore since $H\big[\overline{B_{r_1}}(\mathbf{x}_0)\big] \subseteq \mathcal{U}_2$ we obtain that $H^{-1}|_{H[\overline{B_{r_1}}(\mathbf{x}_0)]}$ is Lipschitz. Next, note that for any $r_2 \in (0, r_1)$ the compact ball $\overline{B_{r_2}}(\mathbf{x}_0)$ satisfies $H\big[\overline{B_{r_2}}(\mathbf{x}_0)\big] \subseteq H\big[\overline{B_{r_1}}(\mathbf{x}_0)\big]$. Hence, by taking a small enough $r_2$ we can guarantee by $H$'s continuity that there exists some $r_3 > 0$ for which $\overline{B_{r_3}}(\mathbf{x}_0)$ satisfies
$$H\big[\overline{B_{r_2}}(\mathbf{x}_0)\big] \subseteq \overline{B_{r_3}}(\mathbf{x}_0) \subseteq H\big[\overline{B_{r_1}}(\mathbf{x}_0)\big] \,.$$
Since $H^{-1}|_{H[\overline{B_{r_1}}(\mathbf{x}_0)]}$ is Lipschitz and $\overline{B_{r_3}}(\mathbf{x}_0) \subseteq H\big[\overline{B_{r_1}}(\mathbf{x}_0)\big]$ we obtain that $H^{-1}|_{\overline{B_{r_3}}(\mathbf{x}_0)}$ is Lipschitz. $\qquad\square$

**Lemma 29.** *Let $f : \mathbb{R}^d \to \mathbb{R}^d$ be a vector field and let $B \subseteq \mathbb{R}^d$ be a bounded and compact space. Suppose $f$ is $N$-Lipschitz within $B$ for some constant $N > 0$. Consider the following system of ODEs:*
$$\dot{\mathbf{x}}(t) = f\big(\mathbf{x}(t)\big) \,.$$
*Consider two initialization points $\mathbf{x}_1(0), \mathbf{x}_2(0) \in B$. Suppose we evolve $\mathbf{x}_1(t), \mathbf{x}_2(t)$ according to the above system. If for any $t \geq 0$ it holds that $\mathbf{x}_1(t), \mathbf{x}_2(t) \in B$, then*
$$\|\mathbf{x}_1(0) - \mathbf{x}_2(0)\|_2 \cdot \exp(-N \cdot t) \leq \|\mathbf{x}_1(t) - \mathbf{x}_2(t)\|_2 \leq \|\mathbf{x}_1(0) - \mathbf{x}_2(0)\|_2 \cdot \exp(N \cdot t) \,.$$

*Proof.* Let $t \geq 0$. Applying the chain rule, we obtain the following:
$$\frac{d}{dt}\|\mathbf{x}_1(t) - \mathbf{x}_2(t)\|_2 = \frac{d}{dt}\sqrt{\big(\mathbf{x}_1(t) - \mathbf{x}_2(t)\big)^\top \big(\mathbf{x}_1(t) - \mathbf{x}_2(t)\big)}$$
$$= \frac{1}{2\|\mathbf{x}_1(t) - \mathbf{x}_2(t)\|_2}\frac{d}{dt}\big(\mathbf{x}_1(t) - \mathbf{x}_2(t)\big)^\top \big(\mathbf{x}_1(t) - \mathbf{x}_2(t)\big)$$
$$= \frac{2}{2\|\mathbf{x}_1(t) - \mathbf{x}_2(t)\|_2} \cdot \big(\mathbf{x}_1(t) - \mathbf{x}_2(t)\big)^\top \frac{d}{dt}\big(\mathbf{x}_1(t) - \mathbf{x}_2(t)\big)$$
$$= \frac{1}{\|\mathbf{x}_1(t) - \mathbf{x}_2(t)\|_2} \cdot \big(\mathbf{x}_1(t) - \mathbf{x}_2(t)\big)^\top \Big(f\big(\mathbf{x}_1(t)\big) - f\big(\mathbf{x}_2(t)\big)\Big) \,.$$
Thus, applying the Cauchy Schwarz inequality we obtain
$$\frac{d}{dt}\|\mathbf{x}_1(t) - \mathbf{x}_2(t)\|_2 \leq \frac{1}{\|\mathbf{x}_1(t) - \mathbf{x}_2(t)\|_2} \cdot \|\mathbf{x}_1(t) - \mathbf{x}_2(t)\|_2 \cdot \|f\big(\mathbf{x}_1(t)\big) - f\big(\mathbf{x}_2(t)\big)\|_2$$
$$= \|f\big(\mathbf{x}_1(t)\big) - f\big(\mathbf{x}_2(t)\big)\|_2 \,.$$
$f$ is $N$-Lipschitz within $B$, and so since $\mathbf{x}_1(t), \mathbf{x}_2(t) \in B$ we obtain
$$\frac{d}{dt}\|\mathbf{x}_1(t) - \mathbf{x}_2(t)\|_2 \leq \|f\big(\mathbf{x}_1(t)\big) - f\big(\mathbf{x}_2(t)\big)\|_2 \leq N \cdot \|\mathbf{x}_1(t) - \mathbf{x}_2(t)\|_2 \,.$$
Finally, plugging Grönwall's inequality [21] results in
$$\|\mathbf{x}_1(t) - \mathbf{x}_2(t)\|_2 \leq \|\mathbf{x}_1(0) - \mathbf{x}_2(0)\|_2 \cdot \exp(N \cdot t) \,.$$
Next, consider the following system of ODEs which we coin the *reversal of $f$*:
$$\dot{\overline{\mathbf{x}}}(t) = -f\big(\overline{\mathbf{x}}(t)\big) \,.$$
Consider the initialization points $\overline{\mathbf{x}_1}(0) = \mathbf{x}_1(t)$ and $\overline{\mathbf{x}_2}(0) = \mathbf{x}_2(t)$. Suppose we evolve $\overline{\mathbf{x}_1}(t), \overline{\mathbf{x}_2}(t)$ according to the reversal of $f$. Then it holds that for any time $\overline{t} \in [0, t]$ and any $i \in [2]$ we have
$$\overline{\mathbf{x}_i}(\overline{t}) = \mathbf{x}_i(t - \overline{t})$$
hence $\overline{\mathbf{x}_i}(\overline{t}) \in B$. As Lipschitz continuity is invariant to sign, $-f$ is $N$-Lipschitz within $B$. Therefore, we can apply the above claim on the reversal of $f$, and obtain that
$$\|\mathbf{x}_1(0) - \mathbf{x}_2(0)\|_2 = \|\overline{\mathbf{x}_1}(t) - \overline{\mathbf{x}_2}(t)\|_2$$
$$\leq \|\overline{\mathbf{x}_1}(0) - \overline{\mathbf{x}_2}(0)\|_2 \cdot \exp(N \cdot t)$$
$$= \|\mathbf{x}_1(t) - \mathbf{x}_2(t)\|_2 \cdot \exp(N \cdot t) \,.$$
The proof concludes by rearranging of the left and right hand side. $\qquad\square$

**Lemma 30.** *Let $f : \mathbb{R}^d \to \mathbb{R}^d$ be a Lipschitz continuous vector field. Consider the following system of ODEs:*

$$\dot{\mathbf{x}}(t) = f\big(\mathbf{x}(t)\big).$$

*Consider two initialization points $\mathbf{x}_1(0), \mathbf{x}_2(0) \in \mathbb{R}^d$. Suppose we evolve $\mathbf{x}_1(t), \mathbf{x}_2(t)$ according to the above system. If $\mathbf{x}_1(0) \neq \mathbf{x}_2(0)$ then for any $t \in \mathbb{R}$ it holds that $\mathbf{x}_1(t) \neq \mathbf{x}_2(t)$.*

*Proof.* The argument follows from the Picard–Lindelöf existence and uniqueness theorem, which states that for a given initialization $\mathbf{x}(0)$, there exists a unique solution $\mathbf{x}(t)$ to the ODE system

$$\dot{\mathbf{x}}(t) = f\big(\mathbf{x}(t)\big).$$

$\qquad\square$

**Lemma 31.** *Let $\mathbf{x} \in \mathbb{R}^d$. Denote $\mathbf{x}$'s representation with the orthogonal subspaces $\mathcal{W}_1$ and $\mathcal{W}_2$ to be*

$$\mathbf{x} = \beta_1 \cdot \mathbf{1} + \beta_2 \cdot \mathbf{v}$$

*where $\mathbf{v} \in \mathcal{W}_2$ is a unit vector. There exist $i, j \in [d]$ such that*

$$|x_i - x_j| \geq \frac{|\beta_2|}{d}.$$

*Proof.* Denote $\mathbf{v}$'s representation with the basis vectors of $\mathcal{W}_2$ to be

$$\mathbf{v} = \sum_{k=2}^{d} \lambda_k (\mathbf{e}_1 - \mathbf{e}_k) = \begin{pmatrix} \sum_{k=2}^{d} \lambda_k \\ -\lambda_2 \\ \vdots \\ -\lambda_d \end{pmatrix}$$

where $\lambda_2, \ldots, \lambda_d \in \mathbb{R}$. Denote $\Lambda := (\lambda_2, \ldots, \lambda_d)^\top \in \mathbb{R}^{d-1}$. As $\mathbf{v}$ is a unit vector, it holds that

$$1 = \|\mathbf{v}\|_2^2 = \left(\sum_{k=2}^{d} \lambda_k\right)^2 + \sum_{k=2}^{d} (-\lambda_k)^2 = \left(\sum_{k=2}^{d} \lambda_k\right)^2 + \sum_{k=2}^{d} \lambda_k^2.$$

Hence,

$$1 - \left(\sum_{k=2}^{d} \lambda_k\right)^2 = \sum_{k=2}^{d} \lambda_k^2.$$

Applying the Cauchy-Schwartz inequality we obtain the following:

$$\left(\sum_{k=2}^{d} \lambda_k\right)^2 = \left(\sum_{k=2}^{d} 1 \cdot \lambda_k\right)^2 = \langle \mathbf{1}, \Lambda \rangle \leq \|\mathbf{1}\|_2^2 \cdot \|\Lambda\|^2 = (d-1) \sum_{k=2}^{d} \lambda_k^2.$$

Therefore, we obtain that

$$\sum_{k=2}^{d} \lambda_k^2 = 1 - \left(\sum_{k=2}^{d} \lambda_k\right)^2 \geq 1 - (d-1) \sum_{k=2}^{d} \lambda_k^2 \implies d \sum_{k=2}^{d} \lambda_k^2 \geq 1 \implies \sum_{k=2}^{d} \lambda_k^2 \geq \frac{1}{d}.$$

Therefore, there exists $i^* \in \{2, \ldots, d\}$ for which $\lambda_{i^*}^2 \geq \frac{1}{d(d-1)}$ and thus $|\lambda_{i^*}| \geq \frac{1}{\sqrt{d(d-1)}}$. If there exists $j \in \{2, \ldots, d\}$ for which $\lambda_j$ has a distinct sign than $\lambda_{i^*}$, then it holds that

$$|v_{i^*} - v_j| = |-\lambda_{i^*} + \lambda_j| \geq |\lambda_{i^*} - 0| \geq \frac{1}{\sqrt{d(d-1)}}.$$

Otherwise, all entries of $\Lambda$ share the same sign, and so it holds that for any $j \in \{2, \ldots, d\}$

$$|v_1 - v_j| = |\sum_{k=2}^{d} \lambda_k - (-\lambda_j)| = \sum_{k=2}^{d} |\lambda_k| + |\lambda_j| \geq |\lambda_{i^*}| \geq \frac{1}{\sqrt{d(d-1)}}.$$

Therefore, there must exist $i, j \in [d]$ for which $|v_i - v_j| \geq \frac{1}{\sqrt{d(d-1)}}$, resulting in the following:

$$|x_i - x_j| = |\beta_1 + \beta_2 \cdot v_i - \beta_1 - \beta_2 \cdot v_j| = |\beta_2| \cdot |v_i - v_j| \geq \frac{|\beta_2|}{\sqrt{d(d-1)}}.$$

$\qquad\square$

**Proposition 20.** *Let* $x, y \in [-s, s]$ *for some* $s > 0$ *such that* $|x - y| \geq b$ *for some* $b > 0$. *Then for any* $k \in \mathbb{N}_{odd}$ *it holds that* $|x^k - y^k| \geq (\frac{b}{2})^k$. *Additionally, if* $y \neq 0$ *then* $|1 - \frac{x^k}{y^k}| \geq (\frac{b}{2s})^k$.

*Proof.* Suppose WLOG that $x \geq y$. By the triangle inequality it holds that $\max\{|x|, |y|\} \geq \frac{b}{2}$. If $x \geq 0 \geq y$, then since $k \in \mathbb{N}_{odd}$ it holds that

$$|x^k - y^k| = |x|^k + |y|^k \geq (\frac{b}{2})^k.$$

Now suppose that $x, y \geq 0$. Then we have that

$$|x^k - y^k| = x^k - y^k \geq (y + b)^k - y^k \geq b^k \geq (\frac{b}{2})^k.$$

The case of $x, y \leq 0$ is identical. As for the second inequality, If $y \neq 0$, we get

$$|1 - \frac{x^k}{y^k}| = |\frac{y^k - x^k}{y^k}| = \frac{|x^k - y^k|}{|y^k|} \geq \frac{(\frac{b}{2})^k}{|y^k|}.$$

Since $y \in [-s, s]$ we get that $|y^k| \leq s^k$ and so

$$\frac{(\frac{b}{2})^k}{|y^k|} \geq (\frac{b}{2s})^k.$$

$\square$

**Proposition 21.** *Let* $x, y, z \in \mathbb{R}$ *such that* $y, x.z \neq 0$, *and let* $\widehat{b} > 0$. *If* $|1 - \frac{x}{y}| \geq \widehat{b}$ *then either* $|1 - \frac{x}{z}| \geq \frac{\widehat{b}}{\widehat{b}+2}$ *or* $|1 - \frac{y}{z}| \geq \frac{\widehat{b}}{\widehat{b}+2}$.

*Proof.* Assume to the contrary that both $|1 - \frac{x}{z}| < \frac{\widehat{b}}{\widehat{b}+2}$ and $|1 - \frac{y}{z}| < \frac{\widehat{b}}{\widehat{b}+2}$. This implies that

$$0 < 1 - \frac{\widehat{b}}{\widehat{b}+2} \leq \frac{x}{z}, \frac{y}{z} < 1 + \frac{\widehat{b}}{\widehat{b}+2}.$$

Hence, we get that

$$\frac{1}{\widehat{b}+1} = \frac{2}{2\widehat{b}+2} = \frac{\widehat{b}+2-\widehat{b}}{\widehat{b}+2+\widehat{b}} = \frac{1 - \frac{\widehat{b}}{\widehat{b}+2}}{1 + \frac{\widehat{b}}{\widehat{b}+2}} < \frac{\frac{x}{z}}{\frac{y}{z}} = \frac{x}{y} < \frac{1 + \frac{\widehat{b}}{\widehat{b}+2}}{1 - \frac{\widehat{b}}{\widehat{b}+2}} = \frac{\widehat{b}+2+\widehat{b}}{\widehat{b}+2-\widehat{b}} = \frac{2\widehat{b}+2}{2} = \widehat{b}+1.$$

Rearranging we obtain

$$-\widehat{b} < \frac{-\widehat{b}}{\widehat{b}+1} = \frac{1}{\widehat{b}+1} - 1 < \frac{x}{y} - 1 < \widehat{b}$$

which implies $|\frac{x}{y} - 1| < \widehat{b}$, contradicting our assumption. $\square$

**Lemma 32.** *Let* $f : \mathbb{R}^d \to \mathbb{R}$ *be a differentiable function. Consider the gradient flow dynamics induced by* $f$, *namely:*

$$\dot{\mathbf{x}}(t) = -\nabla f(\mathbf{x}(t))$$

*initialized at some* $\mathbf{x}_0 \in \mathbb{R}^d$. *Let* $t_1 < t_2$ *be times such that*

$$\{\mathbf{x}(t) : t \in [t_1, t_2]\} \subseteq \left\{\mathbf{z} \in \mathbb{R}^d : f(z) \geq \min_{\mathbf{y} \in \mathbb{R}^d} f(\mathbf{y}) + c\right\}$$

*for some* $c \geq 0$ *and* $f$ *satisfies the PL condition with some coefficient* $\mu > 0$ *in* $\{\mathbf{x}(t) : t \in [t_1, t_2]\}$. *Then*

$$f(\mathbf{x}(t_1)) - f(\mathbf{x}(t_2)) \geq 2(t_2 - t_1) \cdot \mu \cdot c.$$

*Proof.* By the fundamental theorem for line integrals we have

$$f\big(\mathbf{x}(t_1)\big) - f\big(\mathbf{x}(t_2)\big) = -\int_{t_1}^{t_2} \Big\langle \nabla f\big(\mathbf{x}(\tau)\big), \dot{\mathbf{x}}(\tau) \Big\rangle d\tau = \int_{t_1}^{t_2} \|\nabla f\big(\mathbf{x}(\tau)\big)\|^2 d\tau$$

applying the PL condition and Equation (24) we get the required result. □

**Lemma 33.** *Let $g : \mathbb{R}^d \to \mathbb{R}$ be a continuous function and let $r > 0$ and let $B$ be a compact set such that $\mathbf{0} \in B$. For any $\psi \geq 0$ denote the minimum value of $g$ over Diff$(\psi) \cap B$ as*

$$f(\psi) := \min_{\mathbf{x} \in Diff(\psi) \cap B} g(\mathbf{x}).$$

*It holds that $f$ is right side continuous in $\psi = 0$.*

*Proof.* Recalling the definition of Diff$(\psi)$ (Equation (17)), we have that for any $\psi \geq 0$ the set Diff$(\psi) \cap B$ is compact, thus $f(\psi)$ is properly defined for any $\psi \geq 0$ (as by continuity $g$ attains a minimum over the set). Next, note that $f$ is non-increasing since for any $\psi_2 \geq \psi_1 \geq 0$ it holds that

$$\text{Diff}(\psi_1) \cap B \subseteq \text{Diff}(\psi_2) \cap B \implies f(\psi_1) = \min_{\mathbf{x} \in \text{Diff}(\psi_1) \cap B} g(\mathbf{x}) \geq \min_{\mathbf{x} \in \text{Diff}(\psi_2) \cap B} g(\mathbf{x}) = f(\psi_2).$$

Let $\{\psi_n\}_{n=1}^{\infty}$ be a non-increasing sequence of non-negative reals for which

$$\lim_{n \to \infty} \psi_n = 0.$$

As $f$ is non-increasing, the sequence $\{f(\psi_n)\}_{n=1}^{\infty}$ is monotonically non-decreasing and upper bounded by $f(0)$. Hence, the limit $R := \lim_{n \to \infty} f(\psi_n)$ exists and satisfies $R \leq f(0)$. For any $\psi \geq 0$ we let $\mathbf{x}_\psi$ be a minimizer of $g$ over Diff$(\psi) \cap B$, *i.e.*

$$\mathbf{x}_\psi \in \operatorname*{argmin}_{\mathbf{x} \in \text{Diff}(\psi) \cap B} g(\mathbf{x}).$$

The set $B$ is compact and so the sequence $\{\mathbf{x}_{\psi_n}\}_{n=1}^{\infty}$ has a convergent subsequence $\{\mathbf{x}_{\psi_{n_k}}\}_{k=1}^{\infty}$. Denote its limit

$$\lim_{k \to \infty} \mathbf{x}_{\psi_{n_k}} =: \mathbf{x}^* \in \overline{B_r}(\mathbf{0}).$$

Assume on the contrary that $\mathbf{x}^* \notin \text{Diff}(0)$. Hence by definition of Diff$(0)$ it holds that

$$\widetilde{\psi} := \max_{i,j \in [d]} |x_i^* - x_j^*| > 0.$$

However, since $\lim_{n \to \infty} \psi_n = 0$ there exists $\widetilde{k} \in \mathbb{N}$ such that for any $k \in \mathbb{N}$ such that $k \geq \widetilde{k}$ it holds that $\psi_{n_k} \leq \frac{\widetilde{\psi}}{2}$ and thus

$$\mathbf{x}_{\psi_{n_k}} \in \text{Diff}(\psi_{n_k}) \subseteq \text{Diff}(\frac{\widetilde{\psi}}{2}) \wedge \mathbf{x}^* \notin \text{Diff}(\frac{\widetilde{\psi}}{2}).$$

This results in $\lim_{k \to \infty} \mathbf{x}_{\psi_{n_k}} \neq \mathbf{x}^*$ in contradiction. Therefore we obtain $\mathbf{x}^* \in \text{Diff}(0) \cap B$. By $g$'s continuity and $f$'s definition we thus obtain the following

$$\lim_{k \to \infty} g(\mathbf{x}_{\psi_{n_k}}) = g(\lim_{k \to \infty} \mathbf{x}_{\psi_{n_k}}) = g(\mathbf{x}^*) \geq f(0).$$

Per $\mathbf{x}_\psi$'s definition and since $\lim_{n \to \infty} f(\psi_n) = R$ we also obtain

$$\lim_{k \to \infty} g(\mathbf{x}_{\psi_{n_k}}) = \lim_{k \to \infty} f(\psi_{n_k}) = R,$$

*i.e.*, $R \geq f(0)$. Overall we obtain that $R \leq f(0) \leq R$, hence $R = f(0)$. The above result is satisfied for any sequence $\{\psi_n\}_{n=1}^{\infty}$, hence $\lim_{\psi \to 0^+} f(\psi) = f(0)$ as required. □

**Lemma 34.** *Let $f : \mathbb{R}^d \to \mathbb{R}^d$ be a $C^1$ and Lipschitz vector field. Consider the system of ODEs given by*

$$\dot{\mathbf{x}}(t) = f\big(\mathbf{x}(t)\big).$$

*Consider two initialization points $\mathbf{x}_1(0), \mathbf{x}_2(0) \in \mathbb{R}^d$. For any $t \geq 0$ we use $\lambda_{max}(t)$ to denote the maximum over the line segment between $\mathbf{x}_1(t)$ and $\mathbf{x}_2(t)$ of the maximal eigenvalue of the Jacobian of $f$, i.e.*

$$\lambda_{max}(t) := \max_{k \in [0,1]} \lambda_{max}\left(\nabla f\big(k \cdot \mathbf{x}_1(t) + (1 - k) \cdot \mathbf{x}_2(t)\big)\right).$$

*For any $t \geq 0$ it holds that*

$$\|\mathbf{x}_1(t) - \mathbf{x}_2(t)\|_2 \leq \exp\left(\int_0^t \lambda_{max}(\tau)d\tau\right) \cdot \|\mathbf{x}_1(0) - \mathbf{x}_2(0)\|_2.$$

*Proof.* First note that since $f$ is Lipschitz, $\lambda_{max}(t)$ is defined for any $t \geq 0$. Next, if $\mathbf{x}_1(0) = \mathbf{x}_2(0)$ then the trajectories coincide and so the claim trivially follows. Suppose $\mathbf{x}_1(0) \neq \mathbf{x}_2(0)$. By definition, we have that

$$\frac{d}{dt}\big(\mathbf{x}_1(t) - \mathbf{x}_2(t)\big) = f\big(\mathbf{x}_1(t)\big) - f\big(\mathbf{x}_2(t)\big).$$

As $f$ is $C^1$, we obtain by the mean value theorem (see [68]) that there exists $k \in [0,1]$ for which

$$f\big(\mathbf{x}_1(t)\big) - f\big(\mathbf{x}_2(t)\big) = \nabla f\big(k \cdot \mathbf{x}_1(t) + (1 - k) \cdot \mathbf{x}_2(t)\big)\big(\mathbf{x}_1(t) - \mathbf{x}_2(t)\big).$$

Additionally, by the chain rule we also have

$$\frac{d}{dt}\|\mathbf{x}_1(t) - \mathbf{x}_2(t)\|_2 = \frac{d}{dt}\sqrt{\big(\mathbf{x}_1(t) - \mathbf{x}_2(t)\big)^\top \big(\mathbf{x}_1(t) - \mathbf{x}_2(t)\big)}$$

$$= \frac{1}{2\|\mathbf{x}_1(t) - \mathbf{x}_2(t)\|_2} \frac{d}{dt}\big(\mathbf{x}_1(t) - \mathbf{x}_2(t)\big)^\top \big(\mathbf{x}_1(t) - \mathbf{x}_2(t)\big)$$

$$= \frac{2}{2\|\mathbf{x}_1(t) - \mathbf{x}_2(t)\|_2} \cdot \big(\mathbf{x}_1(t) - \mathbf{x}_2(t)\big)^\top \frac{d}{dt}\big(\mathbf{x}_1(t) - \mathbf{x}_2(t)\big)$$

$$= \frac{1}{\|\mathbf{x}_1(t) - \mathbf{x}_2(t)\|_2} \cdot \big(\mathbf{x}_1(t) - \mathbf{x}_2(t)\big)^\top \left(f\big(\mathbf{x}_1(t)\big) - f\big(\mathbf{x}_2(t)\big)\right).$$

Plugging the above yields

$$\frac{d}{dt}\|\mathbf{x}_1(t) - \mathbf{x}_2(t)\|_2 = \frac{\big(\mathbf{x}_1(t) - \mathbf{x}_2(t)\big)^\top \nabla f\big(k \cdot \mathbf{x}_1(t) + (1 - k) \cdot \mathbf{x}_2(t)\big)\big(\mathbf{x}_1(t) - \mathbf{x}_2(t)\big)}{\|\mathbf{x}_1(t) - \mathbf{x}_2(t)\|_2}$$

$$= \|\mathbf{x}_1(t) - \mathbf{x}_2(t)\|_2 \cdot \frac{\big(\mathbf{x}_1(t) - \mathbf{x}_2(t)\big)^\top \nabla f\big(k \cdot \mathbf{x}_1(t) + (1 - k) \cdot \mathbf{x}_2(t)\big)\big(\mathbf{x}_1(t) - \mathbf{x}_2(t)\big)}{\big(\mathbf{x}_1(t) - \mathbf{x}_2(t)\big)^\top \big(\mathbf{x}_1(t) - \mathbf{x}_2(t)\big)} = (*).$$

The right term is bound by the Rayleigh quotient (see [32]), and so the above can be bound by

$$(*) \leq \|\mathbf{x}_1(t) - \mathbf{x}_2(t)\|_2 \cdot \lambda_{max}\left(\nabla f\big(k \cdot \mathbf{x}_1(t) + (1 - k) \cdot \mathbf{x}_2(t)\big)\right) \leq \|\mathbf{x}_1(t) - \mathbf{x}_2(t)\|_2 \cdot \lambda_{max}(t)$$

where the last inequality stems from $\lambda_{max}(t)$'s definition. Dividing both sides by $\|\mathbf{x}_1(t) - \mathbf{x}_2(t)\|_2$ and integrating w.r.t time yields the following

$$\ln(\|\mathbf{x}_1(t) - \mathbf{x}_2(t)\|_2) - \ln(\|\mathbf{x}_1(0) - \mathbf{x}_2(0)\|_2) = \int_0^t \frac{\frac{d}{d\tau}\|\mathbf{x}_1(\tau) - \mathbf{x}_2(\tau)\|_2}{\|\mathbf{x}_1(\tau) - \mathbf{x}_2(\tau)\|_2}d\tau \leq \int_0^t \lambda_{max}(\tau)d\tau.$$

Reorganizing the inequality and taking exponents yields

$$\|\mathbf{x}_1(t) - \mathbf{x}_2(t)\|_2 \leq \exp\left(\int_0^t \lambda_{max}(\tau)d\tau + \ln(\|\mathbf{x}_1(0) - \mathbf{x}_2(0)\|_2)\right)$$

$$= \exp\left(\int_0^t \lambda_{max}(\tau)d\tau\right) \cdot \|\mathbf{x}_1(0) - \mathbf{x}_2(0)\|_2$$

as required. $\qquad\square$

**Lemma 35.** *Let $f : \mathbb{R}^d \to \mathbb{R}$ be a continuous non-negative function with $\min_{\mathbf{x}\in\mathbb{R}^d} f(\mathbf{x}) = 0$. Denote $X^* := \{\mathbf{x} \in \mathbb{R}^d : f(\mathbf{x}) = 0\}$. Suppose that the 1 sub-level set of $f$ defined as $\mathcal{L}_1(f) := \{\mathbf{x} \in \mathbb{R}^d : f(\mathbf{x}) \leq 1\}$ is compact. Then for any $\delta > 0$ there exists $\eta > 0$ such that for any $\mathbf{x} \in \mathbb{R}^d$ if $f(\mathbf{x}) \leq \eta$ then $\mathrm{Dist}(\mathbf{x}, X^*) \leq \delta$.*

*Proof.* Assume on the contrary that there exists $\delta > 0$ such that for any $\epsilon > 0$ there exists $\mathbf{x}_\eta \in \mathbb{R}^d$ for which $f(\mathbf{x}_\eta) \leq \delta$ and $\mathrm{Dist}(\mathbf{x}_\eta, X^*) > \delta$. Consider the sequence $\{\mathbf{x}_{\frac{1}{n}}\}_{n=1}^\infty$. For any $n \in \mathbb{N}$ it holds that $f(\mathbf{x}_{\frac{1}{n}}) \leq \frac{1}{n}$ and $\mathrm{Dist}(\mathbf{x}_{\frac{1}{n}}, X^*) > \delta$, therefore it holds that

$$\lim_{n\to\infty} f(\mathbf{x}_{\frac{1}{n}}) = 0 \,.$$

The sub-level set $\mathcal{L}_1(f)$ is compact and satisfies $\mathbf{x}_{\frac{1}{n}} \in \mathcal{L}_1(f)$ for any $n \in \mathbb{N}$, hence the sequence $\{\mathbf{x}_{\frac{1}{n}}\}_{n=1}^\infty$ is bounded. Therefore, the sequence has a convergent subsequence $\{\mathbf{x}_{\frac{1}{n_k}}\}_{k=1}^\infty$ with some limit $\mathbf{x}^* := \lim_{k\to\infty} \mathbf{x}_{n_k}$. By $f$'s continuity we get that

$$f(\mathbf{x}^*) = f(\lim_{k\to\infty} \mathbf{x}_{n_k}) = \lim_{k\to\infty} f(\mathbf{x}_{n_k}) = \lim_{n\to\infty} f(\mathbf{x}_n) = 0 \,,$$

*i.e.*, $\mathbf{x}^* \in X^*$. This is a contradiction since all $\mathbf{x}_{\frac{1}{n}}$ must remain at distance at least $\delta$ from $\mathbf{x}^*$ on the one hand, and $\mathbf{x}_{\frac{1}{n_k}}$ converges to $\mathbf{x}^*$ on the other hand. $\qquad\square$

**Lemma 36.** *Let $T, S > 0$, $n \in \mathbb{N}$ and $x^* \in (0, 1)$. There exist $q_1, w_1 \in (0, x^*)$ and $q_2, w_2 \in (\frac{1}{2}, 1)$ such that for any $x \in (q_1, w_1)$ and $y \in (q_2, w_2)$ it holds that*

$$1 - y \leq Tx$$

*and*

$$x \leq \frac{S}{y}(1 - y^n)^{\frac{1}{n}} \,.$$

*Proof.* First note that since $T > 0$, the first requirement is equivalent to having

$$\frac{1 - y}{T} \leq x \,.$$

Let $y \in (0, 1)$. It holds that

$$\lim_{y\to 1^-} \frac{S}{y}(1 - y^n)^{\frac{1}{n}} = \lim_{y\to 1^-} \frac{S}{y}\left((1 - y)\sum_{i=0}^{n-1} y^i\right)^{\frac{1}{n}} = S \cdot n^{\frac{1}{n}} \cdot \lim_{y\to 1^-}(1 - y)^{\frac{1}{n}} = 0 \,.$$

Therefore, there exists $y' \in (0, 1)$ such that for any $y \in [y', 1)$ it holds that

$$\frac{S}{y}(1 - y^n)^{\frac{1}{n}} \leq x^* \,.$$

On the other hand, it also holds that

$$\frac{\frac{S}{\frac{y+1}{2}}\left(1 - (\frac{y+1}{2})^n\right)^{\frac{1}{n}}}{\frac{1-y}{T}} = \frac{2TS}{y+1} \cdot \frac{\left((1 - \frac{y+1}{2})\sum_{i=0}^{n-1}(\frac{y+1}{2})^i\right)^{\frac{1}{n}}}{1 - y}$$

$$= \frac{2TS\left(\sum_{i=0}^{n-1}(\frac{y+1}{2})^i\right)^{\frac{1}{n}}}{y+1} \cdot (\frac{1}{2})^{\frac{1}{n}} \cdot \frac{(1-y)^{\frac{1}{n}}}{1 - y} \,.$$

Hence, taking the limit as $y \to 1^-$ we obtain that

$$\lim_{y\to 1^-} \frac{\frac{S}{\frac{y+1}{2}}\left(1 - (\frac{y+1}{2})^n\right)^{\frac{1}{n}}}{\frac{1-y}{T}} = \lim_{y\to 1^-} \frac{2TS\left(\sum_{i=0}^{n-1}(\frac{y+1}{2})^i\right)^{\frac{1}{n}}}{y+1} \cdot (\frac{1}{2})^{\frac{1}{n}} \cdot \frac{(1-y)^{\frac{1}{n}}}{1 - y}$$

$$= TS \cdot n^{\frac{1}{n}} \cdot (\frac{1}{2})^{\frac{1}{n}} \cdot \lim_{y\to 1^-} \frac{1}{(1-y)^{\frac{n-1}{n}}}$$

$$= \infty \,.$$

Therefore, there exists $y'' \in (0, 1)$ such that for any $y \in [y'', 1)$ it holds that

$$\frac{\frac{S}{\frac{y+1}{2}}\left(1 - \left(\frac{y+1}{2}\right)^n\right)^{\frac{1}{n}}}{\frac{1-y}{T}} > 2 \,.$$

and so $\frac{1-y}{T} < \frac{S}{\frac{y+1}{2}}\left(1 - \left(\frac{y+1}{2}\right)^n\right)^{\frac{1}{n}}$. Thus, setting $y^* = \max\{\frac{1}{2}, y', y''\}$ we obtain that the interval $\left(\frac{1-y^*}{T}, \frac{S}{\frac{y^*+1}{2}}\left(1 - \left(\frac{y^*+1}{2}\right)^n\right)^{\frac{1}{n}}\right)$ is not empty and upper bounded by $x^*$. Additionally, for any $y \in (y^*, \frac{y^*+1}{2})$ the following holds:

$$y^* < y \implies \frac{1-y^*}{T} > \frac{1-y}{T} \,,$$

$$\frac{y^*+1}{2} > y \implies \frac{S}{\frac{y^*+1}{2}}\left(1 - \left(\frac{y^*+1}{2}\right)^n\right)^{\frac{1}{n}} < \frac{S}{y}(1-y^n)^{\frac{1}{n}} \,.$$

Hence the interval $\left(\frac{1-y^*}{T}, \frac{S}{\frac{y^*+1}{2}}\left(1 - \left(\frac{y^*+1}{2}\right)^n\right)^{\frac{1}{n}}\right)$ is contained within the interval $\left(\frac{1-y}{T}, \frac{S}{y}(1 - (y^n)^{\frac{1}{n}})\right)$. Noting that $\frac{y^*+1}{2} < 1$, we complete the proof by setting

$$q_1 = \frac{1-y^*}{T} \quad , \quad w_1 = \frac{S}{\frac{y^*+1}{2}}\left(1 - \left(\frac{y^*+1}{2}\right)^n\right)^{\frac{1}{n}}$$

and

$$q_2 = y^* \quad , \quad w_2 = \frac{y^*+1}{2} \,.$$

$\square$

# I  Further Experimental Results

In this appendix we provide additional experimental results omitted from Section 4.

## I.1  Dynamical Characterization

The experiments in Section 4.1 corroborate our dynamical characterization (Section 3.1) by demonstrating that greedy low rank learning of the state transition matrix $A$ occurs under some, but not all, choices of training sequences. This appendix reports the following additional experiments.

- Figure 3 extends the experiments reported in Figure 2 to longer training sequences.
- Figures 4 to 6 include experiments similar to those in Figure 2, where the training sequences are labeled by teachers of higher dimensions.
- Figures 7 and 8 report the results achieved with different random seeds in the settings of Figures 2 and 3, respectively.
- A classical continuous surrogate for the matrix rank is the *effective rank* [67]. Figures 9 to 15 present the effective rank of the transition matrix $A$ throughout optimization in the settings of Figures 2 to 8, respectively, underlining the low effective rank caused by greedy low rank learning.
- Figures 16 to 19 report the values of $\gamma^{(0)}(t)$ (as defined in Proposition 1) observed during optimization in the settings of Figures 2, 3, 7, and 8, respectively, showcasing that larger absolute values of $\gamma^{(0)}(t)$ do not correspond to greedy low rank learning, whereas lower absolute values do.

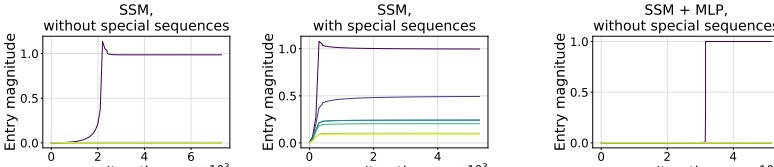

Figure 3: Demonstration of the dynamical characterization derived in Proposition 1—optimization of an SSM, trained individually or as part of a non-linear neural network, implicitly induces greedy learning of the (diagonal) entries of the state transition matrix $A$ under some, but not all, choices of training sequences. This figure is identical to Figure 2, except that the sequences used to train the models were longer, namely, of sequence length 10 as opposed to 6. For further details see Figure 2 as well as Appendix J.1.

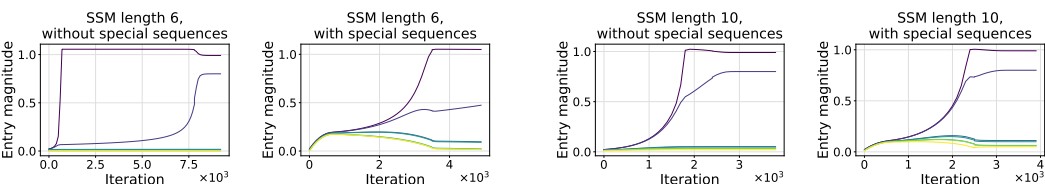

Figure 4: Demonstration of the dynamical characterization derived in Proposition 1—optimization of an individually trained SSM implicitly induces greedy learning of the (diagonal) entries of the state transition matrix $A$ under some, but not all, choices of training sequences. First two plots (left) and last two plots are identical to the first two plots in Figures 2 and 3 respectively, except that the teacher used to label the training sequences is of dimension $d^* = 2$ (as opposed to $d^* = 1$). For further details see Figures 2 and 3 as well as Appendix J.1.

## I.2 Synthetic Clean-Label Poisoning

The experiments in Table 1 corroborate the theoretical results of Section 3.2 by showcasing synthetic regimes in which SSMs are susceptible to clean-label poisoning. Table 3 supplements the experiments of Table 1 by including results in which the SSMs were trained on longer sequences. Tables 4 and 5 report the standard deviations of the quantities presented in Tables 1 and 3 respectively.

Table 3: Demonstration of clean-label poisoning of SSMs. The table is identical to Table 1, except that the sequences used to train the models were longer, namely, of sequence length 10 as opposed to 6. Notice that across all settings, special training sequences significantly deteriorate generalization. For further details see Table 1 as well as Appendix J.2.

| Setting | Without special sequences | With special sequences |
|---|---|---|
| Per Theorem 1 | $7.34 \times 10^{-3}$ | $3.51 \times 10^{-2}$ |
| Standalone SSM beyond Theorem 1 | $1.22 \times 10^{-1}$ | $1.2$ |
| SSM in non-linear neural network | $4.67 \times 10^{-2}$ | $8.93 \times 10^{-2}$ |

## I.3 Real-World Clean-Label Poisoning

The experiments in Table 2 corroborate the theoretical results of Section 3.2 by demonstrating clean-label poisoning of SSM-based neural networks in real-world settings. Table 6 provides additional context to the experiments of Table 2 by reporting the training performance of the trained models, before and after the inclusion of cleanly labeled poisonous examples. Tables 7 and 8 report the standard deviations of the quantities presented in Tables 2 and 6 respectively.

# J Implementation Details

We provide implementation details omitted from Section 4 and Appendix I. Code for reproducing our experiments can be found at `https://github.com/YoniSlutzky98/imp-bias-ssm-poison`.

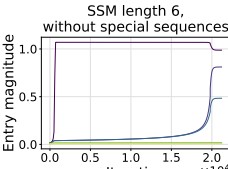
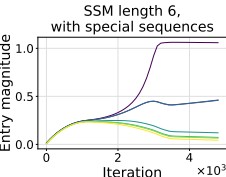
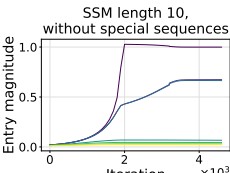
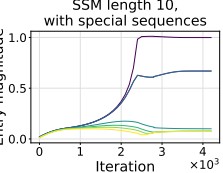

Figure 5: Demonstration of the dynamical characterization derived in Proposition 1. This figure is identical to Figure 4 except that the teacher used to label the training sequences is of dimension $d^* = 3$. For further details see Figure 4 and Appendix J.1.

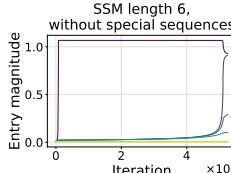
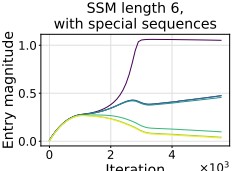
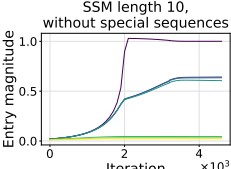
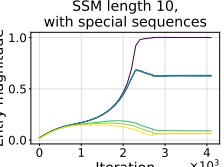

Figure 6: Demonstration of the dynamical characterization derived in Proposition 1. This figure is identical to Figure 4 except that the teacher used to label the training sequences is of dimension $d^* = 4$. For further details see Figure 4 and Appendix J.1.

## J.1 Dynamical Characterization

In this appendix we provide implementation details for the experiments of Section 4.1 and Appendix I.1. All experiments were implemented using Keras [10] and carried out on a single Nvidia RTX 2080 Ti GPU.

### J.1.1 Standalone SSM

**Models.** In the experiments reported in Figures 2, 3, 7, and 8 where a standalone SSM was trained, we used a teacher SSM of dimension $d^* = 1$ with parameters

$$A^* = 1 \ , \ \ B^* = 1 \ , \ \ C^* = 1 \, .$$

All parameters of the student SSM (*i.e.*, $A$, $B$ and $C$) were trained. We set the student SSM dimension to $d = 10$ for Figures 2 and 7 and to $d = 20$ for Figures 3 and 8.

Next, we detail the models used in the experiments with teachers of higher dimensions (Figures 4 to 6). In the experiments reported in Figure 4 we used a teacher SSM model of dimension 2 that was set with the parameters:

$$A^* = \begin{pmatrix} 0.99 & 0 \\ 0 & 0.8 \end{pmatrix} \ , \ \ B^* = \mathbf{1} \ , \ \ C^* = \mathbf{1}^\top \, .$$

In the experiments reported in Figure 5 we used a teacher SSM model of dimension 3 that was set with the parameters

$$A^* = \begin{pmatrix} 0.99 & 0 & 0 \\ 0 & 0.8 & 0 \\ 0 & 0 & 0.5 \end{pmatrix} \ , \ \ B^* = \mathbf{1} \ , \ \ C^* = \mathbf{1}^\top \, .$$

In the experiments reported in Figure 6 we used a teacher SSM model of dimension 3 that was set with the parameters

$$A^* = \begin{pmatrix} 0.99 & 0 & 0 & 0 \\ 0 & 0.8 & 0 & 0 \\ 0 & 0 & 0.5 & 0 \\ 0 & 0 & 0 & 0.3 \end{pmatrix} \ , \ \ B^* = \mathbf{1} \ , \ \ C^* = \mathbf{1}^\top \, .$$

We used student SSM models whose input and output matrices $B(\cdot)$ and $C(\cdot)$ were fixed at $\mathbf{1}$ and $\mathbf{1}^\top$ respectively (due to computational limitations). The student models had dimension $d = 10$ in the

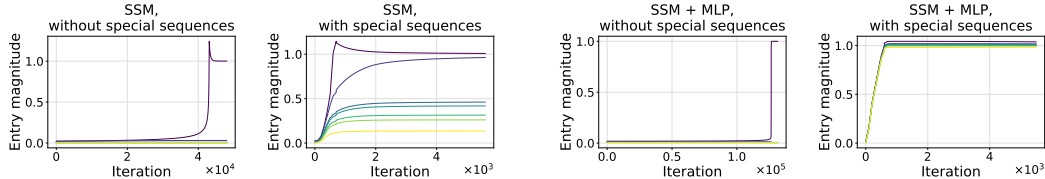

Figure 7: Demonstration of the dynamical characterization derived in Proposition 1. This figure is identical to Figure 2 except that a different random seed was used.

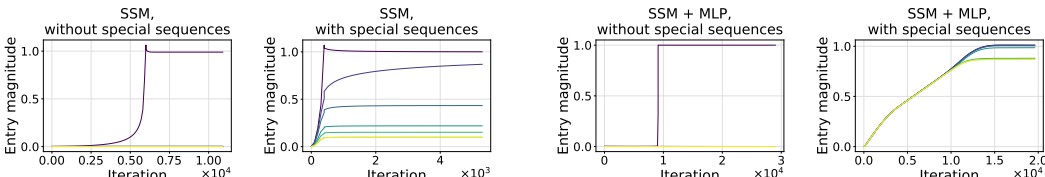

Figure 8: Demonstration of the dynamical characterization derived in Proposition 1. This figure is identical to Figure 3 except that a different random seed was used.

experiments of shorter sequence length and dimension $d = 20$ in the experiments of longer sequence length.

**Data.** In all experiments we used the respective teachers to generate the labels for training sequences. Additionally, we used training sequences of one of two types, referred to as "baseline" and "special", where each type had designated indices of non-zero entries. Table 9 specifies which non-zero indices were present in each sequence type for each experiment. Training sequences of both types had their non-zero entries sampled i.i.d. from $\mathcal{N}(0, 1)$. Table 10 specifies how many training sequences of each type were used in each experiment.

**Initialization.** In all experiments we initialized the student's $A$, $B$ and $C$ parameter matrices in a manner that was inspired by the initialization set $\mathcal{I}$ of Theorem 1.

To initialize (the diagonal) $A$ in each experiment we first sampled $d$ i.i.d. entries from $\mathcal{N}(0, \texttt{sd\_A})$, took their absolute values and then arranged them in descending order. Then, we set the second entry to have the first entry's value minus a constant $\texttt{diff}$. This was done to reflect the near zero initialization on the one hand and the proximity to the reference trajectory on the other hand. In the experiments reported in Figures 5 and 6 we naturally extended this procedure by setting the third entry to have the first entry's value minus $1.01 \cdot \texttt{diff}$ in both experiments, and the fourth entry to have the first entry's value minus $1.05 \cdot \texttt{diff}$ in the latter. Table 11 report the values of $\texttt{sd\_A}$ and $\texttt{diff}$ used in each experiment.

To initialize $B$ in each experiment we first sampled $d$ i.i.d. entries from $\mathcal{N}(0, \texttt{sd\_B\_C})$, took their absolute values and then arranged them in descending order. Then, we set the second entry to have the first entry's value minus a constant $\texttt{diff}$. This was done to reflect the near zero initialization on the one hand and the proximity to the reference trajectory on the other hand. In the experiments reported in Figures 5 and 6 we naturally extended this procedure by setting the third entry to have the first entry's value minus $1.01 \cdot \texttt{diff}$ in both experiments, and the fourth entry to have the first entry's value minus $1.05 \cdot \texttt{diff}$ in the latter. To initialize $C$ we followed the same procedure, without modifying the second to potentially fourth entries. Note that in the experiments reported in Figures 4 to 6 the input and output matrices $B(\cdot)$ and $C(\cdot)$ were not trained. Table 11 report the values of $\texttt{sd\_B\_C}$ used in each experiment.

**Optimization.** In all of the experiments we trained using the empirical mean squared error as a loss function and optimized over full batches of the training sets. In order to facilitate more efficient experimentation in the experiments where a standalone SSM is trained, we optimized using gradient descent with an adaptive learning rate scheme, where at each iteration a base learning rate is divided by the square root of an exponential moving average of squared gradient norms (see appendix D.2 in Razin et al. [66] for more details). We used a weighted average coefficient of $\beta = 0.8$ and a softening constant of $10^{-6}$. Note that only the learning rate (step size) is affected by this scheme, not the direction of movement. Comparisons between the adaptive scheme and optimization with a fixed

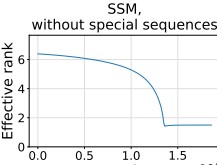 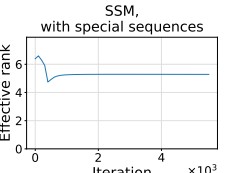 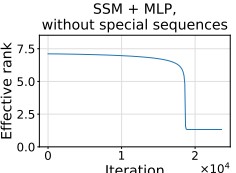 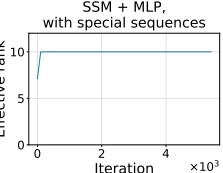

Figure 9: Demonstration of the dynamical characterization derived in Proposition 1 through the lens of the effective rank of $A$—introduction of special sequences to the tranining set results in significantly larger effective rank, in compliance with the disruption of greedy low rank learning. Each plot shows the effective rank of $A$ during the training process reported in Figure 2.

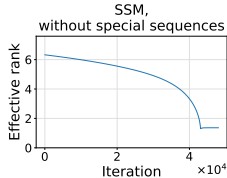 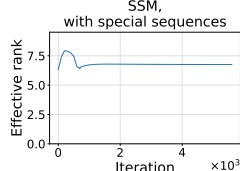 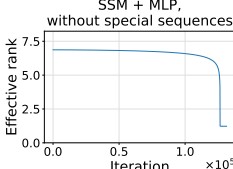 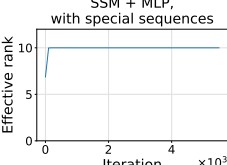

Figure 10: Demonstration of the dynamical characterization derived in Proposition 1 through the lens of the effective rank of $A$. This figure is identical to Figure 9 except that the setting considered is that of Figure 7.

learning rate showed no significant difference in terms of the dynamics, while run times of the former were considerably shorter. Table 12 specifies the base learning rate used in each experiment.

### J.1.2 SSM in a Non-Linear Neural Network

**Models.** In the experiments reported in Figures 2, 3, 7, and 8 where an SSM was trained as a part of a non-linear neural network, we used neural networks comprised of an SSM module whose output was fed to a 2 hidden layers MLP module. Overall, the models used realize the following function:

$$D_{out} \cdot \sigma\left(D_{hidden} \cdot \sigma\left(D_{in} \cdot \phi_{A,B,C}(\mathbf{x})\right)\right)$$

where $d_h \in \mathbb{N}$ is the hidden MLP width, $D_{in} \in \mathbb{R}^{d_h,1}$, $D_{hidden} \in \mathbb{R}^{d_h,d_h}$ and $D_{out} \in \mathbb{R}^{1,d_h}$ are the MLP's parameter matrices, $\sigma(x) := \max\{0, x\}$ is the MLP's activation function and $\phi_{A,B,C}(\mathbf{x})$ is the output of an SSM with the parameter matrices $A, B, C$. All teacher models used had SSM modules of dimension $d^* = 1$ that were set with the parameters

$$A^* = 1 \ , \ \ B^* = 1 \ , \ \ C^* = 1 \, .$$

The teacher models in Figures 2 and 7 had hidden MLP width of $d_h^* = 15$ while the teacher models in Figures 3 and 8 had hidden MLP width of $d_h^* = 25$. In both cases, the teacher models had MLP modules that were set with the following parameter matrices:

$$D_{in}^* = \mathbf{1} \ , \ \ D_{hidden}^* = I_{d_h^*} \ , \ \ D_{out}^* = \frac{1}{2} \cdot \mathbf{1}^\top \, .$$

We trained all of the SSM and MLP parameter matrices of our student models. The student models in Figures 2 and 7 had SSM dimension of $d = 10$ and hidden MLP width of $d_h = 15$, while the student models in Figures 3 and 8 had SSM dimension of $d = 20$ and hidden MLP width of $d_h = 25$.

**Data.** Data for the experiments were generated in the same manner as in Appendix J.1.1. Table 9 specifies which non-zero indices were present in each sequence type for each experiment. Training sequences of both types had their non-zero entries sampled i.i.d. from $\mathcal{N}(0, 1)$. Table 10 specifies how many training sequences of each type were used in each experiment.

**Initialization.** In all experiments we initialized the student's $A$, $B$ and $C$ parameter matrices identically to the standalone SSM experiments (Appendix J.1.1). Table 11 report the values of `sd_A`, `sd_B_C` and `diff` used in each experiment.

To initialize the MLP layers, we initialized all parameter matrices by sampling i.i.d. values from $\mathcal{N}(0, \mathtt{sd\_D})$. We used $\mathtt{sd\_D} = 0.03$ in the original experiments (Figures 2 and 7) and $\mathtt{sd\_D} = 0.1$ in the experiments with longer sequences (Figures 3 and 8).

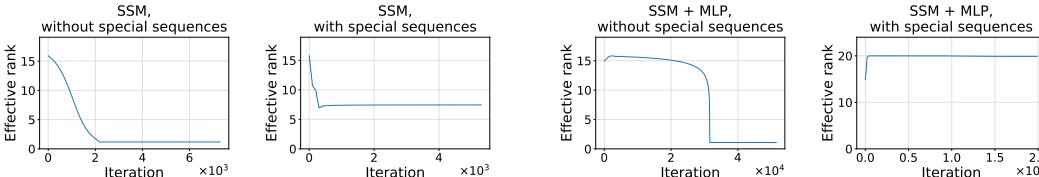

Figure 11: Demonstration of the dynamical characterization derived in Proposition 1 through the lens of the effective rank of $A$. This figure is identical to Figure 9 except that the setting considered is that of Figure 3.

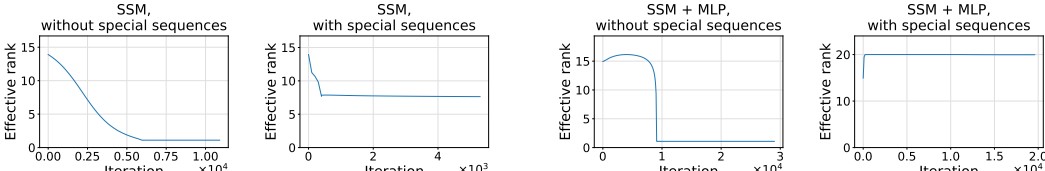

Figure 12: Demonstration of the dynamical characterization derived in Proposition 1 through the lens of the effective rank of $A$. This figure is identical to Figure 9 except that the setting considered is that of Figure 8.

**Optimization.** To speed up the optimization we trained using the default Keras implementation of Adam [36] with its default hyperparameters. Table 12 report the base learning rates used in each of the experiments.

## J.2  Synthetic Clean-Label Poisoning

In this appendix we provide implementation details for the synthetic experiments provided in Section 4.2 and Appendix I.2. All experiments were implemented using Keras [10] and were carried out using a single Nvidia RTX 2080 Ti GPU.

### J.2.1  SSM Per Theorem 1

The main goal of the experiments in the first poisoning setting (standalone SSM per Theorem 1) was to approximate the solution to the system of ODEs induced by gradient flow (Equation (4)) in order to demonstrate the results of Theorem 1. To do so we employed the use of the `odeint` function of SciPy [84] which is a numerical solver for systems of ODEs based on `lsoda` from the FORTRAN library odepack [31]. `odeint`'s arguments are the initial point in parameter space $A(0)$, the timestamps at which the solution is required, and a function which specifies the system by intaking a timestamp $t$ and a point in parameter space $A$ and outputting the derivative in time $t$ at $A$. `odeint` outputs a set of numerical approximations for the solution of the system at the required timestamps.

**Models.** We use teacher and student models according to the setting described in Section 3.2. We used a teacher SSM of dimension $d^* = 2$ that was set with the parameters

$$A^* = \begin{pmatrix} 1 & 0 \\ 0 & 0 \end{pmatrix} \ , \ B^* = \begin{pmatrix} 1 & \sqrt{d-1} \end{pmatrix}^\top \ , \ C^* = \begin{pmatrix} 1 & \sqrt{d-1} \end{pmatrix} .$$

We used student SSM models whose input and output matrices $B(\cdot)$ and $C(\cdot)$ were fixed at $\mathbf{1}$ and $\mathbf{1}^\top$ respectively. The student models had dimension $d = 10$ in the original poisoning experiments (Table 1), and dimension $d = 20$ when training over longer sequences (Table 3).

**Data.** In all experiments we used the respective teachers to generate the labels for all training sequences. Per Theorem 1, we trained using a single "baseline" sequence of the form $\mathbf{e}_1 \in \mathbb{R}^\kappa$ and a single "special" sequence of the form $\mathbf{e}_{\kappa-1} \in \mathbb{R}^\kappa$. The relevant experiments reported in Tables 1 and 3 were trained using sequences of lengths 7 and 9 respectively.

**Initialization.** We initialized the student's diagonal matrix $A$ in a manner that was inspired by the initialization set $\mathcal{I}$ of Theorem 1. In each experiment we first sampled $d$ i.i.d. entries from $\mathcal{N}(0, \mathtt{sd\_A})$ and then arranged them in descending order. We then set the second entry to have the first's value minus a constant $\mathtt{diff}$. This was done to reflect the near zero initialization on the one hand and the

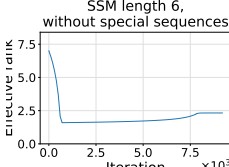
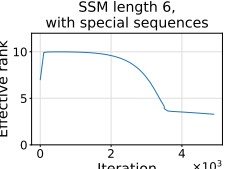
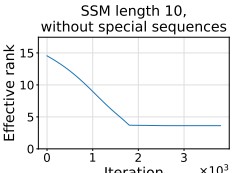
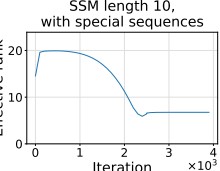

Figure 13: Demonstration of the dynamical characterization derived in Proposition 1 through the lens of the effective rank of $A$. This figure is identical to Figure 9 except that the setting considered is that of Figure 4.

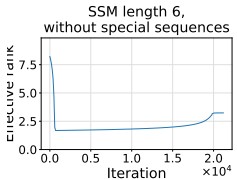
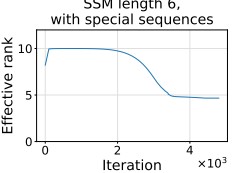
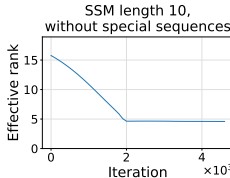
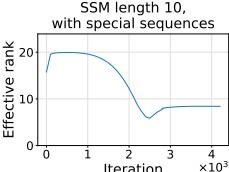

Figure 14: Demonstration of the dynamical characterization derived in Proposition 1 through the lens of the effective rank of $A$. This figure is identical to Figure 9 except that the setting considered is that of Figure 5.

proximity to the reference trajectory on the other hand. Table 13 reports the values of `sd_A` and `diff` used in each experiment.

**Optimization.** We input `odeint` a custom function which computes $-\nabla \ell_{\mathcal{S}}(A)$ (Equation (4)) given the point $A$ to be used for derivative computations. Table 14 reports the timestamps simulated in each experiment. All experiments reached training loss values of less than $10^{-5}$ and stable generalization errors.

**Generalization evaluation.** Generalization errors were measured via impulse responses of length $40$ as defined in Definition 1, divided by the $\ell_{\infty}$ norm of the teacher's length $40$ impulse response (such that the zero mapping has error of one). The same evaluation procedures were used in both the original experiments (Table 1) and in the longer experiments (Table 3).

### J.2.2 SSM Beyond Theorem 1

**Models.** We used the same teacher models as described in Appendix J.1.1. We used student SSM models that were trained end to end (*i.e.* $B(\cdot)$ and $C(\cdot)$ were not fixed). The student models had dimension $d = 10$ in the original poisoning experiments (Table 1), and dimension $d = 20$ in the experiments with longer sequences (Table 3).

**Data.** Data for the experiments were generated in the same manner as in Appendix J.1.1. Table 15 specifies which non-zero indices were present in each sequence type for each experiment. Training sequences of both types had their non-zero entries sampled i.i.d. from $\mathcal{N}(0, 1)$. Table 16 specifies how many training sequences of each type were used in each experiment.

**Initialization.** To speed up optimization we modified the initialized employed in Appendix J.1.1 by adding a small constant of $10^{-1}$ to all entries of $A$ and $B$ at initialization. This modification showed no significant differences in terms of the generalization error achieved by the models when compared to without it, while run times of the former were considerably shorter. Table 17 report the values of `sd_A`, `sd_B_C` and `diff` used in each experiment.

**Optimization.** We followed a training scheme identical to Appendix J.1.1. Table 18 report the base learning rates used in each of the experiments.

We optimized all models to reach a training loss under $0.01$. To verify the generalization errors we report were stable, we trained for additional iterations after reaching sub $0.01$ training loss. We trained the standalone SSM models for $1500$ more iterations, and the models with additional layers for $5000$ more iterations.

**Generalization evaluation.** Generalization errors were measured via impulse responses of length $40$ as defined in Definition 1, divided by the $\ell_{\infty}$ norm of the teacher's length $40$ impulse response

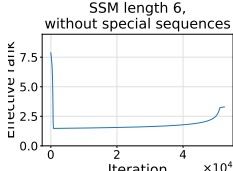 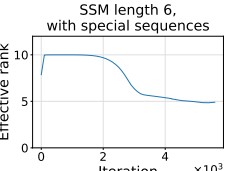 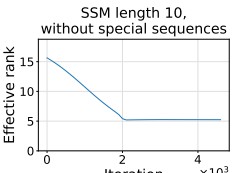 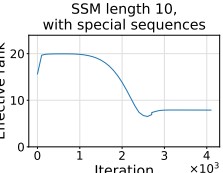

Figure 15: Demonstration of the dynamical characterization derived in Proposition 1 through the lens of the effective rank of $A$. This figure is identical to Figure 9 except that the setting considered is that of Figure 6.

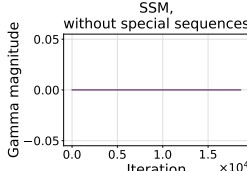 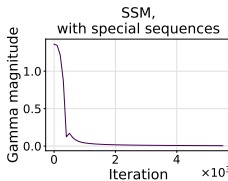 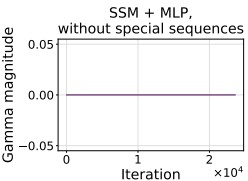 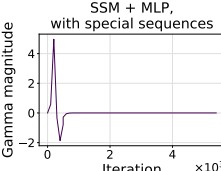

Figure 16: Evolution of $\gamma^{(0)}(t)$ (as defined in Proposition 1) during training. Each plot shows the values of $\gamma^{(0)}(t)$ during optimization in the setting of Figure 2. As can be seen, in compliance with our interpretation of Proposition 1, larger absolute values of $\gamma^{(0)}(t)$ do not correspond to greedy low rank learning, whereas lower absolute values do.

(such that the zero mapping has error of one). The same evaluation procedures were used in both the original experiments (Table 1) and in the longer experiments (Table 3).

### J.2.3 SSM in Non-Linear Neural Network

**Models.** We used the same teacher models as described in Appendix J.1.2. We used student SSM models that were trained end to end (*i.e.* $B(\cdot)$ and $C(\cdot)$ were not fixed). The student models had dimension $d = 10$ in the original poisoning experiments (Table 1), and dimension $d = 20$ in the experiments with longer sequences (Table 3).

**Data.** The data used is identical to that of Appendix J.2.2. Table 16 specify how many training sequences of each type were used in each experiment.

**Initialization.** We initialized the student models identically to Appendix J.1.2. To speed up optimization we modified the initialized employed in Appendix J.1.2 by adding a small constant of $10^{-3}$ to all entries of $A$ and $B$ at initialization. This modification showed no significant differences in terms of the generalization error achieved by the models when compared to without it, while run times of the former were considerably shorter. Table 17 report the values of `sd_A`, `sd_B_C` and `diff` used in each experiment.

**Optimization.** We followed a training scheme identical to Appendix J.1.2. Table 18 report the base learning rates used in each of the experiments.

We optimized all models to reach a training loss under $0.01$. To verify the generalization errors we report were stable, we trained for additional iterations after reaching sub $0.01$ training loss. We trained the standalone SSM models for $1500$ more iterations, and the models with additional layers for $5000$ more iterations.

**Generalization evaluation.** Generalization errors were measured via the root mean square error of a held-out test set of $2000$ correctly labeled sequences of length $40$, divided by the $\ell_2$ norm of the teacher's outputs vector (such that the zero mapping has error of one). The same evaluation procedures were used in both the original experiments (Table 1) and in the longer experiments (Table 3).

### J.3 Real-World Clean-Label Poisoning

In this appendix we provide implementation details for the real-world experiments provided in Section 4.2 and Appendix I.3. All experiments were implemented using PyTorch [60] and were carried out using a cluster of $8$ Nvidia RTX A6000 GPUs. All experiments consist of three stages: (i)

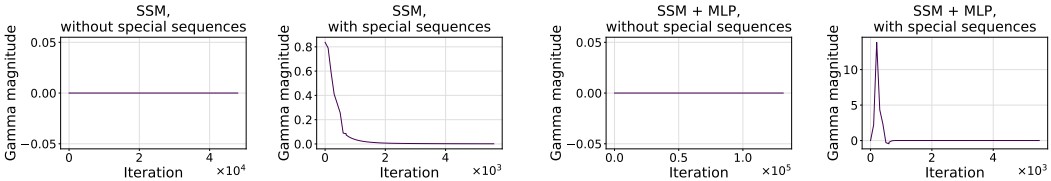

Figure 17: Evolution of $\gamma^{(0)}(t)$ (as defined in Proposition 1) during training. This figure is identical to Figure 16 except that the setting considered is that of Figure 7.

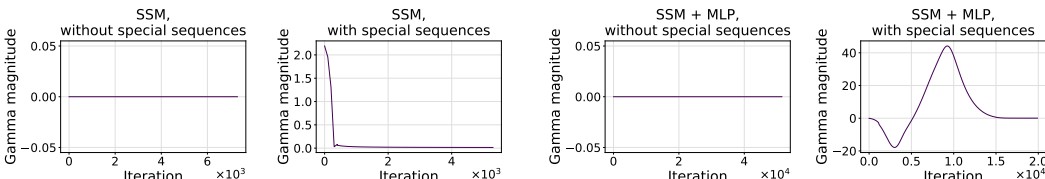

Figure 18: Evolution of $\gamma^{(0)}(t)$ (as defined in Proposition 1) during training. This figure is identical to Figure 16 except that the setting considered is that of Figure 3.

training the attacked model from scratch on a set of 20000 randomly chosen CIFAR-10 train examples, (ii) generating poisonous noise added to a random subset of 10000 train examples using an adapted version of the Gradient Matching method [18] provided in `https://github.com/JonasGeiping/poisoning-gradient-matching`, and (iii) retraining the model on the 30000-examples poisoned train set consisting of the original 10000 examples used for poison generation in their clean form and in their poisoned form, as well as the remaining 10000 original examples. Due to computational limitations, the poison generation process was targeted to disrupt the classification of 3000 randomly chosen test instances, and the poisoning effect carried over to the entire CIFAR-10 test set.

To allow for poisoning of the entirety of the test set, we adapted the Gradient Matching method by strategically partitioning the poisoning budget—namely, 5000 examples were used to poison the 1000 correctly-classified test instances for which the original model's loss was the highest, and the remaining 5000 examples were used to poison the remaining 2000 test instances. Additionally, in order to make poisoning more precise we further split the poisoning examples into groups such that the noise trained for each group's examples targets 50 test instances and ignores the rest. Our adapted approach also included: (a) reshaping the CIFAR-10 image input into a compatible sequence format, (b) introducing regularization that encourages the last elements of an injected noise sequence to be relatively large [15], and (c) using the `untargeted-cross-entropy` target criterion and a `pbatch` value of 500. Apart from the above adaptations, all hyperparameters were kept at their default values.

The SSM-based S4 neural network adheres to the implementation provided in `https://github.com/state-spaces/s4`, utilizing the "minimalist S4" configuration available in `s4d.py` and `s4.py`. The SSM-based Mamba-2 neural network is an adaptation of the minimal implementation provided in `https://github.com/tommyip/mamba2-minimal`. The SSM-based LRU neural network is a python adaptation of the JAX-implementation provided in `https://github.com/NicolasZucchet/minimal-LRU`. We trained all models using the AdamW optimizer and the `CosineAnnealingLR` scheduler over 100 epochs. To ensure the models before and after poisoning were comparable, we kept training the latter until reaching the training loss of the former (but no longer than 300 total epochs). Table 19 details the hyperparameters used for each of the models.

---

[15]Regularization comprised weight decay of $32 - i$ applied to the noise entries corresponding to the $i$th row of an input image across all three channels, where $i \in [32]$ (recall that CIFAR-10 images are of size $3 \times 32 \times 32$).

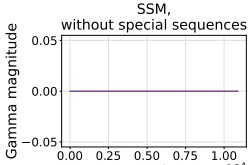 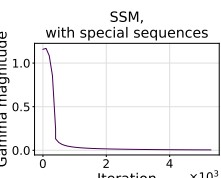 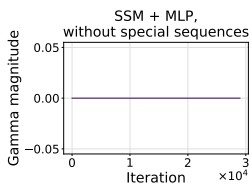 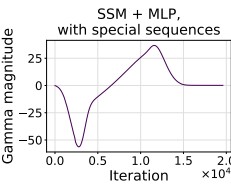

Figure 19: Evolution of $\gamma^{(0)}(t)$ (as defined in Proposition 1) during training. This figure is identical to Figure 16 except that the setting considered is that of Figure 8.

Table 4: Standard deviations of the quantities presented in Table 1. For further details see Table 1 and Appendices I.2 and J.2.

| Setting | Without special sequences | With special sequences |
|---|---|---|
| Per Theorem 1 | $8.2 \times 10^{-5}$ | $3.54 \times 10^{-3}$ |
| Standalone SSM beyond Theorem 1 | 0.195 | 9.56 |
| SSM in non-linear neural network | $1.88 \times 10^{-3}$ | $3.83 \times 10^{-2}$ |

Table 5: Standard deviations of the quantities presented in Table 3. For further details see Table 3 and Appendices I.2 and J.2.

| Setting | Without special sequences | With special sequences |
|---|---|---|
| Per Theorem 1 | $2.71 \times 10^{-4}$ | $1.69 \times 10^{-4}$ |
| Standalone SSM beyond Theorem 1 | $4.3 \times 10^{-2}$ | 0.635 |
| SSM in non-linear neural network | $4.67 \times 10^{-2}$ | $1.99 \times 10^{-2}$ |

Table 6: Demonstration of clean-label poisoning of SSMs in real-world (non-synthetic) settings comprising SSM-based neural networks consisting of the S4, Mamba-2 or LRU layers [24, 14, 58] trained on the CIFAR-10 dataset [38]. This table is similar to Table 2, except that it reports the training performance of the trained models, before and after the inclusion of cleanly labeled poisonous examples. Notice that across all experiments, the models' training performance remains consistent before and after the inclusion of cleanly labeled poisonous examples. Consequently, the drop in test performance reported in Table 2 cannot be attributed to a change in training performance. For further details see Table 2 and Appendices I.3 and J.3.

| SSM-based NN | Loss without / with poisoning | Accuracy without / with poisoning |
|---|---|---|
| S4 [24] | 0.316 / 0.316 | 88.6% / 88.7% |
| Mamba-2 [14] | $6.59 \times 10^{-2}$ / $6.59 \times 10^{-2}$ | 97.9% / 97.8% |
| LRU [58] | 0.334 / 0.299 | 89.1% / 90% |

Table 7: Standard deviations of the quantities presented in Table 2. For further details see Table 2 and Appendices I.3 and J.3.

| SSM-based NN | Loss without / with poisoning | Accuracy without / with poisoning | Noise tail size |
|---|---|---|---|
| S4 [24] | $1.47 \times 10^{-2}$ / $1.71 \times 10^{-2}$ | 0.29% / 0.18% | $1.64 \times 10^{-2}$ |
| Mamba-2 [14] | $3.97 \times 10^{-2}$ / $4.38 \times 10^{-2}$ | 1.32% / 0.67% | $2.1 \times 10^{-2}$ |
| LRU [58] | $4.76 \times 10^{-3}$ / $2.96 \times 10^{-2}$ | 0.71% / 0.53% | $5.52 \times 10^{-2}$ |

Table 8: Standard deviations of the quantities presented in Table 6. For further details see Table 6 and Appendices I.3 and J.3.

| SSM-based NN | Loss without / with poisoning | Accuracy without / with poisoning |
|---|---|---|
| S4 [24] | $1.25 \times 10^{-2}$ / $1.35 \times 10^{-2}$ | 0.45% / 0.61% |
| Mamba-2 [14] | $1.95 \times 10^{-2}$ / $1.98 \times 10^{-2}$ | 0.75% / 0.68% |
| LRU [58] | $0.62 \times 10^{-2}$ / $0.5 \times 10^{-2}$ | 2.4% / 1.76% |

Table 9: Types of sequences used in dynamics experiments (Figures 2 to 8). Last two columns (right) indicate the non-zero indices for each sequence type.

| Setting | Length | Baseline indices | Special indices |
|---|---|---|---|
| SSM / SSM + MLP (Figures 2 and 7) | 6 | $1, 2$ | $5, 6$ |
| SSM / SSM + MLP longer (Figures 3 and 8) | 10 | $1, 2, \ldots, 7$ | $9, 10$ |
| SSM higher dimension (Figures 4 to 6) | 6 | $1, 2$ | $5$ |
| SSM higher dimension longer (Figures 4 to 6) | 10 | $1, 2, \ldots, 7$ | $9$ |

Table 10: Amount of sequences of each type used in dynamics experiments (Figures 2 to 8).

| Setting | Baseline amount | Special amount |
|---|---|---|
| SSM (Figures 2, 3, 7, and 8) | 8 | 10 |
| SSM + MLP (Figures 2, 3, 7, and 8) | 20 | 20 |
| SSM higher dimension (Figures 4 to 6) | 8 | 10 |

Table 11: Values of `sd_A`, `sd_B_C` and `diff` used in the dynamics experiments (Figures 2 to 8).

| Setting | sd_A | sd_B_C | diff |
|---|---|---|---|
| SSM (Figures 2 and 7) | $10^{-2}$ | $10^{-2}$ | $0.05 \times \exp\big(20 \cdot \log_{10}(\texttt{sd\_A})\big)$ |
| SSM longer (Figures 3 and 8) | $10^{-3}$ | $5 \times 10^{-2}$ | $0.05 \times \exp\big(20 \cdot \log_{10}(\texttt{sd\_A})\big)$ |
| SSM + MLP (Figure 7) | $10^{-2}$ | $10^{-2}$ | $0.05 \times \exp\big(0.5 \cdot \log_{10}(\texttt{sd\_A})\big)$ |
| SSM + MLP longer (Figure 8) | $10^{-3}$ | $10^{-3}$ | $0.05 \times \exp\big(2 \cdot \log_{10}(\texttt{sd\_A})\big)$ |
| SSM higher dimension (Figures 4 to 6) | $10^{-2}$ | – | $0.05 \times \exp\big(2 \cdot \log_{10}(\texttt{sd\_A})\big)$ |
| SSM higher dimension longer (Figures 4 to 6) | $10^{-2}$ | – | $0.05 \times \exp\big(3 \cdot \log_{10}(\texttt{sd\_A})\big)$ |

Table 12: Base learning rates used in dynamics experiments (Figures 2 to 8). Last two columns (right) indicate the base learning rate used in the experiments without the addition of "special" sequences and with their addition respectively.

| Setting | Without special sequences | With special sequences |
|---|---|---|
| SSM (Figures 2 and 7) | 0.01 | 0.01 |
| SSM longer (Figures 3 and 8) | 0.01 | 0.01 |
| SSM + MLP (Figures 2 and 7) | 0.01 | 0.001 |
| SSM + MLP longer (Figures 3 and 8) | 0.01 | $5 \times 10^{-5}$ |
| SSM higher dimension (Figures 4 to 6) | 0.01 | 0.001 |
| SSM higher dimension longer (Figures 4 to 6) | 0.001 | 0.001 |

Table 13: Values of `sd_A` and `diff` used in the experiments of the first poisoning setting (Tables 1 and 3).

| Setting | sd_A | diff |
|---|---|---|
| Per Theorem 1 (Table 1) | $10^{-3}$ | $0.05 \times \exp\big(5 \cdot \log_{10}(\texttt{sd\_A})\big)$ |
| Per Theorem 1 longer (Table 3) | $5 \times 10^{-3}$ | $0.05 \times \exp\big(10 \cdot \log_{10}(\texttt{sd\_A})\big)$ |

Table 14: Timestamps simulated for the experiments of the first poisoning setting (Tables 1 and 3). The timestamps used for each experiment are the endpoints of intervals obtained by evenly partitioning the range $(0, \texttt{last\_timestamp})$ into `timestamp_amount` segments.

| Setting | last_timestamp | timestamp_amount |
|---|---|---|
| Per Theorem 1 (Table 1) w/o special | $10^{11}$ | 1000 |
| Per Theorem 1 (Table 1) w/ special | $10^{4}$ | 1000 |
| Per Theorem 1 longer (Table 3) w/o special | $10^{13}$ | 10000 |
| Per Theorem 1 longer (Table 3) w/ special | $10^{7}$ | 1000 |

Table 15: Types of sequences used in the experiments of the second and third poisoning settings (Tables 1 and 3). Last two columns (right) indicate the non-zero indices for each sequence type.

| Setting | Length | Baseline indices | Special indices |
|---|---|---|---|
| $2^{nd}$ and $3^{rd}$ settings of Table 1 | 6 | $1, 2$ | 5 |
| $2^{nd}$ and $3^{rd}$ settings of Table 3 | 10 | $1, 2, \ldots, 7$ | 9 |

Table 16: Amount of sequences of each type used in in the experiments of the second and third poisoning settings (Tables 1 and 3).

| Setting | Baseline amount | Special amount |
|---|---|---|
| Standalone SSM beyond Theorem 1 (Tables 1 and 3) | 8 | 10 |
| SSM in non-linear neural network (Tables 1 and 3) | 20 | 20 |

Table 17: Values of sd_A, sd_B_C and diff used in the experiments of the second and third poisoning settings (Tables 1 and 3).

| Setting | sd_A | sd_B_C | diff |
|---|---|---|---|
| Standalone SSM beyond Theorem 1 (Table 1) | $10^{-3}$ | $10^{-3}$ | $0.05 \times \exp\big(5 \cdot \log_{10}(\texttt{sd\_A})\big)$ |
| Standalone SSM beyond Theorem 1 longer (Table 3) | $10^{-2}$ | $10^{-3}$ | $0.05 \times \exp\big(3 \cdot \log_{10}(\texttt{sd\_A})\big)$ |
| SSM in non-linear neural network (Table 1) | $10^{-2}$ | $10^{-2}$ | $0.05 \times \exp\big(0.5 \cdot \log_{10}(\texttt{sd\_A})\big)$ |
| SSM in non-linear neural network longer (Table 3) | $10^{-3}$ | $10^{-3}$ | $0.05 \times \exp\big(2 \cdot \log_{10}(\texttt{sd\_A})\big)$ |

Table 18: Base learning rates used in the experiments of the second and third poisoning settings (Tables 1 and 3). Last two columns (right) indicate the base learning rate used in the experiments without the addition of "special" sequences and with their addition respectively.

| Setting | Without special sequences | With special sequences |
|---|---|---|
| Standalone SSM beyond Theorem 1 (Table 1) | 0.01 | 0.01 |
| Standalone SSM beyond Theorem 1 longer (Table 3) | 0.001 | 0.001 |
| SSM in non-linear neural network (Table 1) | 0.01 | 0.01 |
| SSM in non-linear neural network longer (Table 3) | 0.01 | $5 \times 10^{-5}$ |

Table 19: Hyperparameters used in the real-world poisoning experiments (Tables 2 and 6).

| Model | Depth | Dropout | d_model | d_state | Base learning rate | Weight decay |
|---|---|---|---|---|---|---|
| S4 | 4 | 0.1 | 128 | 64 | $10^{-2}$ | $10^{-2}$ |
| Mamba-2 | 4 | 0.1 | 128 | 16 | $3 \times 10^{-3}$ | $5 \times 10^{-2}$ |
| LRU | 4 | 0.1 | 128 | 64 | $10^{-3}$ | $5 \times 10^{-2}$ |