# OpenReview forum: "The Implicit Bias of Structured State Space  Models Can Be Poisoned With Clean Labels"
_NeurIPS.cc/2025/Conference — NeurIPS 2025 spotlight_

### Official Review · Reviewer_gug3 · 2025-07-02

**Clarity:** 3
**Significance:** 3
**Originality:** 3
**Rating:** 5
**Confidence:** 3

**Summary:**

The paper proves that a single, correctly-labeled “special” sequence can force a structured state-space model trained with gradient descent to memorize rather than generalize, revealing a provable clean-label poisoning vulnerability. It derives this result via a novel polynomial-ODE analysis of the training dynamics and corroborates it with S4 experiments on CIFAR-10 showing a >40× error spike when <12 % of the data are poisoned.

**Questions:**

### Questions for the Authors
1. Does the attack persist under HiPPO or identity initializations?
2. What happens when \(B,C\) are learned with separate learning rates?
3. How does the phenomenon change with SGD + momentum or Adam?
4. Minimum poison fraction needed for million-sample datasets?
5. Can analogous “suffix-heavy” inputs poison language/audio tasks without trailing zeros?
6. Do weight decay or spectral-norm regularization mitigate the effect?
7. Are attention-plus-SSM hybrids (e.g., Mamba) equally vulnerable?
8. Could monitoring the low-rank signal \(\gamma^{(0)}(t)\) act as an early-warning defense?

**Ethical Concerns:**

["NO or VERY MINOR ethics concerns only"]

**Final Justification:**

From my original reading of the paper as after reading others reviews I believe my original assessment was correct. I still hold that this paper should be an acceptable.

**Quality:**

3

**Strengths And Weaknesses:**

### Strengths
1. **Fills a theoretical gap** – extends clean-label poisoning theory to non-convex SSMs.
2. **Mechanistic insight** – ODE factorization offers an intuitive, entry-wise view of dynamics.
3. **Finite-sample realism** – avoids infinite-data assumptions of earlier implicit-bias work.
4. **Balanced analysis** – provides both success and failure results under the same framework.
5. **Concrete experiments** – ties theory to measurable failure on a real SSM benchmark.

### Weaknesses
- Low-dimensional teacher & diagonal \(A\) may not reflect practical high-dim/banded SSMs.
- Fixed \(B,C\) matrices – real models usually learn these.
- Near-zero initialization – differs from HiPPO/identity inits.
- Special input padding (trailing zeros) – unnatural for real data.
- Gradient-flow analysis – ignores discrete optimizers, momentum, Adam, etc.
- Experiments are small-scale; no language or large data regimes.

---

> ### Author Rebuttal · Authors · 2025-07-30
>
> Thank you for highlighting various strengths of our theory and experiments!
>
> While our paper avoids restrictive assumptions made in prior work (e.g., it avoids infinite data assumptions, as you have mentioned), it is still limited in different ways, including those you mention. We have made a significant effort to be fully transparent about the limitations of the paper, throughout the text and in a dedicated limitations section (Section 5).
>
> Below we address your questions by their enumeration.
>
> Q1 + Q3:
>
> Our proof of clean-label poisoning does not apply to HiPPO or identity initializations, nor does it apply to SGD with momentum or Adam optimizers. However, our real-world (non-synthetic) experiment demonstrates clean-label poisoning for an SSM-based S4 neural network trained using standard near-identity initialization and AdamW optimization. Extending our proof to account for further initializations and optimizers is an important direction for future work.
>
> Q2:
>
> Our proof of clean-label poisoning requires B and C to either be fixed throughout training (Assumption 2), or be learned with a very small learning rate (Appendix B). However, our real-world (non-synthetic) experiment, as well as most of our synthetic experiments, demonstrate clean-label poisoning where B and C are learned with the same learning rate as A. Extending our proof to account for A, B and C learned with the same learning rate is an important direction for future work.
>
> Q4:
>
> Our theory does not imply what is the minimum amount of cleanly-labeled poisonous examples that can facilitate a successful attack in a practical large-scale scenario. Our real-world (non-synthetic) experiment demonstrates successful attacks with various amounts of cleanly-labeled poisonous examples, but we do not know what is the minimum amount with which this is possible. It is an interesting direction to explore.
>
> Q5:
>
> In practical scenarios we typically do not have a teacher (labeling function) at hand, and the way in which clean-label poisoning can be achieved is addition of imperceptible noise to training examples. Our real-world (non-synthetic) experiment demonstrates this on a vision task (CIFAR-10), with noise whose structure is “suffix-heavy” and without trailing zeros.
>
> Q6:
>
> Analyses of implicit bias, i.e., of implicit regularization, typically consider training objectives that do not include explicit regularization (see, for example, [1, 2]). The rationale behind this is that explicit regularization co-mingles with implicit regularization, rendering it difficult to associate phenomena with the latter. Indeed, in the extreme case of very strong explicit regularization (e.g., L2 regularization with a very large coefficient), the training objective will admit a single minimizer, and implicit regularization will have no impact on the solution to which optimization converges.
> Having said the above, studying implicit regularization does make sense in cases of weak to moderate explicit regularization. Indeed, our real-world (non-synthetic) experiment accounts for such explicit regularization. Extending our theory to do so is an interesting direction for future work.
>
> Q7:
>
> While we do not provide a theoretical answer to this question, we are extending our real-world (non-synthetic) experiment to include additional neural architectures, in particular Mamba. The evaluation of Mamba is still ongoing, but thus far, all of its results fully align with those of S4. We will add the extended experimentation to the camera-ready version of the paper. Thank you for raising this question!
>
> Q8:
>
> This is an excellent question! As you suggest, our Proposition 1 implies that training dynamics may be an informative tool for detecting the existence of clean-label poisoning attacks (a capability that can facilitate various countermeasures, the simplest being to prevent deployment of the compromised model). Namely, Proposition 1 implies that if the factor $\gamma^{(0)}(\cdot)$ is small throughout training, then generalization (with a low dimensional teacher) takes place, whereas if $\gamma^{(0)}(\cdot)$ is not small, then generalization may not take place. Figures 16 to 19 (in Appendix I) empirically demonstrate this phenomenon for a standalone SSM as well as one included in a non-linear neural network. A very promising avenue for future research is to extend this observation to practical scenarios, i.e., to identify, for SSMs in practical scenarios, properties of training dynamics that characterize when clean-label poisoning takes place.
>
> An expanded version of the above discussion will appear in the conclusion of the camera-ready version of the paper. We view this as a valuable addition to the text. Thank you for the question!
>
> ---
> **References**
>
> [1] Gunasekar et al. "Implicit Regularization in Matrix Factorization"
>
> [2] Soudry et al. "The Implicit Bias of Gradient Descent on Separable Data"

---

> > ### Comment · Reviewer_gug3 · 2025-08-01
> >
> > I thank the authors for answering my questions. I believe if the authors clean up the text and add what they have promised on the camera ready, it will make for a good paper.

---

> > > ### Author Response · Authors · 2025-08-03
> > >
> > > Thank you for your support! Your thoughtful feedback helped us improve the text.

---

### Official Review · Reviewer_XKJx · 2025-07-02

**Clarity:** 3
**Significance:** 3
**Originality:** 3
**Rating:** 5
**Confidence:** 3

**Summary:**

This paper presents an interesting theoretical study stating the susceptibility of clean-label poisoning in distilling structured state space models. The study starts by analyzing and stating the gradient flow convergence in Proposition.1, known as the analysis of dynamical characterization of SSM . Furthermore in Theorem.1, this study shows both bright and down side of distilling from a low-rank teacher model: training a student SSM with a low dimensional teacher SSM can improve implicit bias, which leads to generalization. However if we further add a training time series sample with its last entries carrying large norms, the distilled SSM may end up with being overfitted to the training data and show poor generalization performances.

**Questions:**

Please discuss if the findings can be used over other model architectures like SSM. Please also provide discussions about how the findings can be used to improve generalization performances.

**Ethical Concerns:**

["NO or VERY MINOR ethics concerns only"]

**Final Justification:**

Thanks for the responses and I have also gone through the rebuttal messages to other reviewers' comments. I would recommend to accept this paper as my final justification.

**Limitations:**

Yes

**Quality:**

3

**Strengths And Weaknesses:**

**Strong points of this paper**
This paper presents a novel and interesting finding: distilled SSM can present poor generalization performances if teacher SSMs have a low-ranked structure, even though the training data samples used are free from poisoning. This study shows the generalization limitation of knowledge distillation in SSMs. I really appreciate the solid theoretical reasoning step by step presented in this paper.

**Weakness of this paper**

It seems this paper focuses on exclusively on SSMs. I don't know if the conclusion made in this paper can be also applied to other model architectures like transformers (probably it can, but needs further discussions). This, somehow, may limit the practical impact of this work.
Furthermore, I'd be curious, if the findings unveiled in this paper can be also used to improve the generalization performance. For example, what should we do to ensure the generalization capability of distilled model? Do we need to choose a high-rank teacher or we need to check the training data samples for distillation?

---

> ### Author Rebuttal · Authors · 2025-07-30
>
> Thank you for highlighting that our findings are novel and interesting, and for appreciating their solid theoretical reasoning!  We address your reservations below.
>
> >It seems this paper focuses on exclusively on SSMs. I don't know if the conclusion made in this paper can be also applied to other model architectures like transformers (probably it can, but needs further discussions). This, somehow, may limit the practical impact of this work.
>
> Clean‑label poisoning has been demonstrated empirically in several non-SSM neural architectures, most notably in CNN-based image classifiers (see, e.g., [1, 2, 3, 4, 5]). Theoretically, clean-label poisoning was—to our knowledge—only explored in convex settings such as linear or kernel models ([6, 7]), or in settings where poisonous examples come instead of (rather than in addition to) original training examples ([8, 9, 10]). To our knowledge, there are no formal guarantees for clean-label poisoning via added examples in Transformers or other attention-based architectures. Extending our analysis to account for such architectures is an ambitious and interesting direction for future work.
>
> >I'd be curious, if the findings unveiled in this paper can be also used to improve the generalization performance. For example, what should we do to ensure the generalization capability of distilled model? Do we need to choose a high-rank teacher or we need to check the training data samples for distillation?
>
> With regards to improving generalization in practical scenarios, Proposition 1 in our paper suggests that training dynamics may be an informative tool for detecting the existence of clean-label poisoning attacks (a capability that can facilitate various countermeasures, the simplest being to prevent deployment of the compromised model). Namely, Proposition 1 implies that if the factor $\gamma^{(0)}(\cdot)$ is small throughout training, then generalization (with a low dimensional teacher) takes place, whereas if $\gamma^{(0)}(\cdot)$ is not small, then generalization may not take place. Figures 16 to 19 (in Appendix I) empirically demonstrate this phenomenon for a standalone SSM as well as one included in a non-linear neural network. A very promising avenue for future research is to extend this observation to practical scenarios, i.e., to identify, for SSMs in practical scenarios, properties of training dynamics that characterize when clean-label poisoning takes place.
>
> An expanded version of the above discussion will appear in the conclusion of the camera-ready version of the paper.  We view this as a valuable addition to the text. Thank you for your comment!
>
> ---
> **References**
>
> [1] Shafahi et al. “Poison Frogs! Targeted Clean-Label Poisoning Attacks on Neural Networks”
>
> [2] Zhu et al. “Transferable Clean-Label Poisoning Attacks on Deep Neural Nets”
>
> [3] Zhao et al. “Clean-Label Backdoor Attacks on Video Recognition Models”
>
> [4] Huang et al. “MetaPoison: Practical General-purpose Clean-label Data Poisoning”
>
> [5] Aghakhani et al. “Bullseye Polytope: A Scalable Clean-Label Poisoning Attack with Improved Transferability”
>
> [6] Suya et al. “Model-Targeted Poisoning Attacks with Provable Convergence”
>
> [7] Blum et al. “Robust Learning Under Clean-Label Attack”
>
> [8] Mahloujifar et al. “Can Adversarially Robust Learning Leverage Computational Hardness?”
>
> [9] Mahloujifar et al. “The Curse of Concentration in Robust Learning: Evasion and Poisoning Attacks from Concentration of
> Measure”
>
> [10] Gao et al. “Learning and Certification under Instance-targeted Poisoning”

---

> > ### Author Response · Authors · 2025-08-05
> >
> > Dear Reviewer,
> >
> > We would greatly appreciate your feedback on our response.  Please let us know if there is any additional information that may be of assistance.
> >
> > Thank you!
> >
> > Authors

---

> ### Author Response · Authors · 2025-08-08
>
> Dear Reviewer **XKJx**,
>
> Thank you again for your thoughtful and positive review. We have carefully addressed each of your comments in our rebuttal. Please let us know if any additional information may be helpful. With the discussion period closing soon, we kindly ask that you confirm acceptance of this message.
>
> Thanks again for your time and consideration.
>
> Best regards,
>
> Authors

---

### Official Review · Reviewer_wYKm · 2025-07-13

**Clarity:** 2
**Significance:** 2
**Originality:** 2
**Rating:** 4
**Confidence:** 3

**Summary:**

This paper prove that while implicit bias leads to generalization under many choices of training data, there exist special examples whose inclusion in training completely distorts the implicit bias, to a point where generalization fails.

**Questions:**

See Strengths And Weaknesses*

**Ethical Concerns:**

["NO or VERY MINOR ethics concerns only"]

**Final Justification:**

Overall, I think this paper is well written. But keep some concerns on experimental results with large-scale SSMs.

So, my final suggestion is 'Borderline accept'

**Limitations:**

See Strengths And Weaknesses*

**Quality:**

2

**Strengths And Weaknesses:**

Strengths:
1) This paper proposes a formal proof of clean-label poisoning in SSMs, extending the understanding of implicit bias in overparameterized models.
2) The use of advanced tools such as the non-resonance linearization theorem brings fresh insights into training dynamics.
3) The dynamical characterization of gradient flow for SSMs is carefully derived and shows under what conditions greedy low-rank learning occurs.
4) Theorems clearly illustrate the existence of cases where adding cleanly labeled sequences induces severe generalization failures.
5) Experiments range from controlled synthetic setups to real-world settings (e.g., CIFAR-10 with S4 models), confirming the theoretical claims.

Weaknesses and Suggestions:
1) While theoretically elegant, it is not always clear how easily an attacker could craft or inject sequences with the required properties in practice. I suggest including more concrete examples or algorithmic procedures to illustrate the feasibility and cost of such attacks in realistic scenarios.

2) While the experiments are diverse, most are conducted with small SSM dimensions and short sequence lengths due to computational challenges. It remains unclear whether the observed phenomena would persist in large-scale SSMs used for long-context language modeling.
I understand the author said "due to vanishing gradients, maintaining reasonable run-times in experiments with near-zero initialization necessitated a relatively small scale (in terms of, e.g., SSM dimension and training sequence length). Addressing these limitations is an important set of directions for future work."  I suggest add some experiments using other initialization methods to experiment on large-scale SSMs.

3) Although the paper establishes the existence of clean-label poisoning vulnerabilities, it does not propose mitigation strategies or even partial defenses. For practitioners, it is unclear what safeguards could be applied to SSM training pipelines.
Include at least a short discussion of potential countermeasures or research directions for making SSMs more robust in practice.

---

> ### Author Rebuttal · Authors · 2025-07-30
>
> Thank you for highlighting the merits of our theory, in particular that it “extend[s] the understanding of implicit bias in overparameterized models” and “brings fresh insights into training dynamics”! Thank you also for noting that our “experiments are diverse,” “confirming the theoretical claims”. We address your reservations below.  If you find our responses satisfactory, we would greatly appreciate it if you would consider raising your score.
>
> >While theoretically elegant, it is not always clear how easily an attacker could craft or inject sequences with the required properties in practice. I suggest including more concrete examples or algorithmic procedures to illustrate the feasibility and cost of such attacks in realistic scenarios.
>
> Our theoretically established cleanly labeled poisonous examples require a teacher (labeling function) in order to be generated, thus are not applicable as-is to most practical scenarios.  However, our real-world (non-synthetic) experiments demonstrate that clean label poisoning can be achieved via addition of human-imperceptible noise whose structure is inspired by our theory. While we defer to future work a thorough empirical exploration of clean-label poisoning techniques born from our theory, we are adding more experiments to the camera-ready version of the paper—see response below.
>
> >While the experiments are diverse, most are conducted with small SSM dimensions and short sequence lengths due to computational challenges. It remains unclear whether the observed phenomena would persist in large-scale SSMs used for long-context language modeling. I understand the author said "due to vanishing gradients, maintaining reasonable run-times in experiments with near-zero initialization necessitated a relatively small scale (in terms of, e.g., SSM dimension and training sequence length). Addressing these limitations is an important set of directions for future work." I suggest add some experiments using other initialization methods to experiment on large-scale SSMs.
>
> Following your feedback, we are conducting experiments in additional real-world (non-synthetic) settings, including large-scale SSMs in Mamba2 and LRU architectures. While these experiments are still ongoing, all of their results thus far fully align with those of S4 on CIFAR-10. We will add the experiments to the camera-ready version of the paper. Thank you!
>
> >Although the paper establishes the existence of clean-label poisoning vulnerabilities, it does not propose mitigation strategies or even partial defenses. For practitioners, it is unclear what safeguards could be applied to SSM training pipelines. Include at least a short discussion of potential countermeasures or research directions for making SSMs more robust in practice.
>
> Proposition 1 in our paper suggests that training dynamics may be an informative tool for detecting the existence of clean-label poisoning attacks (a capability that can facilitate various countermeasures, the simplest being to prevent deployment of the compromised model). Namely, Proposition 1 implies that if the factor $\gamma^{(0)}(\cdot)$ is small throughout training, then generalization (with a low dimensional teacher) takes place, whereas if $\gamma^{(0)}(\cdot)$ is not small, then generalization may not take place. Figures 16 to 19 (in Appendix I) empirically demonstrate this phenomenon for a standalone SSM as well as one included in a non-linear neural network. A very promising avenue for future research is to extend this observation to practical scenarios, i.e., to identify, for SSMs in practical scenarios, properties of training dynamics that characterize when clean-label poisoning takes place.
>
> An expanded version of the above discussion will appear in the conclusion of the camera-ready version of the paper. We view this as a valuable addition to the text. Thank you very much for raising the matter!

---

> > ### Comment · Reviewer_wYKm · 2025-08-06
> >
> > 1. I can not well understand 'clean label poisoning can be achieved via addition of human-imperceptible noise whose structure is inspired by our theory', could you provide more explainations?
> >
> > 2. 'While these experiments are still ongoing, all of their results thus far fully align with those of S4 on CIFAR-10. We will add the experiments to the camera-ready version of the paper. '  , Could you please sharing how long it will take?

---

> ### Author Response · Authors · 2025-08-07
>
> Thank you for the follow-up! We address your questions below. Please let us know if any additional information may be helpful. Otherwise, if you find our response satisfactory, we would be grateful if you would consider raising your score.
>
> 1. In our real-world (non-synthetic) experiment, we do not have access to a teacher (i.e., to a ground truth labeling mapping). Therefore, in order to create poisonous examples, we contaminate original training examples with human-imperceptible additive noise (this ensures that a contaminated example has the same label as its original counterpart). The additive noise has relatively large last elements, in line with our theory.
>
> 2. Additional experiments with Mamba-2 have concluded, and ones with LRU are now coming to an end. The results of those experiments are qualitatively the same as those of S4. Per the suggestion of reviewer **Y3v5**, we are conducting further experiments with more targeted examples. With our existing computational resources we expect these latter experiments to finish in 2-3 weeks.

---

> ### Comment · Reviewer_wYKm · 2025-08-08
>
> >In our real-world (non-synthetic) experiment, we do not have access to a teacher (i.e., to a ground truth labeling mapping). Therefore, in order to create poisonous examples, we contaminate original training examples with human-imperceptible additive noise (this ensures that a contaminated example has the same label as its original counterpart). The additive noise has relatively large last elements, in line with our theory.
>
> Thank you for your response. I understand.
>
> >Additional experiments with Mamba-2 have concluded, and ones with LRU are now coming to an end. The results of those experiments are qualitatively the same as those of S4.
>
> Could you kindly show them in the comments area?

---

> > ### Author Response · Authors · 2025-08-09
> >
> > Thank you for your response. Below are tables showing analogous results to those in Table 2 obtained for Mamba2 and LRU respectively.
> >
> > ---
> >
> > **Mamba2**
> >
> > | # test instances    | # poison examples| Last elements rel. size| Acc. without poison | Acc. with poison |
> > |----------|-------|------|-------|------|
> > | 10   | 500   | 0.999    | 80%    | 10%    |
> > | 10      | 2500    | 0.999    | 80%    | 10%    |
> > | 10 | 5000    | 0.999    | 90%    | 0%    |
> > | 20   | 500   | 0.999    | 90%    | 10%    |
> > | 20      | 2500    | 0.999    | 85%    | 10%    |
> > | 20 | 5000    | 0.999    | 70%    | 5%    |
> >
> >
> >
> > **LRU**
> >
> > | # test instances    | # poison examples| Last elements rel. size| Acc. without poison | Acc. with poison |
> > |----------|-------|------|-------|------|
> > | 10   | 500   | 0.999    | 70%    | 0%    |
> > | 10      | 2500    | 0.999    | 60%    | 10%    |
> > | 10 | 5000    | 0.999    | 80%    | 0%    |
> > | 20   | 500   | 0.999    | 75%    | 0%    |
> > | 20      | 2500    | 0.999    | 85%    | 0%    |
> > | 20 | 5000    | 0.999    | 80%    | 20%    |

---

### Official Review · Reviewer_Y3v5 · 2025-07-18

**Clarity:** 1
**Significance:** 4
**Originality:** 4
**Rating:** 4
**Confidence:** 3

**Summary:**

The highlights a challenge in the generalization of an SSM model. The paper demonstrates that the conventional argument of SSM can be generalized, and that implicit bias can be challenged, while also illustrating how clean label poisoning can compromise the model's generalization capabilities. The paper's conclusions are supported by rigorous theoretical analysis.

**Questions:**

Please refer to the weakness section.

**Ethical Concerns:**

["NO or VERY MINOR ethics concerns only"]

**Final Justification:**

I have read the responses of the authors and other reviewers. While I still hold some reservations regarding how the new experiments will look, I am fine with accepting the paper and have increased my score.

**Limitations:**

Yes

**Quality:**

3

**Strengths And Weaknesses:**

Strength:

1. The paper provides a detailed theoretical analysis behind each of the hypotheses and backs them with proofs.
2. This is the first work to highlight an essential problem of SSM by successfully showing a clean-label poisoning attack on SSM.
3. The proposed method, backed by each theorem, is novel. A non-resonant linearization theorem yields two critical observations that facilitate the successful creation of a clean label poisoning attack.

Weakness:

1. While the paper is theoretically solid, it indeed has several experimental limitations. I appreciate the author being upfront about the scope of the work being limited; however, without further experimental validation, it is challenging to thoroughly stress-test the proposed theoretical analysis.

2. Two core challenges to the generalization of the proposed method, which the authors have not mentioned, are the choice of architecture and dataset. The evaluation is only conducted on one architecture and one dataset. The experimental setting is minimal. Why not evaluate on a few state space models to show the conclusion holds true across the board, such as Mambas?

3. Even with the current choice, there is no rationale behind the options presented in the result section. First, why only the SSM-S4 model? Second, in Table 2, why are only 10/20 instances being attacked? What is the challenge in attacking all the test samples, similar to any conventional adversarial attack evaluation? Also, this is not a standard way to present results. It is very confusing to look at Table 2 and see the fourth column accuracy without poison to be that low. It takes a lot of work on the reader's part to understand that these are accuracies only on the # of test samples. Furthermore, the actual accuracy of the model is not at all reported in the paper. I would appreciate more results with clarity in the presentation.

---

> ### Author Rebuttal · Authors · 2025-07-30
>
> Thank you for highlighting that our paper “provides a detailed theoretical analysis behind each of the hypotheses” and “is the first work to highlight an essential problem of SSM by successfully showing a clean-label poisoning attack”! We address your reservations below. If you find our responses satisfactory, we would greatly appreciate it if you would consider raising your score.
>
> >While the paper is theoretically solid, it indeed has several experimental limitations. I appreciate the author being upfront about the scope of the work being limited; however, without further experimental validation, it is challenging to thoroughly stress-test the proposed theoretical analysis.
>
> >Two core challenges to the generalization of the proposed method, which the authors have not mentioned, are the choice of architecture and dataset. The evaluation is only conducted on one architecture and one dataset. The experimental setting is minimal. Why not evaluate on a few state space models to show the conclusion holds true across the board, such as Mambas?
>
> Thank you for highlighting that our paper is theoretically solid, and that we were upfront about the scope of its experiments being limited.  We would like to note, however, that to our knowledge, compared to formal theoretical papers such as ours, the scope of our experimentation is rather broad.  Nonetheless, following your comments, we are conducting experiments in additional real-world (non-synthetic) settings, including Mamba2 and LRU architectures. While these experiments are still ongoing, all of their results thus far fully align with those of S4 on CIFAR-10. We will add the experiments to the camera-ready version of the paper. Thank you!
>
> >Even with the current choice, there is no rationale behind the options presented in the result section. First, why only the SSM-S4 model? Second, in Table 2, why are only 10/20 instances being attacked? What is the challenge in attacking all the test samples, similar to any conventional adversarial attack evaluation? Also, this is not a standard way to present results. It is very confusing to look at Table 2 and see the fourth column accuracy without poison to be that low. It takes a lot of work on the reader's part to understand that these are accuracies only on the # of test samples. Furthermore, the actual accuracy of the model is not at all reported in the paper. I would appreciate more results with clarity in the presentation.
>
> Thank you for the feedback.
>
> Regarding real-world (non-synthetic) experimentation only with S4, please see our response above on experiments in additional settings, including Mamba2 and LRU.
>
> With regards to your comments surrounding Table 2:
> To our knowledge, in the area of adversarial machine learning, it is common to employ clean-label poisoning for attacking a small number of specific test instances ([1, 2]). In this case the overall test accuracy is barely impacted by the attack, which is precisely the intent of the attacker (their desire is for the attack to go undetected after training, when test accuracy is being evaluated).  While this regime is (to our knowledge) common, we understand your point about it being different from the regime of our theory and synthetic experiments (namely, the regime in which clean-label poisoning leads to significant deterioration in the overall test accuracy).  Accordingly, we are now extending our real-world (non-synthetic) experiments to include attacks on more test instances. While this extension is still ongoing, initial results confirm that the overall test accuracy indeed deteriorates.  In the camera-ready version of the paper, Table 2 will report overall test accuracies.  We believe this change will address your request for more clarity in its presentation.  Thank you very much!
>
> ---
>
> **References**
>
> [1] Huang et al. “MetaPoison: Practical General-purpose Clean-label Data Poisoning”
>
> [2] Geiping et al. “Witches' Brew: Industrial Scale Data Poisoning via Gradient Matching”

---

### Note · Authors · 2025-08-16

Dear Committee,

We sincerely thank you for the time and constructive feedback provided during the review and discussion process.  Many of your suggestions have helped us improve the clarity and quality of the manuscript, and we are incorporating them into the camera-ready version as discussed.

Sincerely,

Authors

---

### Decision · Program_Chairs · 2025-09-17

**Decision:**

Accept (spotlight)

**Comment:**

This paper is the first to identify a fundamental problem of SSMs by demonstrating a clean-label poisoning attack. It provides detailed theoretical analysis using mathematical techniques such as Polynomial-ODE analysis and a non-resonant linearization theorem. In addition, convincing experimental results are presented, and the authors further strengthen their case in the rebuttal by adding results on additional architectures.

Overall, this is a very solid paper with strong theoretical foundations. All reviewers expressed positive opinions about this work.